# Role of the dew water on the ground surface in HONO distribution: a case measurement in Melpitz

**Yangang Ren[1], Bastian Stieger[2], Gerald Spindler[2], Benoit Grosselin[1], Abdelwahid Mellouki[1*], Thomas Tuch[2], Alfred Wiedensohler[2], Hartmut Herrmann[2*],**

1. Institut de Combustion, Aérothermique, Réactivité et Environnement (ICARE), CNRS (UPR 3021), Observatoire des Sciences de l'Univers en région Centre (OSUC), 1C Avenue de la Recherche Scientifique, 45071 Orléans Cedex 2, France

2. Leibniz Institute for Tropospheric Research (TROPOS), Permoserstraße 15, 04318 Leipzig, Germany

* Corresponding author: Abdelwahid Mellouki (abdelwahid.mellouki@cnrs-orleans.fr) and Hartmut Herrmann (herrmann@tropos.de)

**Abstract:** To characterize the role of dew water for the ground/surface HONO distribution, nitrous acid (HONO) measurements with a MARGA and a LOPAP instrument were performed at the TROPOS research site in Melpitz from April 19th to 29th, 2018. The dew water was also collected and analyzed from May 8th to 14th, 2019 using a glass sampler. The high time resolution of HONO measurements showed characteristic diurnal variations that revealed: (i) vehicle emission is a minor source of HONO at the Melpitz station; (ii) heterogeneous conversion of $NO_2$ to HONO on ground surface dominates HONO production at night; (iii) there is significant nighttime loss of HONO with a sink strength of $0.16 \pm 0.12$ $ppbv\ h^{-1}$; (iv) dew water with mean $NO_2^-$ of $7.91 \pm 2.14$ $\mu g\ m^{-2}$ could serve as a temporary HONO source in the morning when the dew droplets evaporate. The nocturnal observations of HONO and $NO_2$ allowed direct evaluation of the ground uptake coefficients for these species at night: $\gamma_{NO2 \rightarrow HONO} = 2.4 \times 10^{-7}$ to $3.5 \times 10^{-6}$, $\gamma_{HONO,ground} = 1.7 \times 10^{-5}$ to $2.8 \times 10^{-4}$. A chemical model demonstrated that HONO deposition to the ground surface at night was 90-100% of the calculated unknown HONO source in the morning. These results suggest that dew water on the ground surface was controlling the temporal HONO distribution rather than straightforward $NO_2$–HONO conversion. This can strongly enhance the OH reactivity throughout morning time or other planted areas that provide large amount of ground surface based on the OH production rate calculation.

**Keywords:** HONO, ground surface, $NO_2$–HONO conversion, dew water, OH production

## 1 Introduction

Nitrous acid (HONO) is important in atmospheric chemistry as its photolysis (R1) is an important source of OH radicals. In the troposphere, OH radicals can initiate daytime photochemistry, not at least leading to the formation of ozone ($O_3$) and secondary organic aerosol (SOA).

$$HONO + h\nu \rightarrow OH + NO \tag{R1}$$

At present, the mechanisms of HONO formation have been and are still widely discussed. In the absence of light, heterogeneous reactions of $NO_2$ occur on wet surfaces (R2) and are considered to be an important source of HONO according to both laboratory studies and field observations (Acker et al., 2004).

$$2NO_2 + H_2O \rightarrow HONO + HNO_3 \tag{R2}$$

Finlayson-Pitts et al. (2003) proposed a mechanism (R3) involving the formation of the $NO_2$ dimer ($N_2O_4$) especially during nighttime. However, this pathway is not important in the real atmosphere (Gustafsson et al., 2008). The surface of soot (Ammann et al., 1998;Arens et al., 2001;Gerecke et al., 1998) or light activated soot (Aubin and Abbatt, 2007;Monge et al., 2010) contain functionalities attached to the large carbonaceous structures or individual condensed organic species, like phenol (R4) (Gutzwiller et al., 2002) and light-activated humic acids (Stemmler et al., 2006), which undergo electron transfer reactions with $NO_2$ yielding HONO (R5, where HA, $A^{red}$, and X are humic acid, activation of reductive centers and oxidants, respectively). This reaction is also postulated for aromatics in the aqueous phase, but only proceeds at a relevant rate at high pH levels (Ammann et al., 2005;Lahoutifard et al., 2002). Gustafsson et al. (2008) provide the evidence that formation of HONO proceeds by a bimolecular reaction of absorbed $NO_2$ and H (R6) on mineral dust, where H formed from the dissociation of chemisorbed water. However, Finlayson-Pitts (2009) indicated that this pathway is probably not transferable from laboratory to real atmosphere. In addition to the direct emission from the vehicle exhaust (Kurtenbach et al., 2001) and homogeneous gas phase reaction of NO with OH (R7) (Pagsberg et al., 1997), some other HONO formation mechanisms have been proposed e.g. homogeneous reaction of $NO_2$, $H_2O$, and $NH_3$ (R8) (Zhang and Tao, 2010); photolysis of nitric acid and nitrate ($HNO_3/NO_3^-$) (R9) (Ye et al., 2016;Zhou et al., 2011) and nitrite emission from soil (R10) (Su et al., 2011).

$$2NO_2\,(g) \leftrightarrow N_2O_4\,(g) \leftrightarrow N_2O_4\,(surface) \leftrightarrow HONO\,(surface) + HNO_3\,(surface) \tag{R3}$$

$$NO_2 + \{C-H\}_{red} \rightarrow HONO + \{C\}_{ox} \tag{R4}$$

$$HA \xrightarrow{h\nu} A^{red} + X;\ A^{red} + X \rightarrow A';\ A^{red} + NO_2 \rightarrow A'' + HONO \tag{R5}$$

$$NO_2\,(ads) + H\,(ads) \rightarrow HONO\,(ads) \rightarrow HONO\,(g) \tag{R6}$$

$$NO + OH \rightarrow HONO \tag{R7}$$

$$NO_2\,(g) + H_2O\,(g) + NH_3\,(g) \rightarrow HONO\,(g) + NH_4NO_3\,(s) \tag{R8}$$

$$HNO_3/NO_3^- + h\nu \rightarrow HONO/NO_2^- + O \tag{R9}$$

$NO_2^-$ (aq) + H+ (aq) → HONO (aq) (R10)

Several studies (Acker et al., 2004;He et al., 2006;Lammel and Perner, 1988;Lammel and Cape, 1996;Rubio et al., 2009;VandenBoer et al., 2013;VandenBoer et al., 2014) reported that deposited HONO on wet surfaces can be a source for observed daytime HONO. He et al. (2006) observed HONO released from a drying forest canopy and their lab studies showed that, on average, ~90% of $NO_2^-$ was emitted as HONO during dew evaporation. Rubio et al. (2009) found a positive correlation between formaldehyde and HONO in dew and the atmosphere.

The dominant loss of HONO is photolysis during daytime, which forms OH radicals (R1). An additional sink of HONO is the reaction with OH radical (R11). Due to the absence of solar radiation and the low OH concentration, the main loss process of HONO during nighttime is dry deposition, which can reach the balance with HONO production and vertical mixing to generate a steady state of HONO mixing ratio.

HONO + OH → $H_2O$ + $NO_2$ (R11)

Due to its significant atmospheric importance, HONO has been measured for many years with various techniques (Febo et al., 1993;Huang et al., 2002;Kanda and Taira, 1990;Platt et al., 1980;Schiller et al., 2001;Wang and Zhang, 2000). LOng Path Absorption Photometer (LOPAP) is a two channel in situ HONO measurement instrument, which detects HONO continuously by wet sampling and photometric detection. LOPAP is very selective without sampling artefact and chemical interferences (e.g. $NO_2$, NO, $O_3$, HCHO, $HNO_3$, $SO_2$ and PAN etc.). In addition, the detection limit of LOPAP can go down to 0.2 pptv (Kleffmann and Wiesen, 2008) by optimizing the parameters like (a) sample gas flow rate, (b) liquid flow rates, and (c) the length of the absorption tubing (Heland et al., 2001). LOPAP was validated and compared with the most established and reliable HONO instrument Differential Optical Absorption Spectroscopy (DOAS) Both were used in the field and in a large simulation chamber under various conditions resulting in excellent agreement (Heland et al., 2001;Kleffmann et al., 2006).

The Monitor for AeRosols and Gases in ambient Air (MARGA) is a commercial instrument combining a Steam-Jet Aerosol Collector (SJAC) and a Wet Rotating Denuder (WRD), which can quantify the inorganic water-soluble PM ions ($Cl^-$, $NO_3^-$, $SO_4^{2-}$, $NH_4^+$, $Na^+$, $K^+$, $Mg^{2+}$, $Ca^{2+}$) and corresponding trace gases (HCl, HONO, $HNO_3$, $SO_2$, $NH_3$). In recent years, MARGA measurements were performed worldwide, which has been summarized by Stieger et al. (2018). Within the cited study HONO concentrations measured by a MARGA system and an off-line batch denuder without an inlet system were compared. Although the slope between both instruments was 1.10 with slightly higher MARGA concentrations in average, both instruments biased equally in the measured concentrations resulting in a high scattering with a coefficient of determination of $R^2 = 0.41$. The probable reason was the off-line analysis of the batch denuder sample as the resulting longer interaction of gas and liquid phase during the transport led to further heterogenous reactions. As both instruments

are based on the same sampling technique, the present study could be a good starting point for an inter-comparison between MARGA and LOPAP for HONO measurements to find possible reasons in the denuder deviations.

In this study, we present parallel measurements of HONO using LOPAP and MARGA in Melpitz, Germany, over two weeks in 2018. For further investigations, dew water was collected and analyzed from May 8th to 14th 2019 using two glass samplers. In addition, other water-soluble compounds, such as gaseous $HNO_3$, $NH_3$ and particulate $NO_3^-$, $SO_4^{2-}$, $NH_4^+$, $Na^+$, $K^+$, $Mg^{2+}$, $Ca^{2+}$, trace gases ($NO_x$, $SO_2$ and $O_3$) and meteorological parameters were also measured simultaneously. Our observations provide a direct inter-comparison between LOPAP and MARGA for HONO field measurement, additional insights into HONO chemical formation processes and examine the relative importance of dew as a sink and source of HONO.

## 2 Experimental

### 2.1 Site description

Measurements were performed at the research station of the Leibniz Institute for Tropospheric Research (TROPOS) in Melpitz (12°56′E, 51°32′N). This rural field site is situated on a meadow and surrounded by flat grass land, agricultural areas and forests. The Melpitz site mainly can be influenced by two different wind direction: west wind origin from the marine crossing a large area of Western Europe and the city of Leipzig (41 km NE), and east wind crossing Eastern Europe (Spindler et al., 2004).

### 2.2 MARGA instrument

The MARGA (1S ADI 2080, The Netherlands) used in this study has already been described in Stieger et al. (2018). Hence, only short information is provided here. An inlet flow of 1 $m^3$ $hr^{-1}$ was drawn into the sampling box after passing through an inside Teflon-coated $PM_{10}$ inlet (URG, Chapel Hill, 3.5 m). Within the sample box, the sampled air laminarly passed a WRD, in which water-soluble gases diffuse into a 10 mg $l^{-1}$ hydrogen peroxide ($H_2O_2$) solution at pH = 5.7. Particles can reach the SJAC because of their smaller diffusion velocities. Within the SJAC, the particles grow into droplets under supersaturated water vapor conditions and were collected by a cyclone. The gas and particle samples are both collected over the course of one hour. Then, the aqueous samples of the WRD (gas phase) and the SJAC (particle phase) were successively injected into two ion-chromatographs (IC) with conductivity detectors (Metrohm, Switzerland) by two syringe pumps for analyzing the anions and cations. The volume of the injection loops for the anions and cations were 250 μl and 500 μl, respectively. The Metrosep A Supp 10 (75/4.0) column and Metrosep C4 (100/4.0) column were used to separate anions and cations, respectively. Lithium bromide was used as the internal standard for both gas- and particle-phase samples added during the sample injection to the IC.

The detection limits and the blanks for the MARGA system were performed before the intercomparison campaign in 2018. The detection limit of HONO was determined as 10 pptv. The blanks were analyzed when the system was set up in the field to consider potential contaminations. For blank measurements, the MARGA blank measurement mode was used that has a duration of six hours. Within the first 4 hours, the MARGA air pump was off and the denuder and SJAC liquids were analyzed. The first- and second-hour samples are discarded as they still include residual concentrations. The evaluation of the blank concentrations was performed for the third- and fourth-hour samples. No discernable peaks above the instrument detection limits were identified in both the gas and particle phase channels.

The precision for HONO quantification is below 4 % indicating a good repeatability. To test the robustness of the ion chromatography within the MARGA, standard solutions with defined $NO_2^-$ concentrations of 70, 120 and 150 μg $L^{-1}$ were injected in the IC system. The correlation between both the predefined concentrations within the standard solutions and the measured concentrations by the MARGA IC resulted in a slope of 1.13 ($R^2 = 0.99$). This value indicates slightly lower measured $NO_2^-$ concentrations, which might be also a result of nonstable $NO_2^-$ in freshly made liquid standard solutions.

**2.3 LOPAP instrument**

The LOPAP (QUMA, Germany) employed in this work was described in previous studies (Bernard et al., 2016;Heland et al., 2001). Only brief description is given here. The LOPAP instrument consists of two sections: a sampling unit and a detection unit. The ambient air was sampled in the sampling unit, which composed of two glass coils in series where the first coil (channel 1) accounted for HONO with interferences and the second coil (channel 2) sampled only interferences assuming that more than 99 % of HONO was absorbed into acidic stripping solution (pH=0) to form diazonium salt in channel 1. This salt reacts with a 0.8mM n-(1-naphthyl)ethylenediamine-dihydrochloride solution to produce final azo dye, which is photometrically detected by long path absorption in a special Teflon tubing (Heland et al., 2001;Kleffmann et al., 2006). During our field campaign in Melpitz, both the acidic stripping solution and 0.8mM n-(1-naphthyl)ethylenediamine-dihydrochloride solution were kept in the dark and were not changed during the whole campaign period. The temperature of the stripping coil was kept constant at 25 ℃ by a thermostat. Automatic zero air (Air liquid, Alphagaz 2, 99.9999%) measurements were performed for 30 min per 12 h measurements to correct for zero drifts. In addition, calibrations using $NO_2^-$ standard solution (Heland et al., 2001) were applied in the beginning (April 17[th]), middle (April 20[th], 24[th], 25[th]) and end (April 29[th]) of the campaign to derive the HONO mixing ratio. The detection limit of LOPAP was approximately 1-2 pptv with a response time of 5 min. The error of HONO mixing ratio was estimated based on these detection limits and a relative error of 10%. The relative error is calculated by error propagation of all systematic errors, i.e. uncertainties in the gas flow ca. 2%, the liquid flow ca. 2 %, the error in the nitrite concentration during calibration 1 % and

errors for the used pipettes/flasks (two times of the specified errors of all volumetric glass ware since all glass ware was not used exactly at 20 ℃ like recommended by the manufacturer).

To investigate the possible sampling inlet and denuder artefacts of the MARGA, two different positions were selected for LOPAP during the measurement period (explained in SI): (M1) sampling unit of LOPAP was connected to the MARGA inlet in the back of the 2 m sampling tube and the $PM_{10}$ inlet of MARGA as shown in Fig. S1a (April 18[th], 2018 13:00 UTC –April 20[th], 2018 08:00 UTC); (M2) the sampling unit of LOPAP was settled in the same level as the sampling head of MARGA (Fig. S1b) (April 20[th], 2018 15:00 UTC –April 29[th], 2018 07:00 UTC).

## 2.4 Dew water collection and analysis

To evaluate the HONO emission from the dew water in the morning, the dew water was collected one year later after the HONO comparison campaign and was analyzed on May 8[th], 11[th], 13[th] and 14[th] 2019. Similar conditions (grass height, dew formation and day length) were observed to improve the evaluation. For dew sampling, a glass sampler was used (as shown in Fig. S2). Two 1.5 $m^2$ glass plates (Plate 1 and Plate 2) were placed 40 cm above the ground with a tilt angle of approximately 10 °. A gutter was installed at the lower end of each plates to collect the running down water. The water is trapped in 500 ml bottles. The dew samplers were prepared each evening before a likely dew event occurred (low dew point difference, clear sky and low winds). Each plate was rinsed with at least 2 L ultrapure water. A squeegee removed the excess water. Afterwards, the plates were cleaned with ethanol and were again rinsed by 2 L ultrapure water. The plate was splashed with ultrapure water and squeegeed six times and the gutter was cleaned. The sample of the sixth splash was collected as blank (~ 50 mL).

The dew water normally was collected from 18:00 to 5:00 (UTC). In the morning, the excess dew on the plate was squeegeed. To achieve the volume of dew ($V_{dew}$), the bottles were weighted before and after sampling by a balance. The pH was measured by a pH meter (mod. Lab 850, Schott Instruments) on a subsample of the total volume. After sampling, the aqueous solutions were filtered and stored in a fridge (~6 ℃). Within six hours, the HONO analyses of the dew and blank samples were performed by double-injection in the MARGA in the manual measurement mode as HONO may volatilize between sampling and analysis. For the other ions ($Cl^-$, $NO_3^-$, $SO_4^{2-}$, Oxalate, $Br^-$, $F^-$, Formate, MSA , $PO_4^{3-}$, $Na^+$, $NH_4^+$, $K^+$ and $Mg^{2+}$, $Ca^{2+}$), the samples were analyzed with laboratory ion chromatogram systems (mod. ICS-3000, Dionex, USA). Blanks from water, the filter, the syringes and bottles were subtracted.

## 2.5 Aerosol measurements

The particle size distributions were measured in the size range from 5 nm to 10 μm employing by a Dual Mobility Particle Size Spectrometer (TROPOS-type D-MPSS) (Birmili

et al., 1999) and an Aerodynamic Particle Size Spectrometer (APSS model 3321, TSI Inc., Shoreview, MN, USA). For the particle number size distribution measurements, the aerosol is sampled through a low flow PM10 inlet and dried in an automatic diffusion dryer (Tuch et al., 2009). The measurements and quality assurance are done following the recommendations given in Wiedensohler et al. (2012) and Wiedensohler et al. (2018). The MPSS derived particle number size distribution was inverted by the algorithm described in Pfeifer et al. (2014), following the bipolar charge distribution of Wiedensohler (1988).

## 2.6 Other measurements

Trace gases of $NO$-$NO_2$-$NO_x$, $SO_2$ and $O_3$ were measured by $NO_x$ analyzer (Thermo Model 42i-TL, Waltham, Massachusetts, USA), $SO_2$ analyzer APSA-360A and $O_3$ analyzer APOA 350 E (both Horiba, Kyoto, Japan) with a time resolution of 1 min. It should be noted that $NO_2$ was converted to NO within the $NO_x$ analyzer by a blue light converter BLC2 (Meteorologie Consult GmbH, Königstein, Germany). The provider for replacement of the Mo-Converter in the 42i-TL analyzer is MLU Messtechnik GmbH, Essen Germany. Meteorological parameters like temperature (T), precipitation, relative humidity (RH) as well as wind velocity and direction were measured by PT1000, a rain gauge (R.M. Young Company, U.S.A.), the CS215 sensor (SensirionAG, Switzerland) and a WindSonic by Gill Instruments (UK), respectively. Global radiation and barometric pressure were recorded by a net radiometer CNR1 (Kipp&Zonen, The Netherlands) and a digital barometer (Vaisala, Germany), respectively.

## 2.7 Calculation of photolysis rate

Off-line NCAR Tropospheric Ultraviolet and Visible (TUV) transfer model (https://www2.acom.ucar.edu/modeling/tropospheric-ultraviolet-and-visible-tuv-radiation-model) was used to estimate the photolysis rate of HONO ($J_{HONO}$), $NO_2$ ($J_{NO2}$) and production rate of $O^1D$ ($J_{O1D}$) at the Melpitz station scaled by the measured global radiation. Aerosol optical depth (AOD), total vertical ozone column, total $NO_2$ column, total cloud optical depth and surface reflectivity (Albedo) were taken from the NASA webpage for the period of measurement (https://neo.sci.gsfc.nasa.gov/blog/).

## 3 Results

### 3.1 Inter-comparison of LOPAP and MARGA

The hourly HONO mixing ratio obtained from MARGA with the 30 seconds and hourly averaged HONO mixing ratios from LOPAP are shown in Fig. 1a and 1b, respectively. It indicates that the MARGA values were higher than the values of LOPAP. In addition, the comparison between both instruments in Fig. 1a shows a delay of the MARGA concentrations after reaching the maximum concentrations in the morning. This pattern was also observed in

previous studies of Volten et al. (2012) and Dammers et al. (2017), who compared miniDOAS instruments with wet denuder systems. Compared to fast responses of the miniDOAS, the denuder-based instruments showed offsets and delays because of inlet memory artefacts by particles or water. Both groups also suggested transport effects of the liquid samples from the sampling to the analysis unit resulting in delays and slow responses.

The comparisons of the MARGA and LOPAP HONO measurements for period M1 and period M2 in Fig. 1c result in slopes of 1.71 and 2.17 using error weighted Deming regression, respectively. This result is consistent with the former intercomparison of both instrument types in the Chinese field campaign (Lu et al., 2010;Xu et al., 2019) where the HONO mixing ratio measured with the wet-denuder-ion-chromatography (WD/IC) instrument was affected by a factor of three on average. Within the present work, we evaluated the relative importance of denuder artefact with the inlet artefact. The heterogeneous reactions of $NO_2$ with $H_2O$ as well as $NO_2$ with $SO_2$ in water described by Spindler et al. (2003) or VOCs with $NO_2$ could explain the artefacts in the denuder solution (Kleffmann and Wiesen, 2008), which could account for ca. 71% (M1, where both LOPAP and MARGA used the common MARGA inlet) of these ca. 117% of overestimated HONO measurement from MARGA. Additional artefacts such as heterogeneous formation of HONO due to the long MARGA inlet system should be responsible for another ca. 46% (the difference between slopes M2 and M1). Hence, the results show that the use of massive sampling inlets, even if they are coated by Teflon, should be avoided for any in-situ HONO instrument. As a result, we chose the LOPAP-measured HONO in the following sections because of its high accuracy.

### 3.2 General results

Fig. 2 and Fig. 3 show an overview of the measured HONO, NO, $NO_2$, $O_3$, meteorological parameters, water-soluble ions in $PM_{10}$ ($NO_3^-$, $SO_4^{2-}$, $NH_4^+$, $Na^+$, $K^+$, $Mg^{2+}$, $Ca^{2+}$) and their corresponding trace gases (HONO, $HNO_3$, $SO_2$, $NH_3$) in the present study. The daytime (D, 04:00-18:00, UTC) and nighttime (N, 18:00-04:00) averages are also provided in Table 1. During the two weeks measurement, the prevailing winds were from the southwest and northwest sectors, indicating a possible influence of city emission from Leipzig, Germany, on the site. The strong wind (maximum 13 m s$^{-1}$) led to low concentration of water-soluble ions in $PM_{10}$ ($NO_3^-$, $SO_4^{2-}$, $NH_4^+$) and their corresponding trace gases ($HNO_3$, $SO_2$, $NH_3$) during the period April 24$^{th}$ to 29$^{th}$, 2018. The air temperature ranged from 5 ℃ to 27 ℃ and the RH showed a clear variation pattern with higher levels during the night and lower levels during daytime. In addition, low mixing ratio of NO and $NO_2$ with a diurnal average of 0.9±1.2 ppbv and 3.7±2.2 ppbv, respectively, were recorded. These observations highlight the nature of our measurement site as a typical background environment. The HONO concentration from the LOPAP measurements varied from 30 pptv to 1582 pptv and showed diurnal variations (with average values of 162±96 pptv and 254±114 pptv during daytime and nighttime, respectively).

Größ et al. (2018) reported the linear function of the global radiation flux vs. OH radical concentration for the campaign EUCAARI 2008 at Melpitz.

$$[OH]=A*Rad \qquad\qquad\qquad\qquad\qquad (Eq.\ 1)$$

with Rad being global solar irradiance in $W\,m^{-2}$ and [OH] is the hydroxyl radical concentration. The proportionality parameter A is $6110\ m^2\,W^{-1}\,cm^{-3}$. On the basis of such a correlation, we derived the OH concentration during the period of this field measurement, with an average of $(2.8\pm0.7)\times10^6$ during daytime.

### 3.3 Diurnal variation of HONO, particles and trace gas species

The diurnal profiles of HONO and related supporting parameters are shown in Fig. 4 for the whole period except for two sets of observations: (1) no HONO peak in the morning of April 23$^{rd}$ and (2) HONO peak observed at 0:00-2:00 (UTC) of April 25$^{th}$ (Fig. 5). Overall, the HONO increased fast after the sunrise and peaked at 7:00 (UTC), which then dropped rapidly and reached a minimum at around 10:00 (UTC) and kept until 17:00 (UTC). Such daytime pattern was also found in Spain, for a site surround by forests and sandy soils (Sörgel et al., 2011). Sörgel et al. (2011) explained this by local emissions, which are trapped in the stable boundary layer before its breakup of the inversion in the morning based on a similar diurnal cycle for NO and $NO_2$, which is different with this work. In this work, the $NO_2$ mixing ratio decreased from the midnight until noon and NO peaked at 5:00 (UTC) then kept low concentration (<1 ppbv) for 18 hours of one day. However, three hypotheses could be expected to explain this HONO morning peak: hypothesis (a) of HONO morning peak might possibly be caused by the photolysis of particle-phase $HNO_3/NO_3^-$ (Ye et al., 2016;Zhou et al., 2003;Zhou et al., 2011), since as shown in Fig. 4a, 4e and 4f, the early morning variation trend of HONO during daytime was similar to the one of $NH_3$ in the gas phase as well as $NO_3^-$ and $NH_4^+$ in $PM_{10}$. Hypothesis (b), as reported by Stemmler et al. (2006), the photosensitized $NO_2$ on humic acid could act as a source of HONO during the daytime. For hypothesis (c), this morning peak of HONO has been reported for Melpitz (April 4$^{th}$-14$^{th}$, 2008) by Acker et al. (2004), who expected that the storage of HONO on wet surfaces can be a source for observed daytime HONO. Exactly, it was observed that dew was formed overnight during our campaign in Melpitz. Gaseous HONO could be deposited in these droplets. Due to evaporation after sunrise, HONO would be reemitted in the atmosphere and lead to a HONO morning peak. These hypotheses will be further discussed in Section 4.

As shown in Fig. 4a and 4b, the HONO and $NO_2$ concentrations started to increase coincidentally at 16:00 (UTC) when the sunshine was weak. This could be explained by the variation of the vertical mixing increasing the level of all near ground emitted of formed species or by the heterogeneous conversion of $NO_2$ to HONO during nighttime and will be discussed in Section 4. The HONO mixing ratio then decreased from 21:00 (UTC) to around 100 pptv even though the $NO_2$ concentration kept constant around 5-6 ppbv. This decrease during nighttime indicates the HONO loss process (dry and wet deposition, trapped in the

boundary layer or dew etc.) surpassing the HONO formation from the $NO_2$-to-HONO conversion. The diurnal cycle of $O_3$ reflects the balance between the photochemical formation of $O_3$ (e.g. $NO_2$ + hv) and $O_3$ consumption (e.g. ozonolysis of terpenes).

### 3.4 HONO in the dew water

Dew water formation on canopy surfaces could be an efficient removal pathway of water soluble pollutants. High solubility of HONO makes dew water an efficient sink and a stable reservoir for atmospheric HONO. Actually, a lot of dew water has been observed on the grass around the Melpitz station during the sampling period of April 19th to 29th 2018. Hence, to investigate the dissolved HONO in the dew water of Melpitz station, the dew water was collected and analyzed from May 8th to 14th 2019 at the same season like the HONO measurements. Many ions e.g. $NO_2^-$, $Cl^-$, $NO_3^-$, $SO_4^{2-}$, Oxalate, $Br^-$, $F^-$, Formate, MSA, $PO_4^{3-}$, $Na^+$, $NH_4^+$, $K^+$ and $Mg^{2+}$, $Ca^{2+}$ were analyzed using MARGA and laboratory IC, but our discussion only focuses on $NO_2^-$. The sample parameters (time, pH etc.) and $NO_2^-$ concentration in the sample ($\mu$g $L^{-1}$) are shown in Table 2 from two glass plates (plate 1 and plate 2). The final dew water $NO_2^-$ was calculated by subtracting the blank $NO_2^-$ from the raw data of dew water analysis in MARGA. The pH of dew water in Melpitz ranged from 6.30 to 7.00. It should be noted that the dew water was frozen until 1 hour after sunrise on May 8th, 13th and 14th 2019 but not on May 11th 2019. At this day, a third sample was collected sampled from 3:30 to 5:20 (UTC) after collecting the first sample (18:00-3:20 UTC). The $NO_2^-$ concentration per $m^2$ of the sampler surface ($F_{NO2^-}$) was calculated from the following equation:

$$F_{NO2^-} = \frac{[NO_2^-] \times V_{dew}}{S \times 1000} \tag{Eq. 2}$$

Where $[NO_2^-]$ is the sample concentration in $\mu$g $L^{-1}$, $V_{dew}$ is the sample volume in ml, S is the surface area of the glass sampler as 1.5 $m^2$. As shown in Table 2, higher $F_{NO2^-}$ was obtained on May 11th where dew water was not frozen. On other days (May 8th, May 13th and May 14th) frozen dew water was observed, which likely inhibited HONO to dissolve. Hence, these frozen samples were not considered in this paper. On May 11th, the final $F_{NO2}^-$ could be obtained by averaging $F_{NO2}^-$ of the sum (9.43 $\mu$g $m^{-2}$) of the first and third sample with the second sample (6.40 $\mu$g $m^{-2}$) on 11th May resulting in mean 7.91±2.14 $\mu$g $m^{-2}$. This value will be used for the following calculation and discussion.

### 4 Discussion

### 4.1 Contribution of vehicle emissions

Because Melpitz site is close to a main national road from Leipzig to Torgau (Germany) that is within the main southwest wind direction, the contribution of vehicle emissions to the

measured HONO mixing ratio should be evaluated. Generally, the HONO/$NO_x$ ratio is usually
chosen to derive the emission factor of HONO in the freshly emitted plumes (Kurtenbach et
al., 2001). As illustrated in Fig. S3, $NO_x$ concentrations were normally lower than 15 ppbv
and NO/$NO_x$ ratios were ~0.4 in this campaign, suggesting the detected air is a mixture of
fresh and aged air during the measurement period. Therefore, a substantial part of HONO is
secondary. Additionally, following the criteria of Li et al. (2018), the bad correlation between
HONO and $NO_x$ ($R^2 \approx 0.35$) suggests that the direct HONO emission from the vehicle emitted
plumes were less important in this work.

## 4.2 Nighttime HONO

The nighttime HONO is different to some reported literatures (Huang et al., 2017;Li et al.,
2012;Wang et al., 2017;Zhou et al., 2007). HONO increased after sunset to a maximum at
21:00 (UTC) and decreased until sunrise.

### 4.2.1 Formation through heterogeneous conversion of $NO_2$

The ratio of HONO/$NO_2$ is generally used as an index to estimate the efficiency of
heterogeneous $NO_2$-HONO conversion because it is less influenced by transport processes
than individual concentrations. However, the ratio might be influenced when a large fraction
of HONO is emitted from the traffic but this is expected to be less important as shown in
section 4.1. Then in this work, a low emission factor of 0.3% was used to correct the directly
HONO emission from vehicles (HONO$_{corr}$) (Kurtenbach et al., 2001) . Six conditions as listed
in Table 3 are selected to calculate the $NO_2$-HONO frequency following the criteria of Li et al.
(2018):

(a) only the nighttime data in the absence of sunlight (i.e., 17:30-06:00 UTC) are used;

(b) both HONO$_{corr}$ and HONO$_{corr}$ / $NO_2$ ratios increased steadily during the target case;

(c) the meteorological conditions, especially surface winds, should be stable.

Fig. S4 presents an example of the heterogeneous HONO formation occurring on April 28[th],
2018. In this case, the HONO mixing ratios increased rapidly after sunset from 100 pptv to
600 pptv. Together with the HONO mixing ratio, the HONO$_{corr}$/$NO_2$ ratio increased almost
linearly between 18:00 to 19:50 UTC. The slope fitted by the least squares regression for
HONO$_{corr}$/$NO_2$ ratios against time can be taken as the conversion frequency of $NO_2$-to-HONO
($k_{het}$).

The ratio of HONO$_{corr}$ / $NO_2$ ranged from 0.055 to 0.161 with mean value of 0.110±0.041
(Table 3) using the data during early nighttime (17:30-00:00 UTC) in the Melpitz campaign.
This mean values are within the wide range of reported values of 0.008-0.13 in the fresh air
masses from the most sampling sites (Alicke et al., 2002;Alicke et al., 2003;Sörgel et al.,
2011;Su et al., 2008;VandenBoer et al., 2013;Wang et al., 2017;Zhou et al., 2007) except for
the study of Yu et al. (2009), who got a high value of 0.3. To our best knowledge, the present
work presents also a high $NO_2$-to-HONO conversion frequency $k_{het}$ of 0.027±0.017 h[-1]

compared with most of the previous studies at urban sites, such as, Alicke et al. (2002) in Milan (0.012 h$^{-1}$), Wang et al. (2017) in Beijing (0.008 h$^{-1}$) and Acker and Möller (2007) in Rome (0.01 h$^{-1}$). However, our value is additionally comparable to Li et al. (2012) with 0.024±0.015 h$^{-1}$, Alicke et al. (2003) with 0.018±0.009 h$^{-1}$ and Acker and Möller (2007) with 0.027±0.012 h$^{-1}$, who also conducted rural measurements in the Pearl River Delta (PRD) area in Southern China, Pabstthum in Germany, and Melpitz, respectively, surrounded by farmland (grasses, trees, small forests). The higher value may suggest that a more efficient heterogeneous conversion from NO$_2$ to HONO is present in rural sites than in urban sites.

### 4.2.2 Relative importance of particle and ground surface in nocturnal HONO production

The particle surface density $S_a$ was calculated as (0.4-9.9) ×10$^{-4}$ m$^2$ m$^{-3}$ from the particle size distribution (Fig. S5a) ranged from 5 nm to 10 μm of APSS and D-MPSS data by assuming the particle are in spherical shape for the whole day period of April 19$^{th}$-29$^{th}$ 2018. Due to the high RH (RH ~100% during nighttime in Fig. S5b), the particle surface density $S_a$ would be strongly enhanced (one magnitude) by the RH correction to be (0.5-1.9) ×10$^{-3}$ m$^2$ m$^{-3}$ with a hygroscopic factor $f$(RH) following the method of Li et al. (2012) and Liu et al. (2008):

$$f(RH)=1+a\times(RH/100)^b \tag{Eq. 3}$$

where the empirical factors a and b were set to 2.06 and 3.6, respectively.

The formation of HONO through heterogeneous NO$_2$ conversion on particle surfaces ($S_a$) can be approximated following the recommendations in Li et al. (2010) by considering 100% HONO yield on the particle surface (NO$_2$+Org/soot/etc):

$$k_{het} = \frac{1}{4}\gamma_{NO2\rightarrow HONO\_a} \times \upsilon_{NO2} \times \frac{S_a}{V} \tag{Eq. 4}$$

where $\upsilon_{NO2}$ is the mean molecular velocity of NO$_2$ (370 m s$^{-1}$) (Ammann et al., 1998); $S_a$/V is the particle surface to volume ratio (m$^{-1}$) representing the surfaces available for heterogeneous reaction, and $\gamma_{NO2\rightarrow HONO\_a}$ is the uptake coefficient of NO$_2$ at the particle surface. Assuming the entire HONO formation was taking place on the particle surface, the calculated $\gamma_{NO2\rightarrow HONO\_a}$ from the Eq. 4 varied from 2.8×10$^{-5}$ to 3.8×10$^{-4}$ with a mean value of (1.7±1.0) ×10$^{-4}$. This number is 2-3 orders of magnitude higher than typical uptake coefficients determined in the lab for the uptake of NO$_2$ in the dark on different substrates, e.g. Teflon/glass/NaCl/TiO$_2$/soot/Phenol etc: 10$^{-6}$ to <10$^{-8}$ (Ammann et al., 1998;Gutzwiller et al., 2002;Kleffmann et al., 1998;Kurtenbach et al., 2001). Thus, this theoretical uptake coefficient clearly shows that formation on particles is not important. In addition, the weak correlations between HONO$_{corr}$ (R$^2$=0.566), HONO$_{corr}$/ NO$_2$ (R$^2$=0.208) and $S_a$ (Fig. S6) confirm that the HONO formed on particle surfaces could be unimportant as previously reported (Kalberer et al., 1999;Sörgel et al., 2011;Wong et al., 2011).

As illustrated above, the heterogeneous NO$_2$ conversion on ground surfaces (including surfaces such as plants, building, soils etc.) contributes mainly to nighttime formation of

HONO, which can be approximated by Eq. 5 following the method in literatures (Kurtenbach et al., 2001;Li et al., 2010;VandenBoer et al., 2013;VandenBoer et al., 2014) and also been applied by Zhang et al. (2016) by considering a 50% HONO yield from R2:

$$k_{het} = \frac{1}{8}\gamma_{NO2\rightarrow HONO\_g} \times \upsilon_{NO2} \times \frac{S_g}{V} \tag{Eq. 5}$$

where $\gamma_{NO2\rightarrow HONO\_g}$ is the uptake coefficient of $NO_2$ at the ground surface, $S_g/V$ represents the ground surface to volume ratio. As described by Zhang et al. (2016), the LAI ($m^2/m^2$) was used to estimate the surface to volume ratio for the vegetation-covered areas, following the method in Sarwar et al. (2008):

$$\frac{S_g}{V} = \frac{2 \times LAI}{H} \tag{Eq. 6}$$

Where $H$ is the mixing layer height, which was calculated from the backward trajectory analysis based on GDAS data under dynamic conditions (Fig. S7). The mixing layer height ranged between 20 m and 300 m from 17:00 until around 00:00 UTC in April 2018 (Fig. S7). The LAI value is multiplied by a factor of 2 to take the areas on both sides of the leaves into account. In Wohlfahrt et al. (2001), the LAI for meadows with different grass heights are given. Regarding the grass height of ~30 cm in April 2018, we used a factor of 6 in present study. If the entire HONO formation was taking place on the ground surface, the calculated $\gamma_{NO2\rightarrow HONO\_g}$ varied from $2.4 \times 10^{-7}$ to $3.5 \times 10^{-6}$ with a mean value of $2.3 \pm 1.9 \times 10^{-6}$. This value agrees well with the reported range of $\gamma_{NO2\rightarrow HONO}$ from $10^{-6}$ to $10^{-5}$ on the ground surface based on the laboratory studies (Donaldson et al., 2014;VandenBoer et al., 2015) and field campaign in Colorado, USA (VandenBoer et al., 2013) during the night time. As the S/V ratio of particles is typically orders of magnitude lower than for ground surfaces, it is suggested that the heterogeneous reactions of $NO_2$ on the ground surface may play a dominant role for the nighttime HONO formation.

In addition, the relationship of $NO_2$-HONO conversion frequency ($k_{het}$ presented in Table 3) with the inverse of wind speed is illustrated in Fig. S8a. As indicated in Fig. S8a, wind speed was predominantly less than 3 m s$^{-1}$ during the field campaign period in Melpitz. High conversion frequency of $NO_2$-to-HONO mostly happened when wind speed was less than 1 m s$^{-1}$, which confirms that HONO formation mainly takes place on the ground. However, one point (in blue in Fig. S8a) showed highest $NO_2$-HONO conversion frequency ($k_{het}$) when wind speed was ca. 4 m s$^{-1}$ according to the second set of observation mentioned in section 3.3 and Fig. 5. The likely reason for the temporary HONO peak is the dew droplet evaporation after increasing wind speed.

**4.2.3 HONO deposition on the ground surface**

As illustrated in Fig. 4a and S4, between midnight and sunrise (19:00-4:00 UTC), the deposition of HONO becomes increasingly important as the absolute amount of HONO decreased. Assuming a constant conversion frequency of $NO_2$-to-HONO, $k_{het}$, the HONO

deposition rate ($L_{HONO}$) can be roughly estimated by:

$$L_{HONO=} \frac{d[HONO]}{dt} + k_{het} \times [NO_2] \qquad \text{(Eq. 7)}$$

The strength of the HONO sink during night is in average $0.16\pm0.12$ ppbv h$^{-1}$ and ranged from 0.04 to 0.45 ppbv h$^{-1}$. This value is similar with reported ones in the literature (He et al., 2006).

The relationship of [HONO]/[NO$_2$] with RH during nighttime (18:00-04:00) is illustrated in Fig. S8a. A positive trend of [HONO]/[NO$_2$] ratio along the RH was found when RH was less than 70%. However, [HONO]/[NO$_2$] performs a negative trend with RH for values over 70%. The same phenomenon was also observed by Yu et al. (2009) in Kathmandu and Li et al. (2012) in PRD region, China. This finding can be associated with larger amounts of water on various ground surfaces (plants and grasses) when ambient humidity approached saturation, leading to an efficient uptake of HONO.

Assuming all the extra HONO were removed through deposition on the ground surface, the change of HONO in the time interval of 22:00-04:00 (UTC) is parameterized using a combination of Eq. 7 and the following equation:

$$L_{HONO} = \frac{1}{4}\gamma_{HONO,ground} \times [HONO] \times \frac{\upsilon_{HONO}}{H} \qquad \text{(Eq. 8)}$$

Where $\gamma_{HONO,ground}$ is the HONO uptake coefficient on the ground surface, $\upsilon_{HONO}$ is the mean molecular velocity of HONO with $3.67\times10^4$ cm s$^{-1}$, $H$ is the mixing layer height calculated from the backward trajectory analysis ranging between 20 m and 150 m with an average of ca. 55 m from 22:00 until 04:00 UTC in April 2018. This approach yielded to a $\gamma_{HONO,ground}$ uptake coefficient in the range of $1.7\times10^{-5}$ to $2.8\times10^{-4}$ with an average of $(1.0\pm0.4)\times10^{-4}$, which is similar to data found in Boulder, Colorado, ranging from $2\times10^{-5}$ to $2\times10^{-4}$ (VandenBoer et al., 2013).

As observed by several studies (He et al., 2006;Rubio et al., 2009;Wentworth et al., 2016), the effective Henry's law solubility of HONO is highly pH-dependent (from borderline soluble at pH = 3 to highly soluble at pH $\geq$ 6), as would be expected for a weak acid. The pH of collected dew water during nighttime in May 2019 was 6.3-7.0 (Table 2), where the effective Henry's law solubility of HONO would be high. The amount of HONO in this dew water was quantified using MARGA and ranged between 42 and 165 μg L$^{-1}$, which is higher than NO$_2^-$ in Santiago's dew waters (Rubio et al., 2009). This could strongly support the obtained HONO uptake coefficient on the ground surface. These field-derived surface parameters of nighttime HONO production from NO$_2$ and surface deposition of HONO are valuable to the model evaluation. However, it should be noted that the measured pH of collected dew from the glass plate might differ compared to the pH of dew found on soil or vegetated surfaces. The chemical nature of the material, with which the water is in contact, can influence the effective pH.

A simple resistance model based on the concept of aerodynamic transport, molecular diffusion

and uptake at the surface (presented in SI) as proposed by Huff and Abbatt (2002) was used to evaluate the factor(s) controlling the potential applicability of the γ-coefficients calculated here for the uptake of $NO_2$ and deposition of HONO. As shown in Fig. S9, the deposition loss of HONO is potentially limited by a combination of aerodynamic transport, molecular diffusion and reaction processes. However, the HONO uptake will be transport-limited if the real uptake coefficients are $\geq 2.8 \times 10^{-4}$ and wind speed was less than 0.5 m s$^{-1}$. In addition, molecular diffusion could play an important role for HONO uptake on the surface. Regarding the uptake of $NO_2$ on the ground surface, the range of $NO_2$ uptake coefficients as $2.4 \times 10^{-7}$ to $3.5 \times 10^{-6}$ obtained in the present work indicates limitation only by the reactive uptake process. The consistency between our findings and the values of these parameters in models (Wong et al., 2011;Zhang et al., 2016) suggests that the broad scale applicability of these field-derived terms for surface conversion of $NO_2$ should therefore be possible. However, those value of γ found for HONO ($\gamma_{HONO, ground}=1.7 \times 10^{-5}$ to $2.8 \times 10^{-4}$) require further exploration from various field environments and controlled lab studies.

### 4.3 Daytime HONO

HONO concentrations started to increase after sunrise and peaked at 7:00 (UTC) (Fig. 4), during that time it also underwent photolysis, eventually reaching a steady state between 10:30–16:30 (UTC). Throughout the day, HONO was observed to reach an averaged minimum mixing ratio of $98 \pm 15$ pptv. Since NO and $NO_2$ have not the same diurnal cycle as HONO (Fig. 4), the R2 and R7 are not expected to be responsible for this HONO morning peak, but could contribute to the daytime HONO for the period of 10:30-16:30 (UTC).

#### 4.3.1 Photostationary state in the gas phase

The measured diurnal daytime HONO could be compared to model results by assuming an instantaneous photo-equilibrium between the gas-phase formation (R7) and gas-phase loss processes (R1 and R11), which is described by the following expression (Kleffmann et al., 2005):

$$[HONO]_{pss} = \frac{k_7[OH][NO]}{J_{HONO}+k_{11}[OH]} \tag{Eq. 9}$$

OH concentration was estimated from linear function of the global radiation flux vs. OH radical concentration as described in the previous section and shown in Fig. 6, $J_{HONO}$ was calculated using TUV model as described in section 2.6. The rate constants of NO+OH ($k_7$) and HONO+OH ($k_{11}$) used are $7.4 \times 10^{-12}$ cm$^3$ molecule$^{-1}$ s$^{-1}$ (Burkholder et al., 2015) and $6.0 \times 10^{-12}$ cm$^3$ molecule$^{-1}$ s$^{-1}$ (Atkinson et al., 2004), respectively. As a result, shown in Fig. 6, the [HONO]$_{pss}$ (PSS, violet curve) could not explain the sudden HONO increase after sunrise but indicates a HONO peak around 4:40 (UTC) according to the relatively high NO concentration. However, some studies (Michoud et al., 2012;Sörgel et al., 2011) already discussed that the stationary state of HONO can be only reached during noontime. Hence, a

model calculation (named Model 1) was also used to discuss the HONO contribution from the gas-phase reaction of NO with OH radical.

$$\frac{d[HONO]}{dt} = k_7[OH][NO] + k_{het}[NO_2] - J_{HONO}[HONO] - k_{11}[HONO][OH] \qquad \text{(Eq. 10)}$$

$k_{het}$ derived from this work is 0.027 h$^{-1}$, [NO] and [NO$_2$] are averaged concentrations from field measurement. The results are shown in Fig. 6 (orange line, Model 1). It is reasonable to indicate that the reaction of R7 only contribute 30-55% to the HONO increase in the early morning (4:30-7:30 UTC). R7 can continually contribute 50% of the measured HONO from 10:30 to 16:30 (UTC). However, regarding on the large uncertainty of [OH] (a factor of 2), the "unknown HONO sources" exist but could be not crucial. Basically, the additional HONO contribution rate could be estimated from the following equation:

$$P_{unknown} = \frac{d[HONO]}{dt} + J_{HONO}[HONO] + k_{11}[OH][HONO] - k_7[OH][NO] \qquad \text{(Eq. 11)}$$

An additional source of 91±41 pptv h$^{-1}$ was derived beside OH reaction with NO according to a HONO mixing ratio 98±15 pptv for the time period of 10:30 to 16:30 (UTC). This could be well explained by the photochemical processes such as R5 and R9 and would be discussed deeply in the next section.

**4.3.2 Evidence for nighttime deposited HONO as a morning source**

As observed in our field measurement and shown in Fig. 2, the HONO concentrations always presented a strong increase from 4:00 – 7:00 (UTC), which induces three hypotheses as also mentioned in section 3.3: (a) photolysis of gas-phase and particulate nitrate, (b) photosensitized conversion of NO$_2$, (c) dew on ground surfaces served as HONO sink during the night and become a morning source by releasing the trapped nitrite back into ambient air. To identify this HONO source, the chemical box model as expressed in Eq. 12 was extended with additional processes. Heterogeneous reaction of NO$_2$ on the wet surface (R2) and HONO deposition on the ground surface were firstly used to quantify the contributions of the well-known HONO production and loss processes. In addition, the HONO deposition on the ground surface independent on RH (24 hours, named Model 2) and with RH dependence (nighttime 17:00-8:00 UTC, named Model 3) are also discussed.

$$\frac{d[HONO]}{dt} = k_7[OH][NO] + k_{het}[NO_2] - J_{HONO}[HONO] - k_{11}[HONO][OH] -$$

$$\frac{1}{4}\gamma_{HONO,ground}[HONO]\frac{\upsilon_{HONO}}{H} \qquad \text{(Eq. 12)}$$

Both the surface production of HONO through NO$_2$ heterogeneous reaction and subsequent loss by ground surface deposition are already termed in Eq. 5 and Eq. 8, respectively. Here, $k_{het}$ is 0.027 h$^{-1}$ and $\gamma_{HONO,ground}$ is (1.0±0.4)×10$^{-4}$ calculated from the present observations. These values are applied to the model calculation to simulate the diurnal cycle of HONO. As shown in Fig. 6, both Model 2 (blue line) and Model 3 (green square) cannot explain the HONO morning peak but Model 3 can well reproduce the nighttime HONO indicating that

surface loss of HONO is an important sink to consider when the RH was saturated. Hence, Model 3 was used as basic run for the following model calculation.

To investigate the contribution of photolysis of nitric acid and nitrate ($HNO_3/NO_3^-$) (R9) on the diurnal HONO based on the hypothesis (a), the following model calculation (Model 4, pink line) was made:

$$\frac{d[HONO]}{dt} = k_7[OH][NO] + k_{het}[NO_2] + J_{HNO3}[HNO_3/NO_3^-] - J_{HONO}[HONO] - k_{11}[HONO][OH]$$

$$- \frac{1}{4}\gamma_{HONO,ground}[HONO]\frac{\upsilon_{HONO}}{H} \qquad \text{(Eq. 13)}$$

Here gas-phase $HNO_3$ and particle $NO_3^-$ are summed up and the photolysis frequency $J_{HNO3}$ was derived from the TUV model by multiplying an enhanced factor of 30 due to a faster photolysis of particle-phase $HNO_3$ (Romer et al., 2018). As a result, the photolysis of $HNO_3/NO_3^-$ (Model 4, pink line) could not reproduce the HONO morning peak shown in Fig. 6. However, it could well reproduce the HONO for the time period of 10:30 to 16:30 (UTC).

To investigate the contribution of photosensitized conversion of $NO_2$ (R5) on the diurnal HONO based on the hypothesis (b), the following model calculation (Model 5) was performed:

$$\frac{d[HONO]}{dt} = k_7[OH][NO] + k_{het}[NO_2] + \frac{1}{4}(\gamma_a\frac{S_a}{V} + \gamma_g\frac{S_g}{V})\upsilon_{NO2}J_{NO2}[NO_2] - J_{HONO}[HONO] -$$

$$k_{11}[HONO][OH] - \frac{1}{4}\gamma_{HONO,ground}[HONO]\frac{\upsilon_{HONO}}{H} \qquad \text{(Eq. 14)}$$

Here the $\gamma_a$ and $\gamma_g$ are the light-enhanced $NO_2$ uptake coefficients both of $2.0\times10^{-5}$ (Zhang et al., 2016) on both the aerosol surface and ground surface, respectively. $J_{NO2}$ was multiplied with $\frac{light\ intensity}{400}$ when the light intensity is $\geq$ 400 W m$^{-2}$. As shown in Fig. 6 (Model 5, cyan line), the photosensitized $NO_2$ on the aerosol and ground surface could not reproduce the HONO morning peak. This favors the third hypothesis that dew evaporation processes release HONO resulting in the sudden morning peak.

Indeed, as shown in Fig. S10, the HONO morning peak always happens according to a fast decrease of RH between 4:30-9:00 (UTC). However, there is one case happened at 1:00 (UTC) on April 25$^{th}$, 2018, possibly due to an upcoming strong wind which decreased the RH and evaporated the dew water on the ground surface. It should be noted that this HONO morning peak was never observed during this field measurement period without a fast RH decrease, in case of dry ground surface as it was observed during the morning of April 23$^{rd}$, 2018. To figure out the relationship between temporary HONO emission from dew water and decreasing RH, the following equation was defined:

$$k_{emission} = \frac{d(\frac{HONO_{unknown}}{99.5-RH})}{dt} = \frac{\frac{HONO_{unknown}}{99.5-RH}(t_2) - \frac{HONO_{unknown}}{99.5-RH}(t_1)}{(t_2-t_1)} \qquad \text{(Eq. 15)}$$

where $HONO_{unknown} = HONO_{measure} - HONO_{Model4}$ was calculated for each day in the whole campaign period. $k_{emission}$ could be obtained from the linear least square analysis of

$\frac{HONO_{unknown}}{99.5-RH}$ vs. the internal time of HONO morning peak (4:30-7:00, UTC) as shown in Fig. 7. The maximum and minimum of $k_{emission}$ are obtained as 0.026±0.008 and 0.006±0.001 pptv %$^{-1}$ s$^{-1}$, respectively, with an average of 0.016±0.014 pptv %$^{-1}$ s$^{-1}$ as presented in Table 4. The average value was used in the following model calculation to reproduce the diurnal cycle of HONO.

$$\frac{d[HONO]}{dt} = k_7[OH][NO] + k_{het}[NO_2] + J_{HNO3}[HNO_3/NO_3^-] + \frac{1}{4}(\gamma_a\frac{S_a}{V} + \gamma_g\frac{S_g}{V})\upsilon_{NO2}J_{NO2}[NO_2] +$$

$$k_{emission}*(99.5-RH) \quad - \quad J_{HONO}[HONO] \quad - \quad k_{11}[HONO][OH] \quad -$$

$$\frac{1}{4}\gamma_{HONO,ground}[HONO]\frac{\upsilon_{HONO}}{H} \tag{Eq. 16}$$

In Fig. 6, the Model 6 (red line) shows that the amount of deposited HONO could represent the amount of HONO during the morning peak. In Fig. S11, the measured atmospheric HONO mixing ratio and the calculated HONO mixing ratio using model 6 with a minimum dew HONO emission ($k_{emission}$ = 0.006 pptv %$^{-1}$ s$^{-1}$) and a maximum dew HONO emission ($k_{emission}$ = 0.026 pptv %$^{-1}$ s$^{-1}$) is shown. HONO emission from the dew water evaporation represented at least 90% and likely in excess of 100% of the calculated unknown HONO morning peak, which may continually serve as HONO source for the whole daytime as long as water evaporates depending on the weather condition.

**4.3.3 HONO emission from dew water evaporation in the morning**

The hypothetical morning HONO mixing ratio (pptv) due to the complete dew water evaporation could be estimated from the following equation by taking the measured dew nitrite and the mixing layer height:

$$[HONO]= \frac{\alpha \times S_g \times F_{NO2-}}{H \times S_g} = \frac{\alpha \times F_{NO2-}}{H} \tag{Eq. 17}$$

$F_{NO2}^-$ is the $NO_2^-$ concentration per m$^2$ of the glass sampler surface. The mean $F_{NO2-}$ from May 11$^{th}$ 2019 was used for the calculation. $S_g$ represents the surface area of the flat ground (analog to the surface area of the glass sampler), $\alpha$ is the enhanced factor for $V_{dew}$ (dew water sample volume of the glass sampler in Eq.2) due to the larger cold surfaces from grass which can get in contact with humid air than the flat glass sampler. $\alpha$ was calculated as 2×LAI to take the areas on both sides of the leaves and the vegetation-covered areas on the ground into account. And a factor of 6 for LAI was assumed and used in section 4.2.2. However, regarding on the possibly different grass height during the HONO field measurement and dew measurements in April 2018 and May 2019, respectively, we would use a range of 1-6 for LAI in this section. During the HONO peak at 6 or 7 UTC, the mixing height ranged between 175 m and 600 m, while the value ranged from 20 m to 200 m at 0:00 – 5:00 UTC. Hence, the overall concentration increase from this source would be 377-2264, 189-1132, 76-122, 38-226 and 13-76 pptv, if all of the deposited HONO is released into the overlying air column for a mixing height of 20, 40, 100, 200 and 600 m, respectively. Since the released HONO was

subjected to photolysis, using a $J_{HONO}$ from TUV model scaled by global radiation (section 2.7), a maximum [HONO] of 176-1053, 88-527, 35-211, 18-105 and 6-35 pptv for the mixing height 20, 40, 100, 200 and 600 m, respectively, would be contributed from the surface nitrite release at 7:00 UTC after the process started from 4:00 UTC. For a reasonable 100 m mixing height, this would account for 5-30% of the observed HONO morning peak in Fig. 6. This low percentage might be a result of the different sampling time of dew measurement compared with HONO measurement and further studies are required for the exact quantification. Although the above calculations may be well simplified, the results do suggest that the release of the deposited HONO on wet/moist canopy surfaces may contribute to the morning HONO concentrations in the overlying atmosphere right after dew evaporation.

Indeed, few field studies (He et al., 2006;Rubio et al., 2009) have reported that dew water can serve as a sink and a temporary reservoir of atmospheric HONO. Previously, the role of dew as a nighttime reservoir and morning source for atmospheric $NH_3$ has been reported by Wentworth et al. (2016). Our results suggest that nocturnally deposited HONO forms a ground surface reservoir, which can be released in the following morning by dew evaporation. Therefore, a significant fraction of the daytime HONO source can be explained for the Melpitz observations.

### 4.3.4 Impact on the primary OH sources

HONO serves as an important primary source of OH during daytime in the troposphere (Kanaya et al., 2007;Kleffmann et al., 2005;Villena et al., 2011). Seiler et al. (2012) reported that the HONO is almost the only source of OH radicals in the early morning. The morning peak of HONO is mainly released from the dew evaporation and could imply a strong supply of OH radicals and, hence, enhances atmospheric oxidizing capacity in the atmosphere around Melpitz. Here, the net rate of OH radical from the HONO photolysis was calculated and compared with that from ozone photolysis, which is typically proposed as the major OH radical source in the atmosphere where water vapor is not limited.

$$O_3 + h\nu \rightarrow O(^1D) + O_2 \ (\lambda < 320 \ nm) \qquad (R12)$$

$$O(^1D) + H_2O \rightarrow 2OH \qquad (R13)$$

$$O(^1D) + M \rightarrow O(^3P) + M \ (M = N_2) \qquad (R14)$$

Other OH sources, such as photolysis of oxidized VOCs, peroxides and ozonolysis of unsaturated VOCs are not considered due to the lack of measurement data for these radical precursors. The net rate of OH production from HONO photolysis ($P_{HONO->OH}$) was calculated by the source strength subtracting the sink terms due to reactions of R7 and R11. The OH production rate ($P_{O3->OH}$) from $O_3$ photolysis can be calculated by using the method proposed by Su et al. (2008) and Li et al. (2018).

$$P_{HONO->OH} = J_{HONO}[HONO] - k_7[NO][OH] - k_{11}[HONO][OH] \qquad (Eq. \ 18)$$

$$P_{O3->OH} = 2J(O^1D)[O_3](\frac{k_{13}[H_2O]}{k_{14}[M]+k_{13}[H_2O]}) \qquad (Eq. \ 19)$$

Where $J(O^1D)$ was obtained from the TUV model scaled by the global radiation. The temperature dependence of $k_{13}$ and $k_{14}$ are taken from JPL/NASA Evaluation Number 18 (Burkholder et al., 2015). As shown in Fig. 8, the photolysis of HONO produced similar amounts of OH compared with photolysis of ozone at the mean daytime (9:00-14:00, UTC), as $(7.2\pm2.0)\times10^5$ molecule $cm^{-3}$ $s^{-1}$. $P_{O3->OH}$ was, as expected, highest during the highest $J$ values and negligible at the sunrise and sunset. $P_{HONO->OH}$ had a similar trend after the noontime but presented a strong OH production around 7:00 (UTC) due to the HONO morning peak. These results demonstrate the significant role of HONO in the atmospheric oxidizing capacity, especially for areas that experience frequent dew formation. In addition, the OH concentration calculated from the global radiation flux measurement was also shown in yellow color in Fig. 8. The different trend of calculated OH concentration compared with $P_{HONO}$ indicate that the morning OH concentration could be highly underestimated.

**5 Conclusion and Atmospheric Implications**

The inter-comparison of MARGA and LOPAP for the HONO measurement was applied from April 19[th] to 29[th], 2018 at the Melpitz site. Higher HONO mixing ratio (ca. 117%) was obtained from MARGA compared with that of LOPAP caused by heterogeneous reactions within the MARGA WRD or potential sampling inlet artefact.

The maximum dew water $NO_2^-$ concentration per $m^2$ of glass sampler surface was determined to be $7.91\pm2.14$ $\mu g$ $m^{-2}$in May 2019. Thus, under consideration of photolytical losses and homogeneous mixing, the maximum contribution to the HONO morning peak from dew water evaporation could be calculated and ranged from $1053\pm45$ to $35\pm1$ pptv for mixing height of 20 to 600 m, respectively.

Well-defined diurnal cycles of HONO with concentration peaks in the early morning and in the evening are found. High time resolution of HONO measurements revealed (i) the vehicle emission is a negligible HONO source at the Melpitz site; (ii) HONO formed from the heterogeneous reaction $NO_2$ on the ground surface is the dominant nighttime source with a high $NO_2$-HONO conversion frequency of $0.027\pm0.017$ $h^{-1}$; (iii) significant amounts of HONO $(0.16\pm0.12$ ppbv $h^{-1})$ deposited to the ground surface at night. The accurate observations of HONO and $NO_2$ allowed direct estimation of the ground uptake coefficients for these species at night: $\gamma_{NO2\rightarrow HONO\_g} = 2.4\times10^{-7}$ to $3.5\times10^{-6}$, $\gamma_{HONO,ground} = 1.7\times10^{-5}$ to $2.8\times10^{-4}$. The ground uptake coefficient of $NO_2$ and HONO are within the ranges of laboratory and model coefficients. The range of HONO uptake coefficient values calculated in this investigation are potentially limited by a combination of transport and diffusion to the ground surface.

A chemical model utilizing observational constraints on the HONO chemical system and known sources and sinks support the hypothesis that dew water on the ground surface, especially on leaf surfaces, behave as a sink at night and a temporary reservoir for atmospheric HONO in the morning. The dew evaporation had a negative relationship with the

RH in the atmosphere and, hence, the HONO emission rate was estimated to be $0.016\pm0.014$ pptv $\%^{-1}\,\text{s}^{-1}$ dependent on the RH after sunrise (start from 4:00, UTC). Furthermore, the formation and evaporation of dew on the ground surface influence significantly the air-surface exchange of HONO and, thus, its temporal distributions in the atmospheric boundary layer in the morning and night. The OH production rate from the photolysis of HONO compared with that from photolysis of $O_3$ showed that this dew emission of HONO can strongly enhance the OH reactivity throughout morning time and, hence, plays a vital role in the atmospheric oxidation.

**Data availability**

The compiled datasets used to produce each figure within this paper are available as Igor Pro files upon request. The Supplement related to this article is available online at doi:10.5194/acp-2019-1088-supplement.

**Author contributions**

RY wrote the paper with input from all authors. BS and GS analyzed the MARGA and dew data and wrote the paper. RY and BG conducted the HONO measurement using LOPAP. TT and AW were responsible for the particle measurement. AM and HH designed the experiments and lead the campaign. All co-authors commented on the manuscript.

**Competing interests**

The authors declare to have no competing interests.

**Acknowledgements**

The authors acknowledge financial support of this study and deployment of the MARGA system by the German Federal Environment Agency (UBA) research foundation under contracts No:351 01 093 and 351 01 070, as well as the European Union (EU) for the Transnational access (TNA) under ACTRIS-2: Comparison of HONO-measurements with MARGA and LOPAP at TROPOS research-site Melpitz (MARLO) is part of the project that has received funding from the European Union's Horizon 2020 research and innovation programme under grant agreement No 654109. For the laboratory analysis and the preparation of solutions, we thank A. Dietze, A. Rödger and S. Fuchs. For the support especially in the field, we thank R. Rabe and A. Grüner. We thank also the TROPOS mechanical workshop for the construction of the dew sampler. The CNRS team (Orléans-France) acknowledges the support from Labex Voltaire (ANR-10-LABX-100-01) and ARD PIVOTS program (supported by the Centre-Val de Loire regional council). Europe invests in Centre-Val de Loire with the European Regional Development Fund.

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

**Table 1**. Mean and mean error as 2 times the standard deviation of the measured HONO (LOPAP) and the other pollutants in the Melpitz station during daytime (D, 04:00-18:00, UTC) and nighttime (N, 18:00-04:00, UTC).

| | D | N | | D | N |
|---|---|---|---|---|---|
| NO (ppbv) | $1.0 \pm 0.5$ | $0.5 \pm 0.3$ | HCl (ppbv) [b] | $0.02 \pm 0.03$ | $0.01 \pm 0.01$ |
| NOx (ppbv) | $4 \pm 1$ | $6 \pm 2$ | $HNO_3$ (ppbv) [b] | $0.2 \pm 0.1$ | $0.2 \pm 0.1$ |
| $NO_2$ (ppbv) | $3 \pm 1$ | $5 \pm 2$ | $NH_3$ (ppbv) [b] | $17 \pm 7$ | $8 \pm 4$ |
| HONO (pptv) [a] | $162 \pm 96$ | $254 \pm 114$ | $Cl^-$ ($\mu g\ m^{-3}$) [b] | $0.03 \pm 0.04$ | $0.01 \pm 0.01$ |
| $O_3$ (ppbv) | $36 \pm 7$ | $19 \pm 13$ | $NO_3^-$ ($\mu g\ m^{-3}$) [b] | $3 \pm 2$ | $2 \pm 1$ |
| $SO_2$ (ppbv) | $0.8 \pm 0.4$ | $0.5 \pm 0.3$ | $SO_4^{2-}$ ($\mu g\ m^{-3}$) [b] | $1.4 \pm 0.5$ | $1.3 \pm 0.6$ |
| T (°C) | $16 \pm 3$ | $11 \pm 5$ | $Na^+$ ($\mu g\ m^{-3}$) [b] | $0.02 \pm 0.03$ | $0.01 \pm 0.01$ |
| RH (%) | $67 \pm 7$ | $85 \pm 11$ | $NH_4^+$ ($\mu g\ m^{-3}$) [b] | $1.1 \pm 0.7$ | $0.8 \pm 0.4$ |
| Wind speed ($m\ s^{-1}$) | $3 \pm 2$ | $1.2 \pm 0.7$ | $K^+$ ($\mu g\ m^{-3}$) [b] | $0$ | $0.001 \pm 0.002$ |
| HONO/NOx (%) | $0.04 \pm 0.02$ | $0.05 \pm 0.02$ | $Mg^{2+}$ ($\mu g\ m^{-3}$) [b] | $0.03 \pm 0.01$ | $0.02 \pm 0.04$ |
| NO/NOx (%) | $0.3 \pm 0.1$ | $0.1 \pm 0.1$ | $Ca^{2+}$ ($\mu g\ m^{-3}$) [b] | $0.2 \pm 0.1$ | $0.2 \pm 0.1$ |
| OH (molecule $cm^{-3}$) | $(2.8 \pm 0.7) \times 10^6$ | | $NO_2^-$ ($\mu g\ m^{-3}$) [b] | $0.01 \pm 0.01$ | $0.03 \pm 0.02$ |

[a] HONO derived from LOPAP;

[b] data obtained from the MARGA instrument

**Table 2.** Nitrite concentration measured in dew water.

| Date 2019 | Plate number | Initial hour (UTC) | Final hour (UTC) | Volume (ml) | Blank $NO_2^-$ ($\mu g\ L^{-1}$) [a] | Final $NO_2^-$ ($\mu g\ L^{-1}$) [b] | $F_{NO2-}$ ($\mu g\ m^{-2}$) | pH [c] |
|---|---|---|---|---|---|---|---|---|
| May 8[th] | 1 | 18:00 | 5:25 | 76.60 | 0.0018 | 41.87 | 2.10 | 6.40 |
| | 2 | | 5:45 | 75.60 | 0.0017 | 42.84 | 2.20 | 6.45 |
| May 11[th] | 1 | 18:00 | 3:20 | 94.00 | 0.0055 | 128.23 | 8.00 | 7.00 |
| | 2 | | 4:20 | 80.00 | 0.0005 | 120.43 | 6.40 | 6.90 |
| | 1 | 3:30 | 5:20 | 13.00 | 0.0006 | 164.62 | 1.43 | 7.00 |
| May 13[th] | 1 | 18:00 | 4:45 | 72.00 | 0.0001 | 43.87 | 2.10 | 6.30 |
| | 2 | | 5:20 | 79.00 | 0.0001 | 58.81 | 3.10 | 6.40 |
| May 14[th] | 1 | 18:00 | 5:00 | 15.00 | 0.0001 | 148.90 | 1.50 | 6.80 |
| | 2 | | 5:00 | 21.00 | 0.0001 | 91.44 | 1.30 | 6.70 |

[a] note that the blank $NO_2^-$ concentration is below the detection limit of 0.02 $\mu g\ L^{-1}$.

[b] Final $NO_2^-$ = Raw $NO_2^-$ - Blank $NO_2^-$

[c] pH was measured by a pH meter on a subsample of the total volume

**Table 3.** The ratio $HONO_{corr}/NO_2$ and the $NO_2$-HONO conversion frequency during early nighttime.

| Date | UTC | $R^2$ | $HONO_{corr}/NO_2$ | $k_{het}$ ($h^{-1}$) |
|------|-----|-------|---------------------|----------------------|
| 19/04/2018 | 17:30-19:50 | 0.45 | 0.118±0.010 | 0.043±0.002 |
| 21/04/2018 | 18:20-20:30 | 0.64 | 0.055±0.004 | 0.012±0.002 |
| 22/04/2018 | 18:10-21:20 | 0.79 | 0.161±0.005 | 0.030±0.002 |
| 25/04/2018 | 17:31-21:20 | 0.69 | 0.061±0.003 | 0.010±0.001 |
| 27/04/2018 | 18:00-23:41 | 0.48 | 0.113±0.006 | 0.016±0.001 |
| 28/04/2018 | 18:00-19:50 | 0.44 | 0.152±0.008 | 0.050±0.004 |
|  |  |  | 0.110±0.041 | 0.027±0.017 |

**Table 4.** Summary of the temporary HONO emission rate from dew water, $k_{emission}$ from April 19th to 29th, 2018.

| Period | $k_{emission}$ (pptv $\%^{-1}$ $s^{-1}$) | |
|---|---|---|
| | Min | Max |
| 21/4/2018 | 0.0054 | 0.0357 |
| 22/4/2018 | 0.0048 | 0.0314 |
| 24/4/2018 | 0.0057 | 0.0192 |
| 26/4/2018 | 0.0067 | 0.0302 |
| 27/4/2018 | 0.0048 | 0.0215 |
| 28/4/2018 | 0.0079 | 0.017 |
| **mean** | 0.006±0.001 | 0.026±0.008 |
| **Total average** | 0.016±0.014 | |

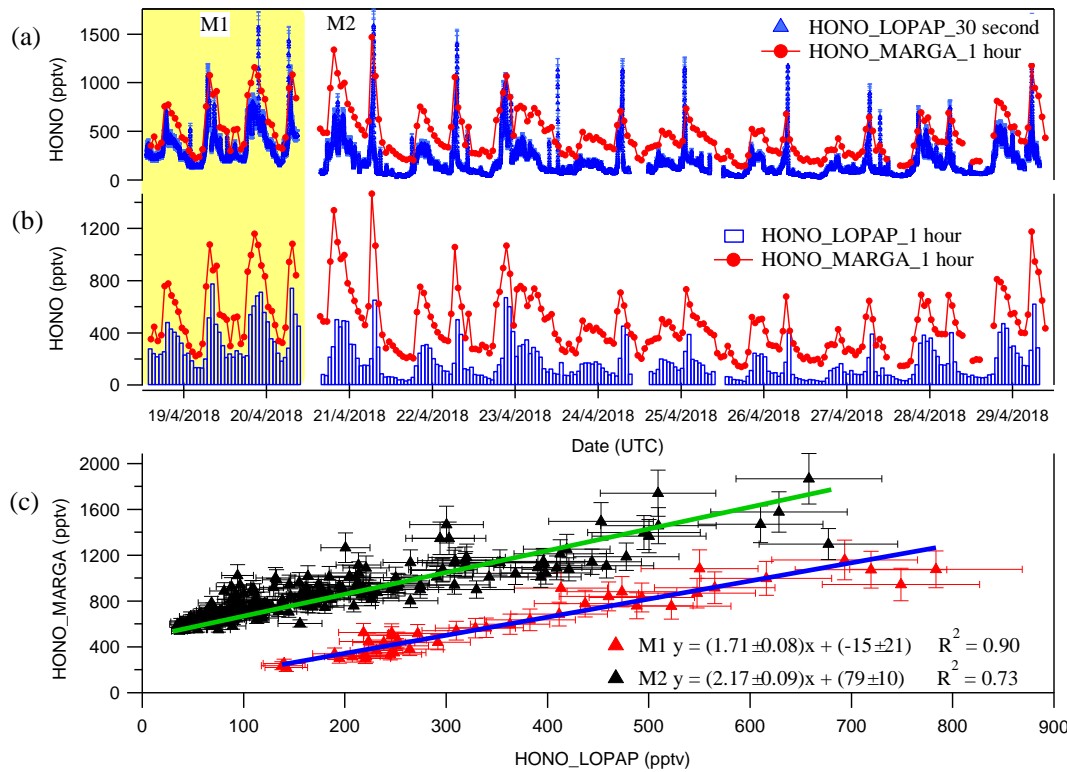

**Fig. 1**. Time courses of HONO as hourly measured by MARGA and 30 seconds measured by LOPAP (a) and normalized hourly for LOPAP (b). (c) blue and green lines represent the error weighted orthogonal regression analysis between MARGA and LOPAP for two different comparison period of M1 and M2, respectively. The error bar in the panel (c) indicates the measurement error of HONO concentrations in LOPAP and MARGA. The HONO concentration of MARGA in panel (c) is shifted 400 pptv for clarity.

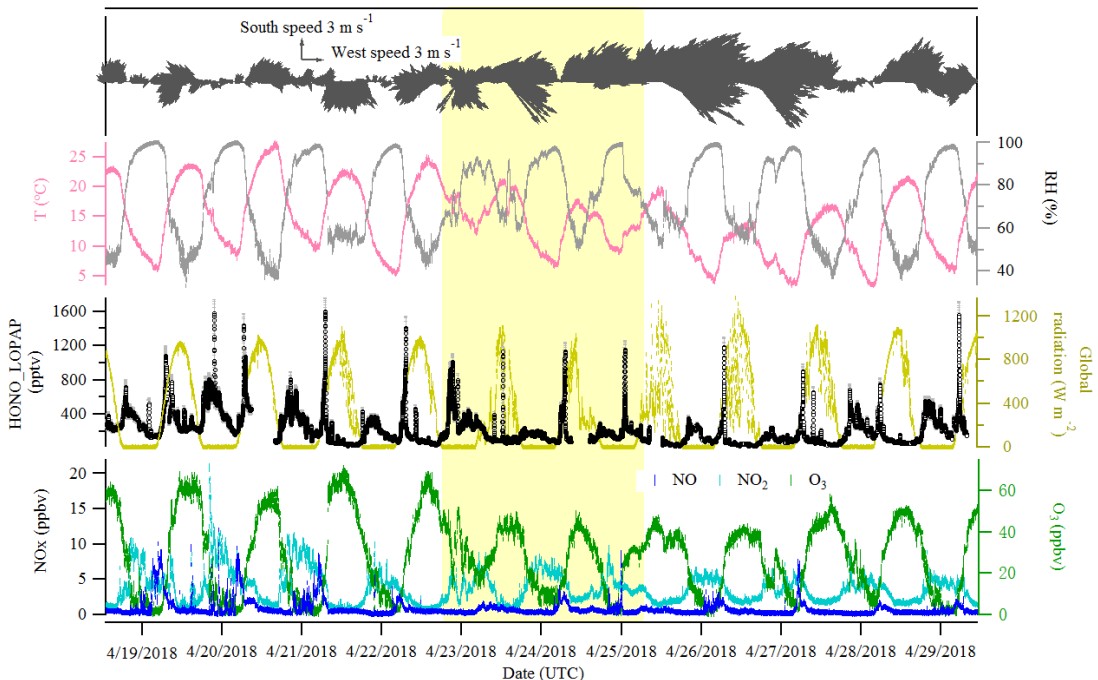

**Fig. 2**. Time series of HONO (LOPAP measurement), NO, $NO_2$, $O_3$, global radiation, temperature (T), relative humidity (RH) and surface wind in Melpitz from April 19th to 29th, 2018. The gaps were mainly due to the maintenance of the instruments. The yellow shadow indicates two sets of observations discussed in section 3.3. The gray color in the HONO panel indicates the measurement error of HONO concentrations.

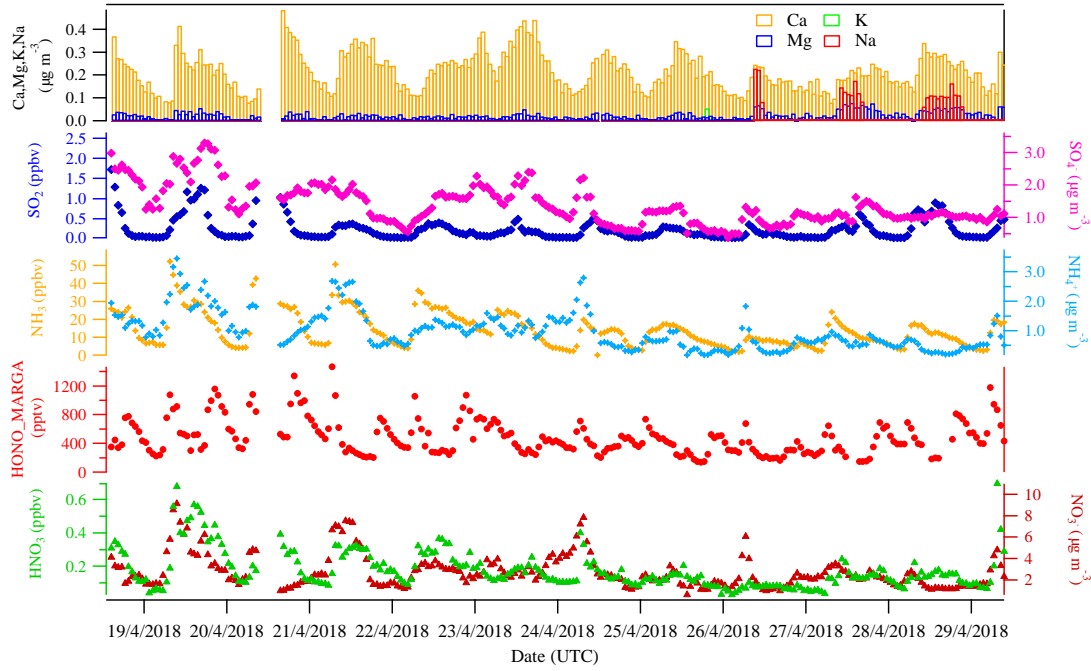

**Fig. 3**. The hourly time-resolved quantification of water-soluble ions in $PM_{10}$ ($NO_3^-$, $SO_4^{2-}$, $NH_4^+$, $Na^+$, $K^+$, $Mg^{2+}$, $Ca^{2+}$) and their corresponding trace gases (HONO, $HNO_3$, $SO_2$, $NH_3$) were measured by MARGA in Melpitz from April 19th to 29th, 2018.

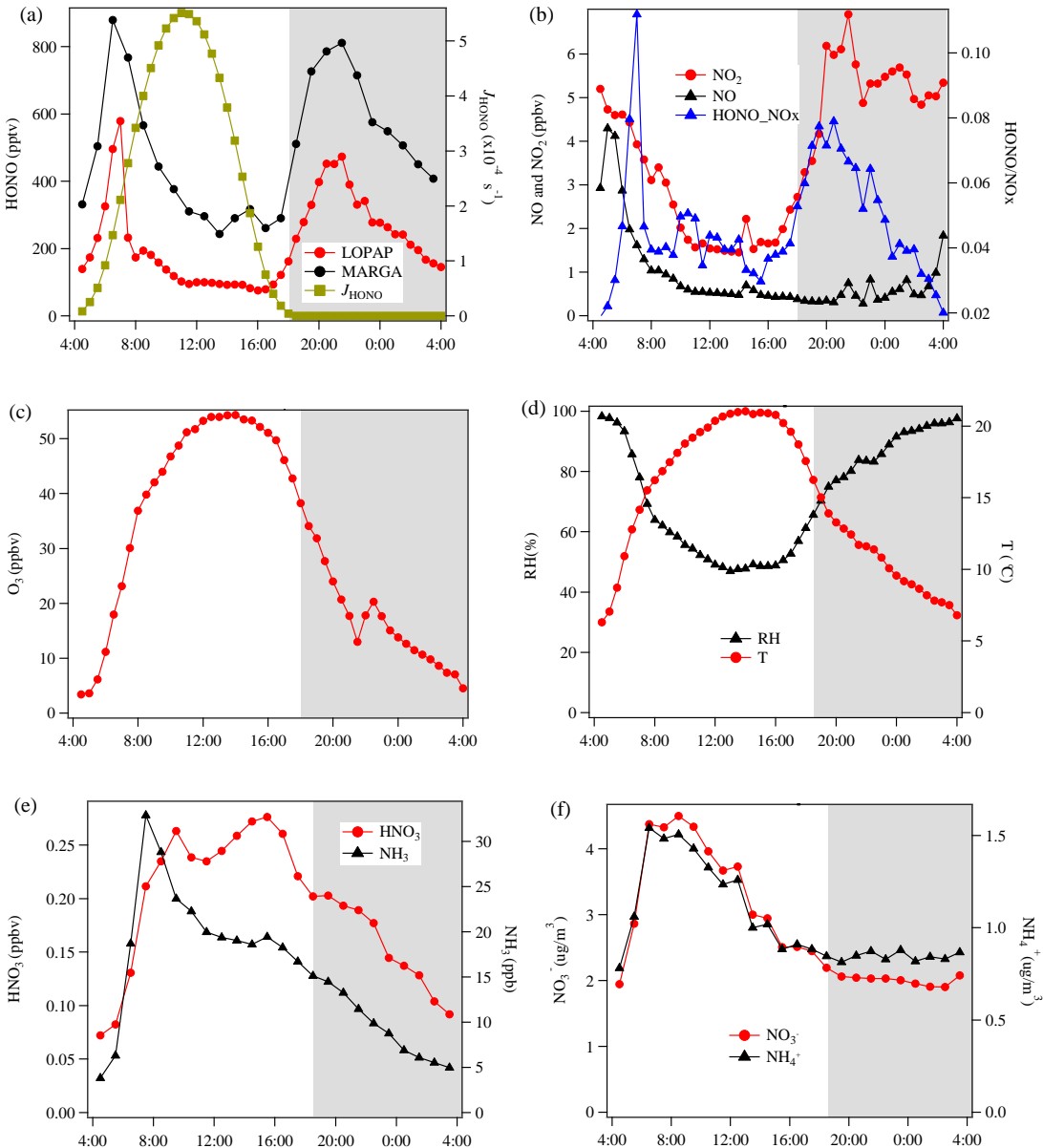

**Fig. 4**. Diurnal variations of HONO and related species during the measurement period except for two sets of observations show in Fig. 5 at Melpitz site. The photolysis rate of HONO was obtained from the TUV model. The grey shaded area indicates the nighttime period (18:00-04:00 UTC).

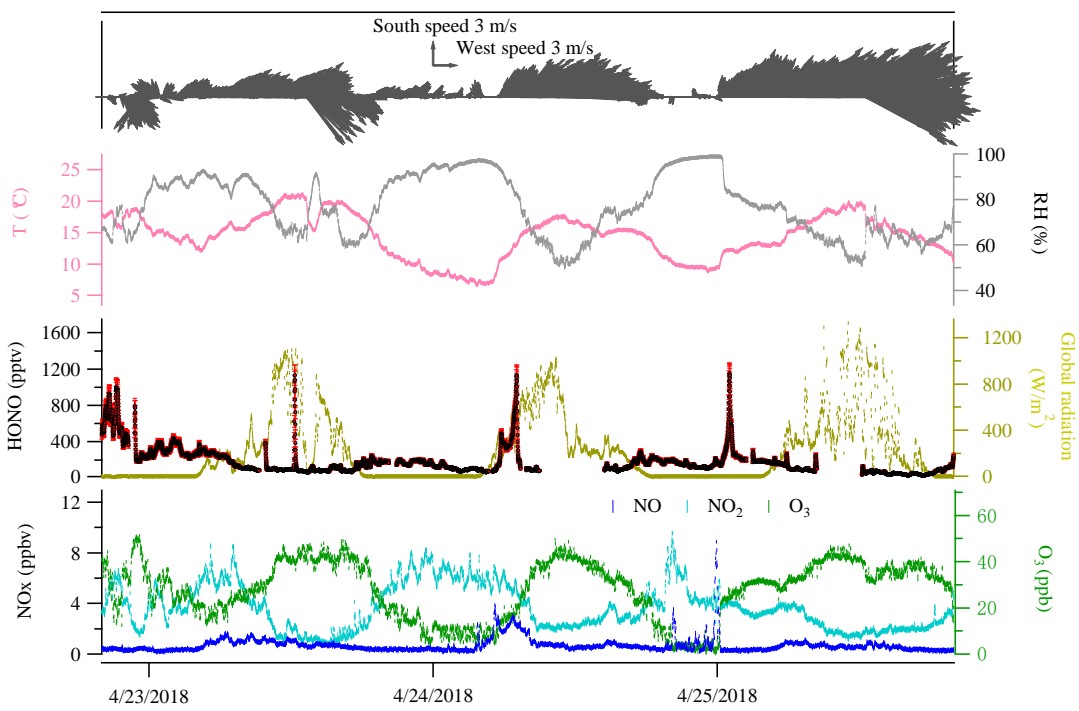

**Fig. 5**. Case events for HONO (LOPAP) and related species at Melpitz site during the day April 23$^{rd}$ to 25$^{th}$, 2018. The red color in the HONO panel indicates the measurement error of HONO concentrations.

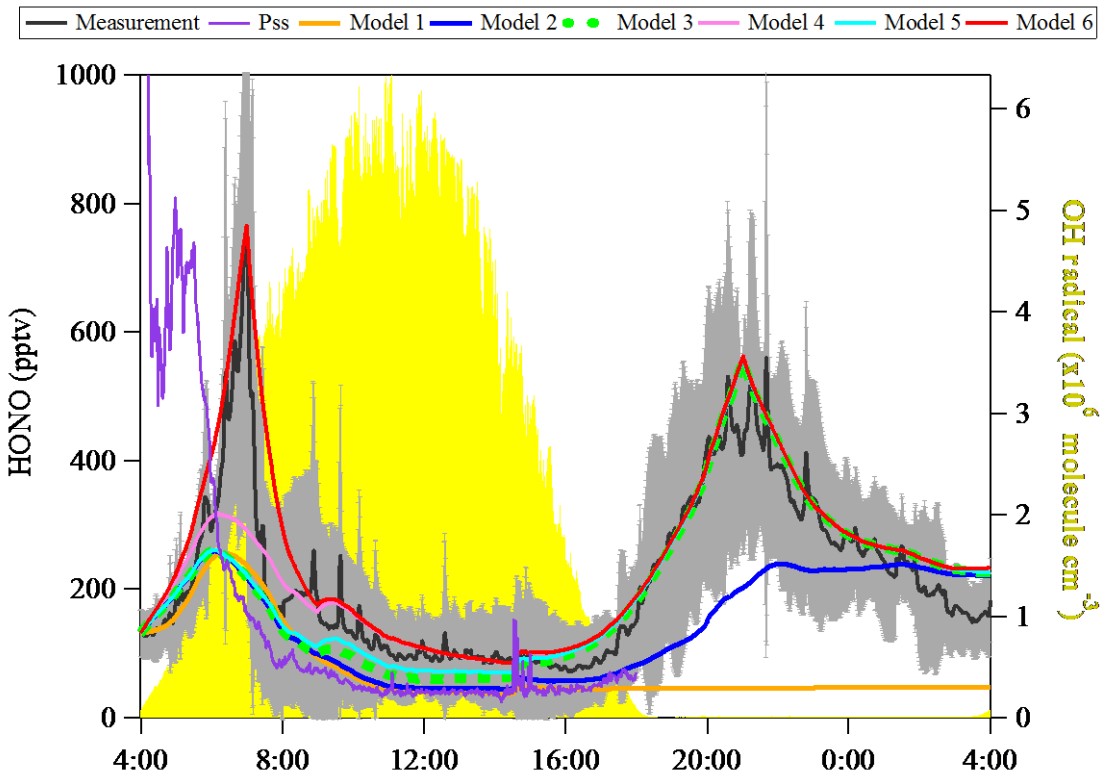

**Fig. 6.** Observed average HONO atmospheric concentration (black line, +-1σ in shaded area) and the model calculated HONO concentration including different HONO production and loss processes. PSS presents model results by assuming an instantaneous photo-equilibrium between the gas-phase formation (R7) and gas-phase loss processes (R1 and R11) of HONO; Model 1 includes R1+R7+R11. Model 2 includes R1+R2+R7+R11+surface deposition (00:00-00:00), whereas Model 3 describes R1+R2+R7+R11+surface deposition (17:00-8:00). And Model 3 is used to be the base to investigate the effect of R9 (Model 4), R5 (Model 5) and the combination of R5+R9+Dew HONO emission (4:30-7:00) (Model 6).

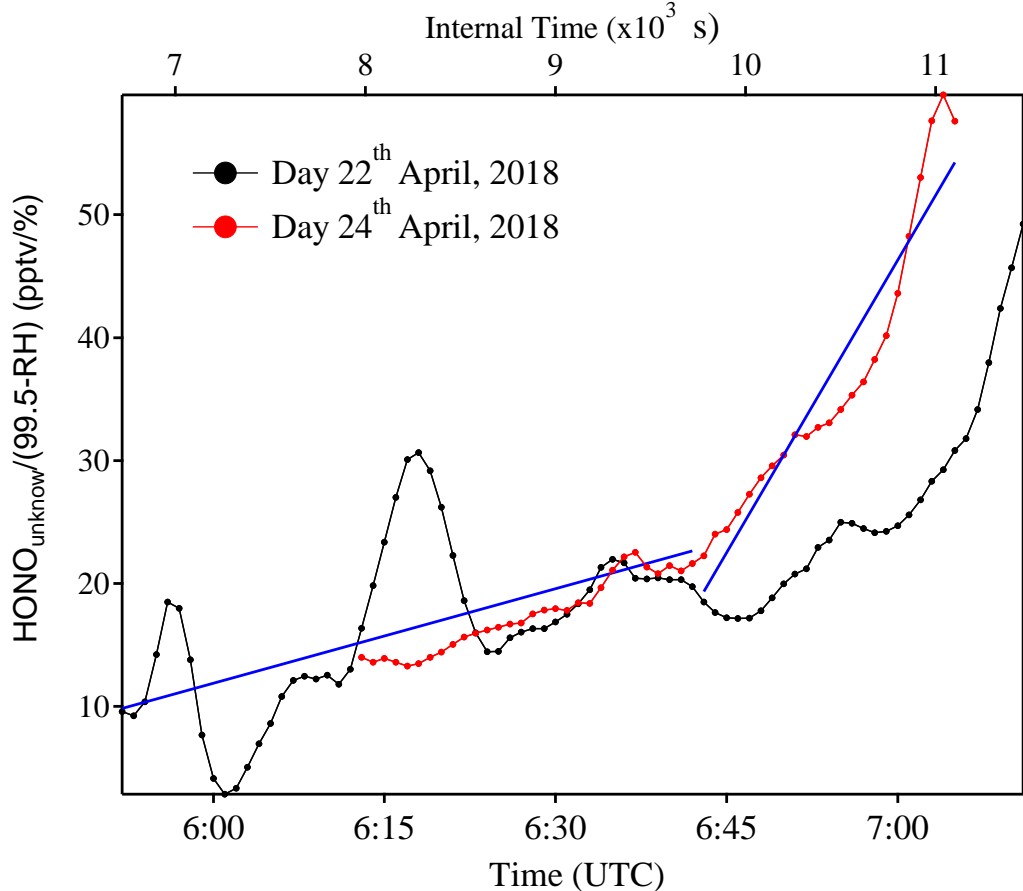

**Fig. 7.** Example of $\frac{HONO_{unknown}}{99.5-RH}$ as a function of time (zero point from time 4:30, UTC) to estimate the temporary HONO emission rate from dew water ($k_{emission}$). Blue line is the linear least-square analysis of $\frac{HONO_{unknown}}{99.5-RH}$ vs. internal time to obtain the minimum (e.g. 22[th] April for the low slope) and maximum (e.g. 24[th] April for the high slope) of $k_{emission}$, respectively.

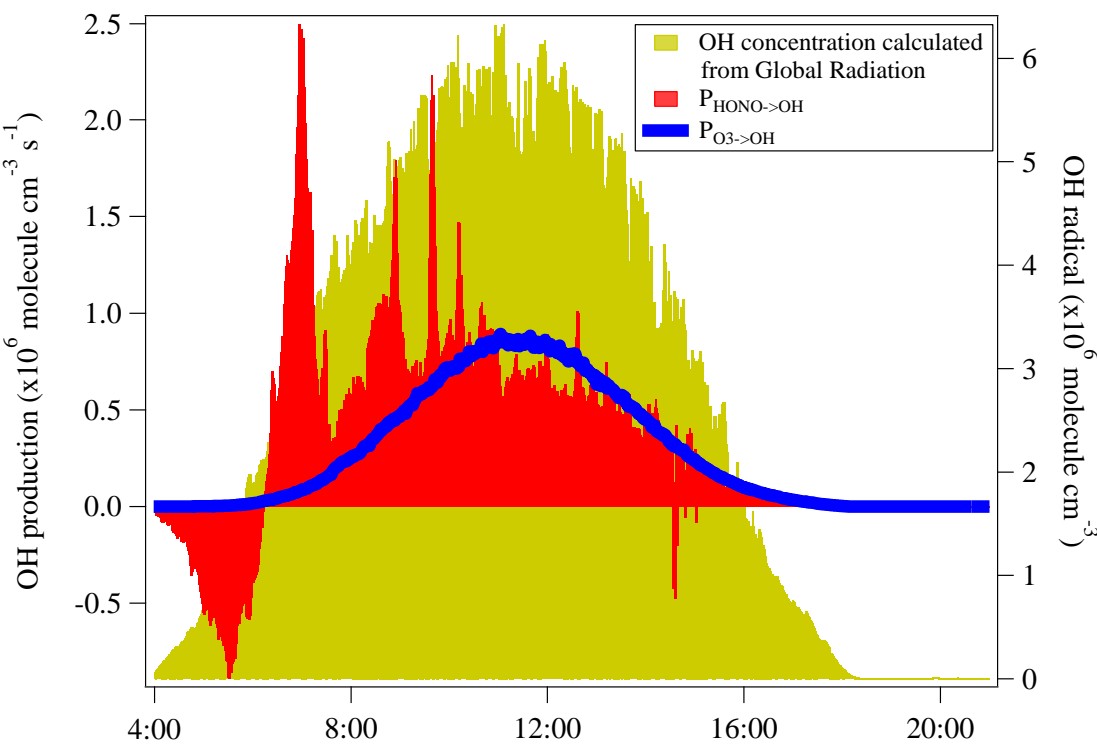

**Fig. 8**. The OH production rates from photolysis of HONO and $O_3$ in Melpitz station from April 19$^{th}$ to 29$^{th}$, 2018. The OH concentration is also shown as yellow area plot, which was calculated from the global radiation flux measurement: [OH]=A*Rad taken from Größ et al. (2018).