# Peer review of "Role of the dew water on the ground surface in HONO distribution: a"

_Atmospheric Chemistry and Physics, 2019_

## Referee Comment (RC1) · Anonymous Referee #1 · 7 Feb 2020

In the manuscript by Y. Ren et al. HONO was measured at the rural station Melpitz in Germany by two different commercial instruments, which were intercompared showing strong interferences and inlet artefacts for one of the instruments. In addition, the measurement data including dew water analysis were used to demonstrate that HONO deposition during night-time and re-emission during the early morning when the relative humidity decrease are important processes in good agreement with former studies. In addition, HONO formation during night- and daytime is discussed and the contribution of HONO photolysis to the daytime formation of OH radicals is compared with the typically proposed main source of OH radicals by O3 photolysis, showing a similar contribution of both sources.

The study contains some interesting information and may be considered for publication

in ACP, after significant concerns have been considered.

Major Concerns:

1) Chemical reactions:

I found the manuscript difficult to read, since all discussed chemical reactions are only summarized in the supplement. At least the important reactions should be shown in the main text.

2) Dew and gas measurements:

In the experimental section, gas phase und dew measurements are explained. However, these measurements were not done in a single field campaign, but the dew measurements were performed more than one year later after the gas phase measurements. Later the average dew nitrite data is used to explain the morning peaks of HONO observed one year earlier. This method will cause large uncertainties, since the gas and dew concentrations, but also other parameters may significantly vary from year to year (apples and oranges. . .). E.g. while the temperature was well above freezing during the gas phase campaign (see Fig. 2), dew water was freezing during the later dew campaign (see line 304). However, if water is freezing, oxidation of nitrite is significantly accelerated (see e.g. Nature, 358, 1992, 736-738). Here parallel gas phase and dew measurements are clearly necessary in the absence of frozen dew water.

3) Intercomparison:

The intercomparison results should be clearer discussed. Since all known chemical interferences are positive interferences (overestimation of the HONO data) and since I expect that both groups can calibrate their instruments with high accuracy, the results shown in Fig. 1 are quite clear. First, the MARGA instrument overestimates HONO during this field campaign at least (the LOPAP instrument may also have interferences. . .) by ca. 90 % (see data M2, where both instruments are operated in their normal way),

and not by 58 % as mentioned in the conclusion (line 619). Second, these ≥90 % are caused by ca. ≥60 % chemical interferences inside the MARGA instrument (see data M1, where both instruments used the common MARGA inlet), e.g. by oxidation of SO2 (see e.g. Spindler et al., 2003) or VOCs by NO2, which are corrected for by the LOPAP instrument. In addition, ca. 30 % of the HONO MARGA data results from heterogeneous formation of HONO in the inlet of this instrument (PM10 inlet + Teflon line), see the difference between the slopes M2 and M1.

So one important conclusion is that MARGA HONO field data should not be used. This result is in excellent agreement with a former intercomparison of both instrument types in a Chinese field campaign (see J. Geophys. Res., 2010, 115, D07303, doi: 10.1029/2009JD012714) where also a RWAD instrument (similar to the MARGA) over-estimated HONO by a factor of three on average. In this context, the statement in line 104 ("first inter-comparison...") is not correct. In addition, the results also show that the use of massive sampling inlets – even if they are coated by Teflon – should not be used for any in-situ HONO instrument.

4) OH data:

For the discussion of the HONO sources the OH data is necessary (see e.g. reaction 3), which was calculated here by a simple linear correlation with the global radiation. However, this method is highly uncertain since first, short wave UV radiation should at least be used (see similar studies using J(O1D)...), cf. main sources of OH-radicals. Here the ratio between the global radiation and J(O1D) will show strong diurnal and seasonal variability (depends mainly on the SZA...). In addition, while the correlation between J(O1D) and OH is indeed often linear, the slope is highly variable and will e.g. depend on the VOC/NOx ratio. Thus, I expect easily a factor of two uncertainties in the calculated OH concentration. Since e.g. half of the HONO daytime levels could be explained by the gas phase reaction (3), see lines 496-497, a factor of two higher OH level could make all discussions about any "unknown HONO sources" obsolete.

In addition, how have the authors calculated the night-time OH levels by this method (ca. 10ˆ4 cmˆ-3, see lines 329-330)? During night-time there is no radiation and calculated OH should be zero... Normal OH night-time levels decrease from ca. 10ˆ6 cmˆ-3 in the early night to 10ˆ5 cmˆ-3 in the later night caused by night-time sources of radicals (O3+alkenes, NO3+alkenes...). This data and the corresponding sections should be removed.

5) Correlation analysis:

In several sections throughout the manuscript correlations were used to identify source processes, which is highly uncertain and which often leads to wrong conclusions. Already in the early 1990 high correlations of Radon with HONO were observed, which are simply caused by the variation of the BLH/vertical mixing for two ground surface sources. Nobody would conclude that Radon is a precursor of HONO. However, for correlations of HONO with different parameters exactly this is done (not only in the present study...). E.g. in lines 356-358 correlation of HONO with relative humidity is explained by the heterogeneous reaction 2NO2+H2O, reaction (2). Besides that this reaction is far too slow to explain the night-time formation (gamma for R2 ca. 10ˆ-7 – 10ˆ-8, one to two orders of magnitude faster kinetics is necessary, see e.g. line 412), the correlation of HONO with humidity may be artificial! During night-time the ground surface is cooling which leads to a) increasing relative humidity, b) decreasing vertical mixing c) increasing surface to volume ratio of the lower nocturnal boundary layer d) increasing rate for any heterogeneous reactions (which scale with S/V...) and increasing levels of ground emitted species, like e.g the proposed HONO-precursor NO2 or the particle surface area of freshly emitted particles. All these changes lead to artificial correlations (e.g. HONO with r.h., with particles,...) from which one should not necessarily conclude source processes. All the correlation analysis should be much more carefully discussed and results should be checked for plausibility, see below.

5) Heterogeneous kinetics

While equation (5) for the calculation of the heterogeneous conversion of NO2 to HONO is correct for small uptake coefficients, at least when a 100 % HONO yield is assumed (not explained here; only valid for fresh soot, a minor constituent of particles...), the calculated uptake coefficients (ca. 10ˆ-15...) to explain the missing night-time formation of HONO on particles (as a limiting case) are completely unreasonable! As the authors later correctly mention, the S/V ratio of particles is typically orders of magnitude lower than for ground surfaces. Thus, a higher (!) gamma is necessary for particles compared to the ground. Typically, formation of HONO by NO2 conversion on particles can be only explained, if gamma values in the range 10ˆ-3 to 10ˆ-4 are used. Otherwise this low number would mean that HONO formation could be easily explained by a reasonable uptake kinetics on particles (ca. 10ˆ-6; see lab studies on several heterogeneous NO2 reactions...)!? Here the authors should check their calculation – I expect some large order of magnitude errors, e.g. by using a wrong unit of the S/V ratio.

Besides this, the use of an uptake coefficient is not recommended when a ground surface conversion is considered (see equation 6), at least for large geometric uptake coefficients and for low night-time vertical mixing (see present study). If a leave area index of 10 is used (see line 409) the obtained "true uptake coefficient" of ca. 10ˆ-5 converts into a "geometric uptake coefficient" of ca. 10ˆ-4. For such high values the transport gets rate limiting for a stable night-time atmosphere!

In this case better a flux concept including resistances for convective mixing and molecular diffusion (Ra and Rb) and a surface resistance (Rc) should be used. Only Rc can be converted into an uptake coefficient and vice versa. The inverse of all resistances leads to the deposition velocity from which a surface uptake flux can be derived by multiplying with the concentration. In many cases this deposition velocity is only depending on the transport resistances, which depend e.g. on the wind speed (Ra and Rb can be estimated by parameterizations, see e.g. VDI 3782). Using constant uptake kinetics makes no sense here.

Also the factor 1/8 in equation 6 (compared to the 1/4 in equation (5)) should be explained, where the authors obviously propose the (too slow, see above) reaction of 2 $NO_2+H_2O$ (R2) as the main HONO source, for which a formal HONO yield of 50 % is used. In contrast, several former gradient studies (e.g. J. Geophys. Res., 2002, 107 (D22), 8192, doi:10.1029/2001JD000390 and Atmos. Chem. Phys, 2017, 17, 6907-6923, doi: 10.5194/acp-2016-1030) found experimental HONO yields from the $NO_2$ uptake on the ground in the very low % range (2-4%...), i.e. only ca. 3% of the deposited $NO_2$ is converted into HONO on the ground during night-time and not 50 %...

Furthermore the HONO deposition on the ground and the derived necessary uptake coefficients (see section 4.2.3) are impossible! Besides the same argument as for the $NO_2$ uptake on ground surfaces (see above, the use of uptake coefficients for fast ground uptake makes no sense, better use the flux concept and a variable deposition velocity...), the values for the HONO uptake coefficient in the range 5.6-19.5 are impossible!? When I was reading the abstract (line 25), I first expected that the authors simply missed the order of magnitude after the given numbers (e.g. $x10^-6...$). Even if one considers a LAI of 6 (see line 410, for the HONO uptake this is not specified...?) the maximum calculated value could be only 6 but never 19.5, which would imply a real uptake coefficient using the true surface area larger the unity. Please check for the definition of the uptake coefficient, e.g. by IUPAC, with a maximum value of one. Reason for the order of magnitude errors is equation (9), where the concept of the deposition velocity is mixed with the concept of the uptake coefficient. To calculate L(HONO) (=dc/dt) a first order rate coefficient k (s-1) is multiplied by the concentration (dc/dt = − k x c). The first order rate coefficient for a heterogeneous reaction is calculated from the uptake coefficient by:

k= 1/4 x gamma x average molecular velocity of HONO x S/V.

S/V can be exchanged by 1/H, as done by the authors. So instead of using the deposition velocity (3.35 cm/s, line 450) in equation (9) the mean molecular velocity of

[Figure]

HONO (ca. $3.7 \times 10^4$ cm/s) should be used, leading to four orders of magnitude lower values. . . And for these high values ($> 10^{-4}$) the HONO uptake is definitely transport limited, see above.

And finally, I do not understand the concept used for the quantification of the deposited HONO, which is later compared to the dew water nitrite. Here the total deposited HONO is given in the unit ppt (see e.g. line 428) but should be given at the end in molecules/cm2 to compare that with the dew nitrite (similar unit). To what S/V or boundary layer height does that "deposited mixing ratio" relate? Do the authors expect that the concentration change for the deposited HONO (dc/dt) is constant in whole boundary layer? Here a surface density (molecules/cm^2) should be derived by integrating the product of the variable (turbulence depending, see above) deposition velocity with the concentration.

Specific Concerns:

The following concerns are listed in the order how they appear in the manuscript.

Lines 22-23: Specify the deposited HONO in the same unit as the dew nitrite surface density (e.g. molecules /cm^2).

Line 25: correct the gamma values (see main concerns).

Line 47: The paper by Gutzwiller et al. is not on the NO2+soot reaction but on the reaction of NO2 with semi-volatile hydrocarbons (see line 49). Better use the study by Arens et al. from the same group or the first studies from 1998 by Ammann et al., or Gerecke et al.

Line 50 and table S1: The authors should distinguish between the oxidation of phenols etc. in the dark (reaction 2b) and the photosensitized conversion of NO2 (see Stemmler et al.) by adding a new reaction for the daytime HONO formation (e.g. new reaction 3).

Line 59-60, the heterogeneous reaction NO+NO2+H2O is completely unimportant and not state of the art. In addition it should be "Andrés-Hernández et al.".

Line 61: There is only one study by Zhang and Tao (delete the a) und the same reference is listed twice in the references (lines 899-904).

Line 61: the reaction NO2*+H2O was studied by Li et al. and not by Finlayson-Pitts. In addition, also this source is completely unimportant (see Carr et al. and Amedro et al.) and was simply an Excimer laser two-photon artefact. . ..

Line 63: The heterogeneous reaction of NO with adsorbed HNO3 is also completely unimportant at atmospheric conditions (gamma <10ˆ-9, see J. Phys. Chem. A, 2004, 108, 5793-5799).

Line 64: delete the "a" for Zhou et al. and delete again one of the double references at the end (lines 908-913).

Line 74: either use R1 or reaction 1 (unify).

Lines 77-78: delete the last sentence; that describes already the results.

Line 85: the instrument can measure down to 0.2 ppt, see Atmos. Chem. Phys., 2008, 8, 6813-6822, doi: 10.5194/acp-8-6813-2008.

Line 95-97: Stieger et al. also intercompared HONO, which should be mentioned here (HNO3 is not the topic of this manuscript. . .).

Line 104-105. The statement is not correct, see major concerns.

Line 121: The used SJAC uses 100 °C hot water steam forming hot steam droplets on which different reactions of NO2 form nitrite (e.g. NO2 + organics, see Gutzwiller et al.) which show positive temperature dependencies. Thus also the aerosol nitrite MARGA data should be used with caution. This interference can be easily tested by spiking HONO free NO2 to the instrument during a field measurement (=> % NO2 interferences for nitrite...).

Line 165: At what temperature were the nitrite solutions stored in the fridge? Should not be below freezing temperature, see above.

Line 195 ff: Were the clear sky J-values from the TUV model scaled by the measured global radiation for short fluctuations by local cloud cover (see figure 2) or was that really done by data from the NASA web page (see line 201)? Normally this is done by scaling the clear sky TUV values with measured radiation (e.g. from a J(NO2) filter radiometer...).

Lines 213-215: I do not understand that statement. While the surface pH of a dry batch Na2CO3 denuder should be very high (pH=10?) the pH of the MARGA is close to neutral (5.7, see line 121), so they are different!?

Lines 262-264: No that statement is not corrcet, the trend of HONO (strongly decreasing during daytime) is different to HNO3 (almost constant), see Figure 4. Reasons are the decreasing HONO precursor concentration NO2 (increase of the BLH; both are ground emitted or formed species), while HNO3 is formed by NO2+OH homogeneously in the gas phase and decreasing NO2 is compensated by increasing levels of OH during daytime...

Line 271-273 and figure 4: Please also show the HONO/NOx ratio in figure 4.

Lines 274-276: A formation or loss reflects dc/dt while a frequency (s-1) is a first order rate coefficient (apples and oranges), reformulate the sentence.

Line 286-289: Under acidic conditions HONO is not highly soluble, cf. the pKa. In addition the dew water was neutral during the dew campaign, see line 297.

Line 326ff: delete that section on NO+OH during night-time, see major concern.

Lines 350-353 and equation 4: This equation (="two point fit") was not used, but the slope from all data, see sentence before, which is the correct procedure. Delete the equation.

Line 354: Should be correlation and not covariance.

Line 355 and Figure 6: A plot of HONO against NO2 during night-time makes no

sense as discussed in many former studies (e.g. by J. Stutz's group), as the HONO to NO2 ratio typically increase during night-time (see equation 4 to determine the "NO2 conversion frequency"). Thus typically the higher data points are those from the later night (with higher r.h. ...) while the lower slope data reflect the early night (with lower r.h., see artificial correlation).

Line 359: I do not understand the high value of the HONO/NO2 ratio of 11.3% while a HONO/NOx ratio of 4-5 % was also obtained (see table 1). Since NO is much lower than NO2 (see figure 2), the HONO/NO2 ratio should be only slightly higher than HONO/NOx ratio. Check the numbers for consistency.

Lines 363-364: Reason for the higher conversion frequency is the different time period used. While here only the initial increase of the HONO/NO2 ratio during the early evening was used (later this is decreasing caused by more efficient HONO uptake on dew surfaces in Melpitz) in most other studies the almost entire night-time increase was evaluated, where formation and deposition overlaps leading to lower conversion frequencies.

Lines 379-380: Here again an artificial correlation is studied (see major concerns). HONO and particles are both formed or emitted near to the ground and variation of the BLH causes the correlation. The question ground vs. particles can be only answered if parallel gradient measurements of HONO, NO2/NOx and particle surface area are performed (see discussion in Atmos. Environ, 2003, 37, 2949-2955).

Line 407: Since the data between 17:30 and 22:00 is considered here, any measured BLH between midnight and 7:00 is meaningless?

408-410: Was a LAI of 6 used (than exchange "we add" by "we used"...) or did you add the value of 6 to the LAI of 4-10? Reformulate...

Lines 415-418: Also reformulate: "...the NO2 uptake coefficient... is larger than the reactive surface...". What you mean here is that the S/V of aerosols is much smaller than

the S/V of the ground and thus the heterogeneous formation takes place on ground surfaces. . .

Lines 436-437 and Fig. 7b: Since the HONO/NO2 ratio is time depending (see above) better plot the average first order rate coefficient for NO2 conversion against the inverse of the WS. You also would not plot the ratio product/reactant in a smog chamber experiment against any variable, but the rate coefficient. Since the WS is a marker for the vertical turbulent mixing the observed anti-correlation is a strong argument for the proposed ground source region of HONO.

Line 474: Delete reaction (6) but add the new reaction 3 (photosensitized conversion of NO2, see major concerns).

Lines 500-501: If NO+OH contributes to ca. 50 % to the daytime HONO levels (see lines 496-497 => significant source!) than its sources strength should be some hundred ppt per hour. Check for the low number!

Lines 504-505, hypotheses (1): Why should HONO formation in the morning be caused only by HNO3/NO3- photolysis and not by a more reasonable photosensitized conversion of NO2 (new reaction 3)? In the morning NO2 levels are still high while HNO3 is high only later during the day (cf. Fig. 4)! Please plot the "unknown source" against a) the product (HNO3/NO3- x J(HNO3)) and b) (NO2 x J(NO2)). I expect the latter correlation is much better. . .

Line 507: Should be equation 11.

Equations 13, 14, 15: the last term in the equations is again not correct (the deposition velocity should be exchange by the mean molecular velocity of HONO). And again, the concept of the uptake coefficient does not work for ground surfaces and fast uptake (transport limited uptake at gamma > 10ˆ-5 - 10ˆ-4, depending on Ra and Rb).

Line 530: The used value for J(HNO3) for the photolysis of surface HNO3/NO3- is too low, here "enhancement factors" (ratio surface photolysis/gas phase photolysis, see

e.g. Environ Sci. Technol., 2018, 52, 13738-13746 and references therein) between 7 and 1000 have been proposed in recent studies for this reaction.

Line 546-547: Why are there two values (max/min) of k(emission) for each emission peak?

Lines 556-559: Again I do not understand the units of the integrated emission/deposition. Is that a concentration from a layer of 1 m or of 500 m? The numbers of released HONO molecules would be different by more than two orders of magnitude. . .

Section 4.3.3: If I understand the concept used (equation 17) correctly (?) than the authors take all dew nitrite and mix that after hypothetical evaporation as HONO homogeneously into a layer of variable height. Then they plot the resulting average (homogeneous) layer concentration against the height (see Figure S9). But in this case the resulting gradient does not reflect a real gradient in the atmosphere. If the concentration is calculated e.g. for a lower layer of 20 m, than all nitrite is already consumed and there is nothing left for any higher layers. Thus, the real concentrations would be much lower! But maybe I did not understand that correctly. . .

Line 577: delete phenol, nitrophenol and HCHO, that is not the topic here.

Lines 578-579: While emission of NH3 during evaporation might be reasonable, re-emission of the highly sticky HNO3 is not expected. While HONO evaporates already e.g. at 80 % r.h., where you still find many formal monolayers of adsorbed water, HNO3 will still strongly stick to such humid surfaces. . . In addition, if you have acids (HNO3, H2SO4) and ammonia in the dew water, low volatile ammonium salts will be formed (e.g. ammonium sulphate) which will also not evaporate when the dew water has gone.

Line 619: should be 90 % and not 58 %.

Lines 623-625: If my interpretation of the gradient data is correct (see above) delete that section.

Lines 631-633: Correct for the numbers, see above.

References: Line 677: Brüggemann

Line 682: 107, D22, 8196,...

Line 683: Pätz H.-W.

Line 685: 108, D4, 8247,...

Line 691: 157-160

Line 692: Rössler

Line 695: Andrés-Hernández

I stopped here, there are numerous errors in the reference list.

Table 1: What is the difference between HONO and HNO2 (unify...)? In addition, which HONO data is shown here (LOPAP or MARGA)? Specify, should be the LOPAP data, see intercomparison results.

Figure caption 2: "The gaps were..." You find at least three gaps in the HONO data...

Figure 3: The nitrite MARGA data should be used with caution, because of high interferences by different NO2 reactions, see above. In addition, I would not use the artificial HONO data by the MARGA, see intercomparison results.

Figure 4: add the HONO/NO2 or HONO/NOx ratio.

Supplement:

Table S1: Reaction 2 and 2a are similar only with different complexity. And reaction 2a only works at ppm levels of NO2, otherwise uptake coefficient of N2O4(g) higher than one are necessary...

Add a new photosensitized conversion of NO2 on organic substrates. Here I would use a new number (e.g. 3a), since reaction 2a is a disproportionation reaction (red + ox of

NO2) while in reaction 2b (dark) and in the new photosensitized reaction (e.g. 3b) NO2 is an oxidant.

Also use a new reaction (4) for 2c, also different mechanism.

You may remove reaction (4), (6) and (7), they are definitely unimportant.

Figure S3: delete the 20 in the lower y-axis

Figure S5: specify the unit of the colour code. Is that the particle surface density Sa? Specify in the caption.

Figure S8: Are the two fits (blue lines) used for the data from both days? Why isn't all shown data fitted (than the slopes should be similar)? See also my question above to table 4.

Figure S9: Please integrate HONO over 600 m (unit should be HONO/area). Is that number similar to the dew nitrite surface density used? I expect it is much larger, see above. If yes remove that figure and the corresponding section.

―――――――――――――――――――――

---

## Editor Comment (EC1) · Hang Su (Editor) · 21 Feb 2020

"In her/his major concern 3) intercomparison, the referee #1 would also like to point the authors to another recent paper in which also a MARGA system was intercompared with a LOPAP: Xu et al., AMT.12, 6737-6748, 2019, with very similar results to the other reference mentioned there (strong overestimation of HONO by the MARGA).

---

## Referee Comment (RC2) · Anonymous Referee #2 · 25 Feb 2020

General Comments

An exploration of the role of dew from field measurements has been long overdue. This work builds very well on prior field observations and lab experiments regarding effective HONO sequestration in dew. The Authors present a convincing study of the uptake and release of HONO from their observations and controlled field collections of dew nitrite. The production and loss process for HONO are parameterized to rates for HONO uptake into dew as well as subsequent release, which are then coupled to a box-model, finding that their observed rapid HONO increases on mornings with dew evaporation can be reproduced. The rates for NO2-to-HONO conversion and HONO deposition are compared against prior uptake observations quite well. Overall, the scientific quality of this manuscript makes it an excellent candidate for publication in Atmospheric Chem-

istry and Physics following major revisions to improve the manuscript clarity and data quality.

Major Revisions

1. Typos, phrasing, and writing clarity throughout have major issues in communicating the scientific findings of this work. Several instances where this makes following the discussion nearly impossible are noted in the detailed comments below. In other places, sentences are started with abbreviations or chemical structures. In many instances abbreviations are used first with the full spelling in brackets, when these should be presented the other way around. The entire manuscript should be revisited for clarity of writing by all of the Authors.

In three sections of the results and discussion (Sections 3.3, 4.2.1, and 4.2.3) there is no synthesis of the cases or findings to complete the sections. They have been left incomplete and should be revised.

2. The intercomparison between the MARGA and LOPAP is a very weak component of this manuscript. Detection limits are not given for either technique and cannot be assumed to be the same as from the prior reports cited by the Authors. These need to be determined at each field site from controlled calibrations and careful collection of field blanks. The collection and correction of field blanks from overflowing the MARGA inlet with zero air are not presented. Were they collected and was a correction applied? How were backgrounds in the MARGA determined?

Calibration techniques for each instrument are also not presented and the arguments for the measurement bias being high for the MARGA are incorrect. Prior studies with similar wet denuder systems have shown that the $NO_2$-$SO_2$ and $NO_2$-$H_2O$ corrections very small (VandenBoer et al., 2014) and cannot possibly explain this discrepancy. Further to this, the same work also demonstrates that in high $NH_3$ atmospheres, similar to those observed in this work, that the denuder pH is sufficient alkalinity and buffer capacity to collect the observed HONO quantitatively as nitrite. This prior work also

makes a comparison with a home-made version of the LOPAP with inlets separated vertically by several meters, where intercomparison was only made when both instruments we calibrated with the same sodium nitrite solution. A strong capability to accurately measure HONO by both instruments was demonstrated. Similar attention to measurement quality must be made by the Authors here to improve the quality of their intercomparison.

The results presented in the intercomparison (Section 3.1) are challenging to follow. The Authors mention 'batch denuder' (Line 215) and 'offline batch denuder' (Line 217) but this is not explained clearly anywhere. What are these batch denuders? How were they prepared and why are they relevant to measurements being compared between the MARGA (an online instrument) and the LOPAP?

The Authors conclude their intercomparison to say that a long inlet on the MARGA could explain the higher HONO they are measuring. Is this hypothesized to be from NO2 hydrolysis on the inlet? The mechanism of interference on the inlet is assumed. It must be made clearly. If yes, can all MARGA daytime HONO data below 500 pptv (e.g. two to three hours from every minimum in the afternoons when NO2 conversion to HONO on surfaces is minimized) be used to determine whether a relationship between measured HONO and NO2 due to an inlet effect is likely? It should also be possible to determine whether the magnitude of this effect is really as high as the 58-90% enhancement observed here. The photos of the inlet configurations (Figure S1a-b) do not make it very easy to understand what the sampling flows, line volumes, and therefore residence times, of the sampled gases were in M1 versus M2. The red text on Figure S1b is not possible to read. Was the sample residence time in M1 much smaller than in M2 due to a change in the inlet flow rate?

In Figure 1, the Authors present the findings from their intercomparison and the systematic offset between the two techniques (i.e. the uncertainty in the slope seems small) suggests that the poor comparison is due a calibration or blank-correction issue rather than significant sources of interferences. The plot of the intercomparison regressions does not include the 1-sigma error evaluated in either the slopes or the intercepts and should be added. Looking closely at the measurements in Figure 1, there seem to be a number of observations of much higher HONO by the 5 min LOPAP measurements over the MARGA. Previously, (Sörgel et al., 2011) demonstrated that the LOPAP could sample fog droplets to result in such a positive bias, drawing off of prior work by (Bröske et al., 2003; Kleffmann et al., 2006). Were any fog events observed during the field campaign and was particulate nitrite observed by the MARGA and in the LOPAP HONO channel? Presumably, with so much dew, the meteorological conditions were also favourable for fog formation? This could further support the uptake and deposition of HONO into dew at the surface, as the MARGA nitrite measurements would observed this directly.

3. The reactions from Table S1 are referenced regularly throughout the manuscript and should be moved, either as a table or as separated reactions corresponding to their first presentation throughout the introduction.

4. The Authors suggest that NH3 and HNO3 are released from dew similarly in the morning, as their mixing ratios also increase shortly after sunset. However, the observed particulate NH4NO3 also increases, which is likely more consistent with aerosol aloft being mixed down into the nocturnal boundary layer and repartitioning to release NH3 and HNO3. Volatilization of HNO3 from surfaces containing water does not seem plausible given the strong acid nature of this species. If this were the case, deposition of HNO3 would not be represented as a terminal sink in atmospheric models. While NH3 may be released from dew, this would be challenging to discern from the data presented here. The several instances where this justification is made to bolster the release of HONO from dew should be removed from the manuscript and SI, along with the associated figures. The direct dew observations, model rate parameterizations, and subsequent model-measurement comparison are sufficiently strong to make this case.

Detailed Comments

Line 37: 'were' should be 'are'. This correction is needed frequently throughout the manuscript and should be made where appropriate throughout.

Line 45: 'induced' should be 'activated'. The Authors should also cite the work of (Aubin and Abbatt, 2007) on this topic.

Lines 56-65: This is an extremely long list of reactions that need to be broken down into organized categories. Typically, where lists have entries that contain commas, the list items are separated using ';' so they can be easily distinguished.

Lines 65-67: The potential role of dew releasing HONO was also discussed in (Lammel and Cape, 1996; Lammel and Perner, 1988; VandenBoer et al., 2014).

Lines 81, 88, and 90: Instrument full names should be given first, followed by their abbreviations.

Line 82: 'detect' should be 'detects'

Line 90: should be 'found excellent agreement'

Lines 94-97: Why does an HNO3 artefact matter? Should this be HONO? Given the subsequent lack of clarity in discussing the intercomparison issues below, additional details about the artefact between the MARGA and batch and coated denuders with shorter inlet lines should be very clearly outlined here.

Line 117: Sentences should not start with numbers. Consider rephrasing to 'An inlet flow of . . . '

Lines 127-129: Was LiBr used in both the gas and particulate channels? It seems unlikely that this is easily done in the SJAC. Please clarify. Also, the second sentence should read '. . .both collected over the course. . .'

Lines 136-137: Two reactions for the derivatization of nitrite to the azo dye are mentioned here but not presented anywhere. The reaction notation conflicts with the reaction numbers presented in Table S1. Please clarify and add the reactions, if desired.

Line 139: 'air zero' should be 'zero air'

Lines 153-155: The methodology for cleaning the glass plates and collecting samples is presented, but no blank collections are presented where deionised water applied to the glass plates was measured. This should be included in Table 2 to increase the strength of these findings. It would also be valuable, if the Authors have such data, to present the nitrite recovered from washing these glass plates after nights when no dew formation occurred to compare the deposited quantities. This would strongly support the magnitude of calculated uptake of HONO into the bulk water on these surfaces during dew formation events.

Line 224: Why was PM10 nitrite not measured? If there was fog, or reactive coarse particulate matter as observed by (He et al., 2006; VandenBoer et al., 2015), then you may observe the partitioned HONO directly with the MARGA as in (VandenBoer et al., 2014).

Line 249 onward: The discussion from here on is vague about which measurement is being used for each section. This needs to be clearly denoted. Presumably the LOPAP measurement is being taken as having the best accuracy for measuring HONO, but this needs to be clearly stated.

Line 262: 'concentrated' should be 'concentration'

Lines 264-266: This sentence is unclear and hard to follow. Rephrase.

Line 267: 'during the campaign' is referring to the prior campaigns just cited in the preceding sentence or in this work? Please clarify.

Line 268: 'HONO would be reemitted in the atmosphere', also here 'lead' should be 'led' (there are many instances of this throughout the manuscript that need to be corrected)

Line 276: 'NO2-to-HONO conversion frequency'

[Figure]

Lines 279-284: These are two sets of observations, not cases. Please indicate why these are nicely categorized for further exploration by clarifying their utility. At Line 282 the Authors indicate that HONO increased with wind speed, but usually this corresponds to a decrease in concentration due to dilution. Is this increase related to wind direction? This section then ends without further exploration of the two cases and needs to be completed.

Lines 286-289: These two sentences are misleading and contradict an accurate discussion that follows regarding the pH-dependent effective partitioning of HONO between the gas phase and aqueous solution. Acidic water would readily volatilize HONO. Revise this for consistency with the following discussion. In particular, the work of (He et al., 2006) demonstrated that leaf surface washings were alkaline, which drove favourable HONO partitioning to the surface. Were any grass washings collected here to investigate the effective pH of the vegetated surface? What about the glass plates after exposure over nights where no dew formed? Either of these, but ideally both, would make a stronger case for effective dew uptake of HONO.

Line 295: 'concern' should be 'focuses'

Lines 303-307: This needs to be clarified. I see no temperatures where dew water would be frozen in April. Did this actually occur in May? And why would higher F(NO2-) be observed on frozen dew? As the Authors state, the phase transition would act to inhibit HONO partitioning. This section needs to be revisited and revised for clarity. It is also surprising that this high F(NO2-) is included in the averaging over the other dew F(NO2-) observations. This would deliberately bias the use of this value later. A better approach would be to statistically exclude the outlier using the Grubb's test or, better yet, include all of the observations in the average since the number of dew samples is very limited.

Line 312: End first sentence after 'evaluated'. Start next sentence with 'Generally'

Lines 315-319: If the Authors have no data that meet these criteria, then why bother

listing them. Simply state that a direct emission number could not be quantified here and then apply one that has been widely used, such as the study by (Kurtenbach et al., 2001) cited on Line 314. This can then be applied to correct the HONO formation and loss rates at night, when the correction can be accurately used. This cannot be simply ignored.

Lines 331-334: This is a most unusual exercise. This pathway has been long ignored as it is well known to be negligible.

Line 341: 'cases' should be 'conditions'

Line 362: High HONO/NO2 values were also reported in (VandenBoer et al., 2013).

Line 370: This discussion is incomplete. What chemistry is happening here and why?

Lines 388-392: This needs to be revised. This logic is very hard to follow and is the inverse approach from what is typically presented to make such a comparison between a modeled rate and observations. The uptake coefficient comparison does not seem to follow logically from the prior calculation. The Authors should revisit this. It would be better to show that typical aerosol uptake coefficients used by (Tsai et al., 2018; Wong et al., 2012) fail to produce the observed quantities of HONO and to quantify the fraction that aerosol conversion represents so it is clear that aerosol conversion is trivial.

Line 414: These references are not for ground proxies, but atmospheric aerosols. Remove and add studies that use real soils and soil proxies, such as (Donaldson et al., 2014; VandenBoer et al., 2015). These are both consistent with the observations made in this work and are more representative.

Lines 430-431: The trends discussed are not plotted on Figure 7a and need to be added. It seems unlikely that these trends are very robust. The kinetics of reaction 2 are not well constrained for increasing surface availability of H2O and this sentence should be rephrased carefully.

Line 453: The Authors should expand on why the difference between their observations and those from Boulder are so dramatic. The observations constraining HONO uptake on the ground surface presented in (VandenBoer et al., 2013) are all for nights where dew did not form, but deposition was observed to increase with RH (see equation 3 in Section 3.1 of that paper). Perhaps it is possible to estimate an effective HONO uptake coefficient from the flow tube work of (He et al., 2006) to compare to?

Line 461: Discussion is incomplete (see Major comment on this). Summarize the importance of your findings.

Line 465: 'above...' should be 'to reach an average minimum...'

Line 466: Why is the daytime maximum presented here? It is distracting from the point of this part of the discussion. Remove.

Lines 469-471: See Major comment on NH4NO3 thermodynamic partitioning and mixing. Remove this argument.

Lines 499-501: Fix subscript typo on 'unknown' in the equation. The final sentence is hard to follow and this section does not end very clearly. What is the 'additional source'? P(unknown)? Revise for clarity.

Lines 514-515: The parameterization of boundary layer height used in the box model has not been explained and needs to be added somewhere. Were static or dynamic conditions of boundary layer height used? What measurements were used to set these boundary layer conditions and how are they justified to be suitable for use in this model?

Line 518: Following '12.5' the authors should add 'that we calculated from our observations'

Line 522: 'would be' should be 'was'

Line 526: 'preceded' should be 'made'

Lines 529-531: There is a lot of literature stating that particulate NO3- photolyzes 10-

1000 times faster than gaseous HNO3. How do the Authors justify why this was not included in the model?

Lines 539-540: Was it dry ground or rain that was observed on the night of 23 April? Please specify.

Line 548: What is 'this value'? Is it the average? Or is it one of the maximum or minimum?

Lines 577-579: Remove NH4NO3 discussion.

Line 581: 'consistently' should be 'previously'

Line 583: 'these' should be 'our'

Line 613: 'are rich of ground surface (forest and grass)' is inaccurate. Consider rephrasing to 'experience frequent dew formation'. The type of surface likely does not matter much when bulk water is available. If this were the case, the glass dew collectors would not have done a representative job of collecting the dew composition.

Line 616: Conclusions and Atmospheric Implications: Rewrite fully to reflect manuscript changes made.

Table 1: Clearly indicate the MARGA measurements.

Table 2: The methods are unclear whether the pH of the dew was measured on a subsample of the total volume. Add some details on this to the caption and to the method section. It could be possible that direct measurement of the dew sample could lead to ion contamination from the salt bridge of the pH meter being in contact with the sample. Add blank sample values of collected nitrite here (i.e. deionised water passed over clean glass plates and trough into sampling vessel) for comparison. It would also be useful to see what nitrite deposition occurs to these surfaces at night in the absence of dew, so as to contrast the magnitude of change that the presence of dew makes.

Figure 1: The panel order in this figure is unusual. Typically, panels are lettered from

a, starting at the top, with the letters located outside of the axes. The top panel intercomparison is not easy to read. The slopes should be positive on the plot, with HONO_LOPAP (pptv) on the bottom axis, not the top.

Figure 3: Change 'HONO_MARGA' to 'HONO'

Figure 4: Remove panels e) and f)

Figure 5: Are these measurements from the LOPAP? How was the error determined? This was not presented in the methods and should be added.

Figure 6: This is not very convincing as there is a lot of overlapping data. Maybe create RH bins for each 20 % increment. Each bin would be centred at the average HONO/NO2 with x and y error bars corresponding to standard deviations of HONO and NO2 of all data collected in that RH bin (e.g. 0-20 %)?

Figure 7: Lettered labels on the panels are different from others above. Please make these consistent across all figures. It is very hard to take anything away from 7b. Consider moving to the SI. Trend lines need to be added to 7a to address comments above.

Figure 8: I would like to see a third panel here that depicts the OH radical concentration, measured HONO, and HONO from Model 5 that includes upper and lower limits on the model output according to the minimum and maximum k(emission) rates observed. This would provide a better comparison to the range of observations and would likely package the argument of dew partitioning even better.

Again, move the panel letters outside of the axes and keep them consistent with prior figure formatting.

Figure S1: Why are both sets of photos separately labeled? Add schematics to depict tubing diameters, lengths, and flows for each sampling configuration. The schematic needs to be consistent with the requested revisions to the sampling methods section. Fix caption to be accurate.

Figure S2: What is the trough material made of?

Figure S8: This should be in the main manuscript. It is a great figure for this paper.

Figure S9: Should 'evolution' be 'structure'? Evolution implies time dependence which is not what is described in the text of the manuscript.

References

Aubin, D. G. and Abbatt, J. P. D.: Interaction of NO2 with hydrocarbon soot: Focus on HONO yield, surface modification, and mechanism, J. Phys. Chem. A, 111(28), 6263–6273, doi:10.1021/jp068884h, 2007.

Bröske, R., Kleffmann, J. and Wiesen, P.: Heterogeneous conversion of NO2 on secondary organic aerosol surfaces: A possible source of nitrous acid (HONO) in the atmosphere?, Atmos. Chem. Phys., 3(3), 469–474, doi:10.5194/acp-3-469-2003, 2003.

Donaldson, M. A., Berke, A. E. and Raff, J. D.: Uptake of gas phase nitrous acid onto boundary layer soil surfaces, Environ. Sci. Technol., 48(1), 375–383, doi:10.1021/es404156a, 2014.

He, Y., Zhou, X., Hou, J., Gao, H. and Bertman, S. B.: Importance of dew in controlling the air-surface exchange of HONO in rural forested environments, Geophys. Res. Lett., 33(2), 2–5, doi:10.1029/2005GL024348, 2006.

Kleffmann, J., Lörzer, J. C., Wiesen, P., Kern, C., Trick, S., Volkamer, R., Rodenas, M. and Wirtz, K.: Intercomparison of the DOAS and LOPAP techniques for the detection of nitrous acid (HONO), Atmos. Environ., 40(20), 3640–3652, doi:10.1016/j.atmosenv.2006.03.027, 2006.

Kurtenbach, R., Becker, K. H., Gomes, J. A. G., Kleffmann, J., Lörzer, J. C., Spittler, M., Wiesen, P., Ackermann, R., Geyer, A. and Platt, U.: Investigations of emissions and heterogeneous formation of HONO in a road traffic tunnel, Atmos. Environ., 35(20), 3385–3394, doi:10.1016/S1352-2310(01)00138-8, 2001.

[Figure]

Lammel, G. and Cape, J. N.: Nitrous acid and nitrite in the atmosphere, Chem. Soc. Rev., 25(5), 361–369, doi:10.1039/cs9962500361, 1996.

Lammel, G. and Perner, D.: The atmospheric aerosol as a source of nitrous acid in the polluted atmosphere, J. Aerosol Sci., 19(7), 1199–1202, doi:10.1016/0021-8502(88)90135-8, 1988.

Sörgel, M., Trebs, I., Serafimovich, A., Moravek, A., Held, A. and Zetzsch, C.: Simultaneous HONO measurements in and above a forest canopy: Influence of turbulent exchange on mixing ratio differences, Atmos. Chem. Phys., 11(2), 841–855, doi:10.5194/acp-11-841-2011, 2011.

Tsai, C., Spolaor, M., Fedele Colosimo, S., Pikelnaya, O., Cheung, R., Williams, E., Gilman, J. B., Lerner, B. M., Zamora, R. J., Warneke, C., Roberts, J. M., Ahmadov, R., De Gouw, J., Bates, T., Quinn, P. K. and Stutz, J.: Nitrous acid formation in a snow-free wintertime polluted rural area, Atmos. Chem. Phys., 18(3), 1977–1996, doi:10.5194/acp-18-1977-2018, 2018.

VandenBoer, T. C., Brown, S. S., Murphy, J. G., Keene, W. C., Young, C. J., Pszenny, A. A. P., Kim, S., Warneke, C., De Gouw, J. A., Maben, J. R., Wagner, N. L., Riedel, T. P., Thornton, J. A., Wolfe, D. E., Dubé, W. P., Öztürk, F., Brock, C. A., Grossberg, N., Lefer, B., Lerner, B., Middlebrook, A. M. and Roberts, J. M.: Understanding the role of the ground surface in HONO vertical structure: High resolution vertical profiles during NACHTT-11, J. Geophys. Res. Atmos., 118(17), doi:10.1002/jgrd.50721, 2013.

VandenBoer, T. C., Markovic, M. Z., Sanders, J. E., Ren, X., Pusede, S. E., Browne, E. C., Cohen, R. C., Zhang, L., Thomas, J., Brune, W. H. and Murphy, J. G.: Evidence for a nitrous acid (HONO) reservoir at the ground surface in Bakersfield, CA, during CalNex 2010, J. Geophys. Res. Atmos., 119, 1–14, doi:10.1002/2013JD020971, 2014.

VandenBoer, T. C., Young, C. J., Talukdar, R. K., Markovic, M. Z., Brown, S. S., Roberts, J. M. and Murphy, J. G.: Nocturnal loss and daytime source of nitrous acid through

reactive uptake and displacement, Nat. Geosci., 8(1), 55–60, doi:10.1038/ngeo2298, 2015.

Wong, K. W., Tsai, C., Lefer, B., Haman, C., Grossberg, N., Brune, W. H., Ren, X., Luke, W. and Stutz, J.: Daytime HONO vertical gradients during SHARP 2009 in Houston, TX, Atmos. Chem. Phys., 12(2), 635–652, doi:10.5194/acp-12-635-2012, 2012.

————————————————————

---

## Author Comment (AC1) · 7 May 2020

In the manuscript by Y. Ren et al. HONO was measured at the rural station Melpitz in Germany by two different commercial instruments, which were intercompared showing strong interferences and inlet artefacts for one of the instruments. In addition, the measurement data including dew water analysis were used to demonstrate that HONO

15  deposition during night-time and re-emission during the early morning when the relative humidity decrease are important processes in good agreement with former studies. In addition, HONO formation during night- and daytime is discussed and the contribution of HONO photolysis to the daytime formation of OH radicals is compared with the typically proposed main source of OH radicals by $O_3$ photolysis, showing a similar contribution of both sources.

20  The study contains some interesting information and may be considered for publication in ACP, after significant concerns have been considered.

Major Concerns:

1) Chemical reactions:

I found the manuscript difficult to read, since all discussed chemical reactions are only

25  summarized in the supplement. At least the important reactions should be shown in the main text.

*Response:* All the discussed reactions were now moved to the main text.

2) Dew and gas measurements:

30  In the experimental section, gas phase und dew measurements are explained. However, these measurements were not done in a single field campaign, but the dew measurements were performed more than one year later after the gas phase measurements. Later the average dew nitrite data is used to explain the morning peaks of HONO observed one year earlier. This method will cause large uncertainties, since the gas and dew concentrations, but also other

35    parameters may significantly vary from year to year (apples and oranges: : :). E.g. while the temperature was well above freezing during the gas phase campaign (see Fig. 2), dew water was freezing during the later dew campaign (see line 304). However, if water is freezing, oxidation of nitrite is significantly accelerated (see e.g. Nature, 358, 1992, 736-738). Here parallel gas phase and dew measurements are clearly necessary in the absence of frozen dew

40    water.

*Response:* As the reviewer mentioned, our gas phase measurement was mainly conducted in April 2018. We agree with the reviewer that large uncertainties may occur by comparing the dew measurements of 2019 with the intercomparison period of 2018. However, it was important for our study to get an idea how many HONO is dissolved in dew and to estimate

45    how much of this evaporated HONO can explain the observed morning peak. To achieve nearly identical conditions (nearly the same temperature and global radiation), we performed the dew measurements approximately one year later. As example, the dew experiments were realized in the same temperature range as it was the case for the second week in our intercomparison campaign of 2018. The gas phase HONO measurement in May 2019 was

50    conducted by MARGA and results are shown in Figure R1. As shown in Figure R1, HONO morning peak was also found in the day of May 8$^{th}$, May 13$^{th}$ and May 14$^{th}$ 2019, although frozen dew water was found for the day of May 8$^{th}$ and May 14$^{th}$ 2019. We agree with the reviewer that the present results give only an imagination how strong the dew evaporation source to atmospheric HONO is. More exact measurements have to be performed to clearly

55    quantify the role of dew evaporation on found atmospheric HONO concentrations. This has to be done in future studies.

Additionally, we cannot exclude that frozen dew was also present in the intercomparison campaign 2018 as we observed in 2019 that frozen dew was also formed for air temperatures above 0 ℃.

60    The reviewer is right. We observed frozen dew on May 8$^{th}$, May 13$^{th}$ and May 14$^{th}$. We also analyzed the defreezed dew samples but they were not used for further analyses because of the interference with an enhanced oxidation. As mentioned in the text (Line 342-348), the dew water of May 11$^{th}$ 2019 was not frozen and the obtained $F_{NO2-}$ (NO$_2^-$ concentration per m$^2$ of the sampler surface) was used to discuss the HONO morning distribution.

[Figure]

65

**Figure R1**. Time series of HONO (MARGA measurement) in Melpitz from May 8[th] to May 15[th] 2019. The gap was mainly due to HONO quantification of dew samples and the maintenance of the instrument.

70    3) Intercomparison:

The intercomparison results should be clearer discussed. Since all known chemical interferences are positive interferences (overestimation of the HONO data) and since I expect that both groups can calibrate their instruments with high accuracy, the results shown in Fig. 1 are quite clear. First, the MARGA instrument overestimates HONO during this field

75    campaign at least (the LOPAP instrument may also have interferences: : :) by ca. 90 % (see data M2, where both instruments are operated in their normal way), and not by 58 % as mentioned in the conclusion (line 619). Second, these _90 % are caused by ca. _60 % chemical interferences inside the MARGA instrument (see data M1, where both instruments used the common MARGA inlet), e.g. by oxidation of $SO_2$ (see e.g. Spindler et al., 2003) or

80    VOCs by $NO_2$, which are corrected for by the LOPAP instrument. In addition, ca. 30 % of the HONO MARGA data results from heterogeneous formation of HONO in the inlet of this instrument (PM10 inlet + Teflon line), see the difference between the slopes M2 and M1. So one important conclusion is that MARGA HONO field data should not be used.

This result is in excellent agreement with a former intercomparison of both instrument types

85    in a Chinese field campaign (see J. Geophys. Res., 2010, 115, D07303, doi: 10.1029/2009JD012714) where also a RWAD instrument (similar to the MARGA) overestimated HONO by a factor of three on average. In this context, the statement in line 104 ("first inter-comparison: : :") is not correct. In addition, the results also show that the use of massive sampling inlets − even if they are coated by Teflon − should not be used for any

90    in-situ HONO instrument.

*Response:* More discussion has been added for the inter-comparison of MARGA and LOPAP as below in Line 252-262 "The evaporation of dissolved HONO from the off-line sample and heterogeneous reactions of $NO_2$ and $H_2O$ as well as $NO_2$ and $SO_2$ in water as described by Spindler et al. (2003) or VOCs by $NO_2$ could explain the artefacts in the denuder solution

95 (Kleffmann and Wiesen, 2008), which could account for ca. 58% (M1, where both LOPAP and MARGA used the common MARGA inlet) of these ca. 90% of overestimated HONO measurement from the MARGA. Additional artefacts as heterogeneous formation of HONO due to the long MARGA inlet system should be responsible for another ca. 32% (the difference between slopes M2 and M1). Hence, the results show that the use of massive

100 sampling inlets, even if they are coated by Teflon, should be avoided for any in-situ HONO instrument. As a result, we chose the LOPAP-measured HONO in the following sections because of its high precision."

Line 682, in the conclusion, the overestimation of the HONO data from MARGA than LOPAP was changed as ca. 90 %.

105 We apologize for this mistake and included the cited study in our manuscript as comparison in Line 237-240. We changed our statement in line 115 as "Our observations provide a direct inter-comparison between LOPAP and MARGA for HONO field measurement".

**4 (1) OH data:**

110 For the discussion of the HONO sources the OH data is necessary (see e.g. reaction 3), which was calculated here by a simple linear correlation with the global radiation. However, this method is highly uncertain since first, short wave UV radiation should at least be used (see similar studies using J(O1D)...), cf. main sources of OH-radicals. Here the ratio between the global radiation and J(O1D) will show strong diurnal and seasonal variability (depends

115 mainly on the SZA: : :). In addition, while the correlation between J(O1D) and OH is indeed often linear, the slope is highly variable and will e.g. depend on the VOC/NOx ratio. Thus, I expect easily a factor of two uncertainties in the calculated OH concentration. Since e.g. half of the HONO daytime levels could be explained by the gas phase reaction (3), see lines 496-497, a factor of two higher OH level could make all discussions about any "unknown

120 HONO sources" obsolete.

*Response:* As mentioned by the reviewer, [OH] showed a close relationship with the UV solar flux (Rohrer et al. 2006) as generally expected. In addition, the UV solar flux is closely correlated with global solar irradiance (Boy and Kulmala, 2002). On the basis of such a correlation, Größ et al., 2018 devised the linear function between global radiation flux (0.3 - 3

125 μm) measured by a pyranometer and [OH] measured by CIMS for EUCAARI 2008 at Melpitz, the same atmospheric research station as conducted for the present work. Hence, [OH] in the present work was estimated by using this linear function since we did not apply a direct OH measurement, and [OH] could have a factor of two uncertainties. However, regarding on the large uncertainty of [OH] but also large variability of HONO concentration, the "unknown

130 HONO sources" could be not crucial but they could also exist according to the observation of Figure 6. Then the discussion about "unknown HONO sources" has been improved in line 543-550 as below:

"Reaction 3a can continually contribute 50% of the measured HONO from 10:30 to 16:30 (UTC). However, regarding on the large uncertainty of [OH] but also large variability of

135 HONO mixing ratio, the "unknown HONO sources" could be not crucial but it could exist due to the observation of Figure 6. Basically, the additional HONO contribution rate could be estimated from following equation:

$$P_{unknow} = \frac{d[HONO]}{dt} + J_{HONO}\,[HONO] + k_9[OH][HONO] - k_{3a}[OH][NO] \qquad \text{(Eq. 10)}$$

However, a quite low additional source of $91\pm41$ pptv h$^{-1}$ was derived beside OH reaction

140 with NO…"

**4 (2) In addition, how have the authors calculated the night-time OH levels by this method (ca. 10^4 cm^-3, see lines 329-330)? During night-time there is no radiation and calculated OH should be zero: : : Normal OH night-time levels decrease from ca. 10^6 cm^-3 in the early night to 10^5 cm^-3 in the later night caused by night-time sources of radicals (O3+alkenes,**

145 NO3+alkenes: : :). This data and the corresponding sections should be removed.
*Response:* We agree to the statement of reviewer on the night-time OH radical, and accordingly the night-time OH concentration and corresponding section has been removed in Page 11.

150 5) Correlation analysis:

In several sections throughout the manuscript correlations were used to identify source processes, which is highly uncertain and which often leads to wrong conclusions. Already in the early 1990 high correlations of Radon with HONO were observed, which are simply caused by the variation of the BLH/vertical mixing for two ground surface sources. Nobody

155 would conclude that Radon is a precursor of HONO. However, for correlations of HONO with different parameters exactly this is done (not only in the present study: : :). E.g. in lines 356-358 correlation of HONO with relative humidity is explained by the heterogeneous

reaction 2NO2+H2O, reaction (2). Besides that this reaction is far too slow to explain the night-time formation (gamma for R2 ca. 10ˆ-7 –10ˆ-8, one to two orders of magnitude faster kinetics is necessary, see e.g. line 412), the correlation of HONO with humidity may be artificial! During night-time the ground surface is cooling which leads to a) increasing relative humidity, b) decreasing vertical mixing c) increasing surface to volume ratio of the lower nocturnal boundary layer d) increasing rate for any heterogeneous reactions (which scale with S/V: : :) and increasing levels of ground emitted species, like e.g the proposed HONO-precursor NO2 or the particle surface area of freshly emitted particles. All these changes lead to artificial correlations (e.g. HONO with r.h., with particles,: : :) from which one should not necessarily conclude source processes. All the correlation analysis should be much more carefully discussed and results should be checked for plausibility, see below.

*Response:* As mentioned by the reviewer, the correlation analysis is a general method used in the literature (Su et al., 2008; Kukui et al., 2014; Michoud et al., 2014; …) which can provide a first insight of relationships between different species. In the present study, the correlation of HONO with $NO_2$ (previous Figure 6, has been removed) and the correlation of HONO with particle (Figure S6) have been analysed. As shown in Figure S4 and described in the text, the nighttime HONO formation may cause by Reaction 2 (heterogeneous conversion of $NO_2$ to HONO). This resulting gamma ($\gamma_{NO2\rightarrow HONO\_g}$) for Reaction 2 varied from $2.4\times10^{-7}$ to $3.5\times10^{-6}$ with a mean value of $2.3\pm1.9\times10^{-6}$ in line 439 but not ca. $10^{-7}$ - $10^{-8}$ of $\gamma$ for R2 mentioned above. This gamma value is in good agreement with literatures (Kurtenbach et al., 2001;Kleffmann et al., 1998;VandenBoer et al., 2013). The RH dependence of HONO formation has been suggested by numerous lab studies (Finlayson-Pitts, 2009; Finlayson-Pitts et al., 2003; Miller et al., 2009; Ramazan et al., 2006) and also in the field (Stutz et al., 2004). Exactly, Stutz et al. (2004) found a likelihood of increased HONO/$NO_2$ at high RH, in particular, suggesting that HONO formation from heterogeneous conversion of $NO_2$ was often enhanced at RH above 60% in the field by taking into account of some parameters e.g. S/V ratio. However, regarding the weak correlation between HONO and RH in the previous Figure 6 (now removed) and as discussed by the reviewer, we would like to remove the description of previous Figure 6 and discussion of the HONO formation dependence on RH in the section 4.2.1.

Other correlation analyses, e.g. the HONO/NO2 with particle surface density was also checked and is discussed in the following parts of this response letter.

6) Heterogeneous kinetics

*Response:* In present work, we would like to derive the uptake coefficient of $NO_2$ and HONO using Eq.3, Eq.4 and Eq.7. Here we improved our calculation by following the reviewer's

comments and suggestion, and also a simple resistance model is now applied.

**#6 (1)** While equation (5) for the calculation of the heterogeneous conversion of NO2 to HONO is correct for small uptake coefficients, at least when a 100 % HONO yield is assumed (not explained here; only valid for fresh soot, a minor constituent of particles: : :), the calculated uptake coefficients (ca. 10ˆ-15: : :) to explain the missing night-time formation of HONO on particles (as a limiting case) are completely unreasonable! As the authors later correctly mention, the S/V ratio of particles is typically orders of magnitude lower than for ground surfaces. Thus, a higher (!) gamma is necessary for particles compared to the ground. Typically, formation of HONO by NO2 conversion on particles can be only explained, if gamma values in the range 10ˆ-3 to 10ˆ-4 are used. Otherwise this low number would mean that HONO formation could be easily explained by a reasonable uptake kinetics on particles (ca. 10ˆ-6; see lab studies on several heterogeneous NO2 reactions: : :)!? Here the authors should check their calculation – I expect some large order of magnitude errors, e.g. by using a wrong unit of the S/V ratio.

*Response:* To calculate the uptake coefficient of $NO_2$ on the particle surface ($\gamma_{NO2 \rightarrow HONO\_a}$) in the present study, we assumed that the entire HONO formation was taking place on the particle surface (Line 412-414). Particle surface density $S_a$ was calculated from the particle size distribution of APSS and D-MPSS data as shown in Figure S5 and ranged from $9 \times 10^{-4}$ to $9 \times 10^{-3}$ $m^2$ $m^{-3}$. In addition, a hygroscopic factor $f$(RH) following the method of Li et al. (2012) was applied to correct $S_a$ to the aerosol surface density in the real atmosphere. However, the calculation of $f$(RH) (Line 404) was wrong in our previous version and lead to a very low $\gamma_{NO2 \rightarrow HONO\_a}$. We excuse for this error and thank the reviewer for his insight leading to the correction of this error. Finally, an uptake coefficient $\gamma_{NO2 \rightarrow HONO\_a}$ of $(8.8 \pm 5.0) \times 10^{-6}$ ranged from $1.5 \times 10^{-6}$ to $1.9 \times 10^{-5}$ was obtained and, correspondingly, corrected in the text (line 413).

**#6 (2)** Besides this, the use of an uptake coefficient is not recommended when a ground surface conversion is considered (see equation 6), at least for large geometric uptake coefficients and for low night-time vertical mixing (see present study). If a leave area index of 10 is used (see line 409) the obtained "true uptake coefficient" of ca. 10ˆ-5 converts into a "geometric uptake coefficient" of ca. 10ˆ-4. For such high values the transport gets rate limiting for a stable night-time atmosphere!

In this case better a flux concept including resistances for convective mixing and molecular diffusion ($R_a$ and $R_b$) and a surface resistance ($R_c$) should be used. Only $R_c$ can be converted into an uptake coefficient and vice versa. The inverse of all resistances leads to the deposition velocity from which a surface uptake flux can be derived by multiplying with the concentration. In many cases this deposition velocity is only depending on the transport resistances, which depend e.g. on the wind speed ($R_a$ and $R_b$ can be estimated by

230   parameterizations, see e.g. VDI 3782). Using constant uptake kinetics makes no sense here.

*Response:* We appreciate this comment of the reviewer and followed his suggestion to use a simple resistance model according to the description of Seinfeld and Pandis (2006) which had been proposed by Huff and Abbatt (2002). This part has been then been added to the SI as follows:

235   "**Investigating resistance limitations in transport of HONO and NO$_2$ to the ground surface during the Melpitz measurement**

In order to assess limitations of NO$_2$ conversion and HONO deposition in the surface parameterizations derived for the Melpitz dataset, a simple resistance model according to the description provided by Seinfeld and Pandis (2006), which has been proposed by Huff and Abbatt

240   (2002) (Equation S1) was set up.

$$v_d = \frac{1}{R_a + R_b + R_c} \tag{S1}$$

Here, $v_d$ is the observed deposition velocity (cm s$^{-1}$), $R_a$ is the aerodynamic transport resistance (Equation S2), $R_b$ is the molecular diffusion resistance (Equation S3) and $R_c$ is the reactive loss resistance (Equation S4). Each term can be calculated as follows

245   $$R_a = \left(\frac{1}{\mu \, \kappa^2}\right)\left[\ln\left(\frac{z}{z_0}\right)\right]^2 \tag{S2}$$

$$R_b = \frac{z_0}{D} \tag{S3}$$

$$R_c = \frac{4}{\gamma \, c} \tag{S4}$$

Where $\kappa$ is the von Kármán constant (0.4) (VandenBoer et al., 2013), $\mu$ is the wind speed as 0.1 - 6 m s$^{-1}$ in the pesent study, $z_0$ is an estimate of the roughness length of the surface (~ 0.03 m

250   according to a 0.3 m grass height), $z$ represents the surface layer height and is set to 15 m as example for nighttime values in Melpitz. Values for the local surface roughness length and surface layer height were approximated for atmospheric conditions with wind speeds less than 6 m s$^{-1}$ (Huff and Abbatt, 2002;Seinfeld and Pandis, 2006). D is the molecular diffusivity of HONO and NO$_2$ as 7.2 $\times 10^{-5}$ and 1.5 $\times 10^{-5}$ m$^2$ s$^{-1}$, respectively, at 760 Torr (Hirokawa et al., 2008;Langenberg

255   et al., 2019), $\gamma$ is the reactive uptake coefficient and c is the mean molecular speed (~ 367 m s$^{-1}$ for HONO and NO$_2$). These values were derived assuming that the upper limit to the observed HONO reactive uptake was limited equally by molecular diffusion and aerodynamic transport (Equation S1).

[Figure]

260

**Figure S9**. Estimated contributions of resistance parameters to the observable ground surface processes for the HONO and $NO_2$ uptake values derived from Melpitz station. A series of grey shaded regions define the borders of the reactive uptake resistance ($R_c$), the $R_c$ values calculated from upper and lower limit uptake values of HONO and $NO_2$ in this work are shown in green and

265    pink column, respectively. The aerodynamic transport resistance ($R_a$, red line) and diffusion resistance ($R_b$, blue line for HONO and yellow for $NO_2$) are shown in the Figure."

Figure S9 shows the results of the calculated resistances using data limitations from the observation data set and compared to the observed range for HONO and $NO_2$ in Melpitz. This

270    result presents that the aerodynamic transport resistance increases with decreasing windspeed and could play the main role for the HONO deposition when the wind speed was less than 0.5 m s$^{-1}$. Regarding on the calculated $R_c$ range (region indicated by green bar) using the reactive uptake values observed for HONO ($1.7 \times 10^{-5}$ to $2.8 \times 10^{-4}$), limitation of the observed uptake of HONO was potentially significant from the molecular diffusion resistance term in the data range at wind

275    speeds larger than ~1m s$^{-1}$. Therefore, the range of HONO uptake coeffcient values calculated in this investigation are potentially limited by a combination of both transport and diffusion to the ground surface. Since such limitations are realistic for the atmosphere, the γ-coefficients calculated here could have a broad scale applicability used for simulation of HONO production and loss at night when constrained by the observations. As shown in Figure S9, the $R_c$ range

280    (region indicated by pink bar) calculated based on the reactive uptake values observed for $NO_2$ ($2.4 \times 10^{-7}$ to $3.5 \times 10^{-6}$) indicate limitation by the reactive uptake process, which may play the main role rather than aerodynamic transport limitations and molecular diffusion limitations.

**#6 (3)** Also the factor 1/8 in equation 6 (compared to the 1/4 in equation (5)) should be explained, where the authors obviously propose the (too slow, see above) reaction of 2 NO2+H2O (R2) as the main HONO source, for which a formal HONO yield of 50 % is used. In contrast, several former gradient studies (e.g. J. Geophys. Res., 2002, 107 (D22), 8192, and Atmos. Chem. Phys, 2017, 17, 6907-6923, doi: 10.5194/acp-2016-1030) found experimental HONO yields from the NO2 uptake on the ground in the very low % range (2-4%...), i.e. only ca. 3% of the deposited NO2 is converted into HONO on the ground during night-time and not 50 %: : :

*Response:* While night time concentrations of HONO can be reasonably explained by the heterogeneous conversion of $NO_2$ surfaces on humid surface (Kleffmann, 2007), Eq. 4 (previously Eq.6) as $k_{het} = \frac{1}{8}\gamma_{NO2\rightarrow HONO\_g} \times \upsilon_{NO2} \times \frac{S_g}{V}$ was used in present study to derive the uptake coefficient of $NO_2$ on the ground surface as suggested in the literatures (Li et al., 2010;Kurtenbach et al., 2001;VandenBoer et al., 2013;VandenBoer et al., 2014). As defined in Reaction 2, $2NO_2 + H_2O \rightarrow HONO + HNO_3$, two deposited $NO_2$ would form 1 molecule of HONO. Hence a formal yield of 50% (1/8) was used in this equation. In addition, certain field studies also found relatively high $NO_2$-to-HONO ground conversion ratio (Lu et al., 2018;Yu et al., 2009;Li et al., 2012;Su et al., 2008) from ca. 10% to ca. 34% (5.5-16.3% obtained in this work) and even higher. However, precisely lab work should be conducted to observe a more reliable $NO_2$-to-HONO ground conversion ratio.

**#6 (4)** Furthermore the HONO deposition on the ground and the derived necessary uptake coefficients (see section 4.2.3) are impossible! Besides the same argument as for the NO2 uptake on ground surfaces (see above, the use of uptake coefficients for fast ground uptake makes no sense, better use the flux concept and a variable deposition velocity: : :), the values for the HONO uptake coefficient in the range 5.6-19.5 are impossible!? When I was reading the abstract (line 25), I first expected that the authors simply missed the order of magnitude after the given numbers (e.g. x10ˆ-6: : :). Even if one considers a LAI of 6 (see line 410, for the HONO uptake this is not specified: : :?) the maximum calculated value could be only 6 but never 19.5, which would imply a real uptake coefficient using the true surface area larger the unity. Please check for the definition of the uptake coefficient, e.g. by IUPAC, with a maximum value of one. Reason for the order of magnitude errors is equation (9), where the concept of the deposition velocity is mixed with the concept of the uptake coefficient. To calculate L(HONO) (=dc/dt) a first order rate coefficient k (s-1) is multiplied by the concentration (dc/dt = $-$ k $\times$c). The first order rate coefficient for a heterogeneous reaction is calculated from the uptake coefficient by:

k= 1/4 x gamma x average molecular velocity of HONO x S/V.

S/V can be exchanged by 1/H, as done by the authors. So instead of using the deposition velocity (3.35 cm/s, line 450) in equation (9) the mean molecular velocity of HONO (ca. 3.7x10ˆ4 cm/s) should be used, leading to four orders of magnitude lower values: : : And for these high values (>10ˆ-4) the HONO uptake is definitely transport limited, see above.

*Response:* For the calculation of HONO uptake coefficient, the HONO deposition velocity of 3.35 cm s$^{-1}$ was used and the HONO uptake coefficient ranged from 5.6 to 19.5 was obtained and we are sorry for this mistake. As suggested by the reviewer, the mean molecular velocity of HONO as $3.67 \times 10^{4}$ cm s$^{-1}$ was used in Eq. 7:

$$L_{HONO} = \frac{1}{4}\gamma_{HONO,ground} \times [HONO] \times \frac{\upsilon_{HONO,ground}}{H} \tag{Eq. 7}$$

In addition, the value of *H* was calculated from the backward trajectory based on GDAS data and ranged between 20 m and 300 m from 22:00 until around 04:00 UTC in April 2018 during the present study. Then we obtained a value ranged from $1.7 \times 10^{-5}$ to $2.8 \times 10^{-4}$ for HONO uptake coefficient (Line 481-482).

**#6 (5)** And finally, I do not understand the concept used for the quantification of the deposited HONO, which is later compared to the dew water nitrite. Here the total deposited HONO is given in the unit ppt (see e.g. line 428) but should be given at the end in molecules/cm2 to compare that with the dew nitrite (similar unit). To what S/V or boundary layer height does that "deposited mixing ratio" relate? Do the authors expect that the concentration change for the deposited HONO (dc/dt) is constant in whole boundary layer? Here a surface density (molecules/cmˆ2) should be derived by integrating the product of the variable (turbulence depending, see above) deposition velocity with the concentration.

*Response:* We are sorry for the mistake. Exactly, the total deposited HONO (in pptv) on the ground surface was assumed same as the total night-time HONO loss of 970±730 pptv (6 h), calculated by integrating L$_{HONO}$ from 22:00 to 4:00 (UTC) from the nighttime measurement. Hence, we removed this description in section 4.2.3. However, the HONO deposition flux (F in molecule cm$^{-2}$) was not proposed in this study since the calculation of F = v$_d$*C is also a function of surface layer height (z) and must be related to a reference height at which C is specified. In the present study we just obtained the HONO concentration at the sampling point (3.5 meters from the ground).

Finally, the description and discussion about the uptake coefficient in the text was improved Line 495-510 in the manuscript – this text now reads: "A simple resistance model based on the concept of aerodynamic transport, molecular diffusion and uptake at the surface

(presented in SI) as proposed by Huff and Abbatt (2002) was used to evaluate the factor(s) controlling the potential applicability of the γ-coefficients calculated here for the uptake of $NO_2$ and deposition of HONO. As shown in Figure S9, the deposition loss of HONO is potentially limited by a combination of aerodynamic transport, molecular diffusion and reaction processes. However, the HONO uptake will be transport-limited if the real uptake coefficients are $\geq 2.8\times10^{-4}$ and wind speed was less than 0.5 m s$^{-1}$. In addition, molecular diffusion could play an important role for HONO uptake on the surface, especially when the winds speed is larger than ~1 m s$^{-1}$. Regarding the uptake of $NO_2$ on the ground surface, the range of $NO_2$ uptake coefficients as $2.4\times10^{-7}$ to $3.5\times10^{-6}$ obtained in the present work indicates limitation only by the reactive uptake process. The consistency between our findings and the values of these parameters in models (Wong et al., 2011;Zhang et al., 2016) suggest that the broad scale applicability of these field-derived terms for surface conversion of $NO_2$ should therefore be possible. However, those value of γ found for HONO ($\gamma_{HONO, ground}$=$1.7\times10^{-5}$ to $2.8\times10^{-4}$) require further exploration from various field environments and controlled lab studies."

**Specific Concerns:**

The following concerns are listed in the order how they appear in the manuscript.

Lines 22-23: Specify the deposited HONO in the same unit as the dew nitrite surface density (e.g. molecules /cm^2).

*Response:* Here we would like to present a measured nighttime loss of HONO with strength sink of $L_{HONO}$=0.16±0.12 ppbv h$^{-1}$. Hence, the "nighttime ground surface deposition" was changed to "nighttime loss" in Line 21.

Line 25: correct the gamma values (see main concerns).

*Response:* The ground uptake coefficients for HONO and $NO_2$ were corrected as followed: $\gamma_{NO2\rightarrow HONO}$ = $2.4\times10^{-7}$ to $3.5\times10^{-6}$, $\gamma_{HONO,ground}$ = $1.7\times10^{-5}$ to $2.8\times10^{-4}$ in line 25.

Line 47: The paper by Gutzwiller et al. is not on the NO2+soot reaction but on the reaction of NO2 with semi-volatile hydrocarbons (see line 49). Better use the study by Arens et al. from the same group or the first studies from 1998 by Ammann et al., or Gerecke et al.

*Response:* Line 48-49, the references of Ammann et al., 1998;Arens et al., 2001;Gerecke et al., 1998 were used to replace the paper of Gutzwiller et al. 2002.

Line 50 and table S1: The authors should distinguish between the oxidation of phenols etc. in the dark (reaction 2b) and the photosensitized conversion of NO2 (see Stemmler et al.) by adding a new reaction for the daytime HONO formation (e.g. new reaction 3).

*Response:* Line 69, a new reaction 3b as HA $\overset{h\nu}{\to}$ A$^{red}$ + X;   A$^{red}$ + X → A′; A$^{red}$ + NO$_2$ → A′′ + HONO was added in the text.

Line 59-60, the heterogeneous reaction NO+NO2+H2O is completely unimportant and not state of the art. In addition it should be "Andrés-Hernández et al.".

*Response:* We removed the reaction as NO+NO$_2$+H$_2$O from the text.

Line 61: There is only one study by Zhang and Tao (delete the a) und the same reference is listed twice in the references (lines 899-904).

*Response:* Line 62, the reference of Zhang and Tao., 2010 has been corrected.

Line 61: the reaction NO2*+H2O was studied by Li et al. and not by Finlayson-Pitts. In addition, also this source is completely unimportant (see Carr et al. and Amedro et al.) and was simply an Excimer laser two-photon artefact: : :.

*Response:* We removed the reaction NO$_2$*+H$_2$O from the text.

Line 63: The heterogeneous reaction of NO with adsorbed HNO3 is also completely unimportant at atmospheric conditions (gamma <10ˆ-9, see J. Phys. Chem. A, 2004, 108, 5793-5799).

*Response:* We removed the heterogeneous reaction of NO with adsorbed HNO$_3$ from the text.

Line 64: delete the "a" for Zhou et al. and delete again one of the double references at the end (lines 908-913).

*Response:* The reference of Zhou et al., 2011 has been corrected in Line 65.

Line 74: either use R1 or reaction 1 (unify).

*Response:* Line 82, R1 has been corrected to "reaction 1"

Lines 77-78: delete the last sentence; that describes already the results.

*Response:* The sentence "However, this is not the case in this work." has been deleted from the text.

Line 85: the instrument can measure down to 0.2 ppt, see Atmos. Chem. Phys., 2008, 8, 6813-6822, doi: 10.5194/acp-8-6813-2008.

*Response:* Line 93, the detection limit of LOPAP has been corrected to 0.2 pptv and the reference Kleffmann and Wiesen, 2008 was referred.

Line 95-97: Stieger et al. also intercompared HONO, which should be mentioned here (HNO3 is not the topic of this manuscript: : :).

*Response:* The description of HNO$_3$ has been removed here and the description of HONO intercomparison was added "The cited group found a large scattering ($R^2$ =0.41) for the HONO comparison between MARGA and an off-line batch denuder without an inlet system.

The probable reason was the off-line analysis of the batch denuder sample as the resulting longer interaction of gas and liquid phase during the transport led to further heterogenous reactions." in Line 104-107.

Line 104-105. The statement is not correct, see major concerns.

*Response:* The statement was corrected "Our observations provide a direct intercomparison between LOPAP and MARGA for HONO field measurement" in Line 115.

Line 121: The used SJAC uses 100 _C hot water steam forming hot steam droplets on which different reactions of NO2 form nitrite (e.g. NO2 + organics, see Gutzwiller et al.) which show positive temperature dependencies. Thus also the aerosol nitrite MARGA data should be used with caution. This interference can be easily tested by spiking HONO free NO2 to the instrument during a field measurement (=> % NO2 interferences for nitrite...).

*Response:* We agree with the reviewer, the particulate nitrite data of the MARGA system could be interfered by the different reactions of $NO_2$ in the SJAC system, hence uncorrected aerosol $NO_2^-$ data were not used by the authors.

Line 165: At what temperature were the nitrite solutions stored in the fridge? Should not be below freezing temperature, see above.

*Response:* The dew samples were stored in a fridge with approximately 6 ℃ (was added in line 193). We avoided long storage times between sampling and analysis.

Line 195 ff: Were the clear sky J-values from the TUV model scaled by the measured global radiation for short fluctuations by local cloud cover (see figure 2) or was that really done by data from the NASA web page (see line 201)? Normally this is done by scaling the clear sky TUV values with measured radiation (e.g. from a J(NO2) filter radiometer: : :).

*Response:* The J-values from the TUV model was scaled by the measured global radiation.

Lines 213-215: I do not understand that statement. While the surface pH of a dry batch Na2CO3 denuder should be very high (pH=10?) the pH of the MARGA is close to neutral (5.7, see line 121), so they are different!?

*Response:* We improved our explanation in Line 247-251. It now reads "Genfa et al. (2003) reported that they found a discrepancy between two denuder systems working with $Na_2CO_3$ and $H_2O_2$ resulting in different pH. However, in the comparison by Stieger et al. (2018), the MARGA system and the off-line batch denuder had the same pH and the found differences (scattering) cannot be explained by pH differences."

Lines 262-264: No that statement is not corrcet, the trend of HONO (strongly decreasing during daytime) is different to HNO3 (almost constant), see Figure 4. Reasons are the decreasing HONO precursor concentration NO2 (increase of the BLH; both are ground

emitted or formed species), while HNO3 is formed by NO2+OH homogeneously in the gas phase and decreasing NO2 is compensated by increasing levels of OH during daytime: : :

*Response:* We are sorry for the wrong statement and appreciate the explanation; we deleted the statement for the "HNO$_3$".

Line 271-273 and figure 4: Please also show the HONO/NOx ratio in figure 4.
*Response:* The HONO/NOx ratio was added in Figure 4.

Lines 274-276: A formation or loss reflects dc/dt while a frequency (s-1) is a first order rate coefficient (apples and oranges), reformulate the sentence.
*Response:* The sentence was reformulated as "This decrease during nighttime indicates the HONO loss process (dry and wet deposition, trapped in the boundary layer or dew etc.) surpassing the HONO formation from the NO$_2$-to-HONO convention" in Line 317-319.

Line 286-289: Under acidic conditions HONO is not highly soluble, cf. the pKa. In addition the dew water was neutral during the dew campaign, see line 297.
*Response:* The sentence "might be slightly acidic due to acid contribution from acidic aerosols (such as NH$_4$HSO$_4$)" has been removed from the text.

Line 326ff: delete that section on NO+OH during night-time, see major concern.
*Response:* The section of NO+OH was deleted.

Lines 350-353 and equation 4: This equation (="two point fit") was not used, but the slope from all data, see sentence before, which is the correct procedure. Delete the equation.
*Response:* The previous equation 4 was deleted.

Line 354: Should be correlation and not covariance.
*Response:* This part has been deleted.

Line 355 and Figure 6: A plot of HONO against NO2 during night-time makes no sense as discussed in many former studies (e.g. by J. Stutz's group), as the HONO to NO2 ratio typically increase during night-time (see equation 4 to determine the "NO2 conversion frequency"). Thus typically the higher data points are those from the later night (with higher r.h: : :.) while the lower slope data reflect the early night (with lower r.h., see artificial correlation).
*Response:* Previous Figure 6 has been removed from the paper.

Line 359: I do not understand the high value of the HONO/NO2 ratio of 11.3% while a HONO/NOx ratio of 4-5 % was also obtained (see table 1). Since NO is much lower than NO2 (see figure 2), the HONO/NO2 ratio should be only slightly higher than HONO/NOx ratio. Check the numbers for consistency.
*Response:* The HONO/NOx ratio presented in Table 1 was calculated for the daytime and

nighttime using the time period 04:00-18:00, UTC and 18:00-04:00, UTC, respectively. However, the HONO/NO2 ratio was calculated for the early evening 17:30-00:00 UTC. That made the difference between the value HONO/NOx and HONO/NO2.

490    Lines 363-364: Reason for the higher conversion frequency is the different time period used. While here only the initial increase of the HONO/NO2 ratio during the early evening was used (later this is decreasing caused by more efficient HONO uptake on dew surfaces in Melpitz) in most other studies the almost entire night-time increase was evaluated, where formation and deposition overlaps leading to lower conversion frequencies.

495    *Response:* We agree to the statement by the reviewer. However, the calculation of HONO/NO2 followed the criteria as already mentioned in the text:
(a) only the nighttime data in the absence of sunlight (i.e., 17:30-06:00 UTC) were used;
(b) both HONO concentrations and $[HONO]/[NO_2]$ ratios increased steadily during the target case;

500    (c) the meteorological conditions, especially surface winds, should be stable.
In addition, some literatures also calculated the HONO/NO$_2$ by only using the initial increase of the HONO/NO2 ratio during the early evening. e.g. Alicke et al. (2002), Wang et al. (2017), (Alicke et al., 2003)…

Lines 379-380: Here again an artificial correlation is studied (see major concerns). HONO
505    and particles are both formed or emitted near to the ground and variation of the BLH causes the correlation. The question ground vs. particles can be only answered if parallel gradient measurements of HONO, NO2/NOx and particle surface area are performed (see discussion in Atmos. Environ, 2003, 37, 2949-2955).

*Response:* We agree to the statement from the reviewer. Accordingly, we reformulated the
510    sentences as "Moreover, given the weak correlation between HONO ($R^2$=0.566), $[HONO]/[NO_2]$ ($R^2$=0.208) and S$_a$ (Figure S6), this work concludes that, as previously reported (Wong et al., 2011;Sörgel et al., 2011;Kalberer et al., 1999), the HONO formation through heterogeneous NO$_2$ conversion on particle surfaces needs to be regarded as unimportant." in Line 417-421.

515    Line 407: Since the data between 17:30 and 22:00 is considered here, any measured BLH between midnight and 7:00 is meaningless?
*Response:* We corrected our description and terms as "Where H is the mixing layer height calculated from the backward trajectory based on GDAS data and a range of 20 m to 300 m from 17:00 until around 00:00 UTC in April 2018 was used in present study (Figure S7)" in
520    Line 433-435.

408-410: Was a LAI of 6 used (than exchange "we add" by "we used": : :) or did you add the

value of 6 to the LAI of 4-10? Reformulate: : :

*Response:* Line 438, "we add" was changed to "we used".

Lines 415-418: Also reformulate: ": : :the NO2 uptake coefficient: : : is larger than the reactive surface: : :". What you mean here is that the S/V of aerosols is much smaller than the S/V of the ground and thus the heterogeneous formation takes place on ground surfaces: : :

*Response:* Line 443-447, the sentence was reformulated as "However, it should be noted that the obtained $NO_2$ uptake coefficient on the ground surface is closely to the reactive surface provided by aerosols, but as the S/V ratio of particles is typically orders of magnitude lower than for ground surfaces, it is suggested that the heterogeneous reactions of $NO_2$ on ground surface may play a dominant role for the nighttime HONO formation."

Lines 436-437 and Fig. 7b: Since the HONO/NO2 ratio is time depending (see above) better plot the average first order rate coefficient for NO2 conversion against the inverse of the WS. You also would not plot the ratio product/reactant in a smog chamber experiment against any variable, but the rate coefficient. Since the WS is a marker for the vertical turbulent mixing the observed anti-correlation is a strong argument for the proposed ground source region of HONO.

*Response:* The Figure 7b was moved to SI as Figure S8. The relationship of $NO_2$-HONO conversion frequency ($k_{het}$) with the inverse of wind speed during nighttime (18:00-04:00) is illustrated in Figure S8b. In addition, the resistance model as shown in Figure S9 could indicate that the uptake of $NO_2$ on the ground surface would mainly be caused by its reaction process rather than aerodynamic transport and diffusion.

Line 474: Delete reaction (6) but add the new reaction 3 (photosensitized conversion of NO2, see major concerns).

*Response:* The former reaction (6) was replaced by a new reaction 3b (photosensitized conversion of $NO_2$) in Line 521.

Lines 500-501: If NO+OH contributes to ca. 50 % to the daytime HONO levels (see lines 496-497 => significant source!) than its sources strength should be some hundred ppt per hour. Check for the low number!

*Response:* Line 549-550, an additional source of 91±41 pptv h$^{-1}$ was derived beside OH reaction with NO for the daytime HONO after carefully checking, according to a HONO mixing ratio 98±15 pptv for the time period of 10:30-16:30 UTC.

Lines 504-505, hypotheses (1): Why should HONO formation in the morning be caused only by HNO3/NO3- photolysis and not by a more reasonable photosensitized conversion of NO2 (new reaction 3)? In the morning NO2 levels are still high while HNO3 is high only later during the day (cf. Fig. 4)! Please plot the "unknown source" against a) the product

(HNO3/NO3- x J(HNO3)) and b) (NO2 x J(NO2)). I expect the latter correlation is much better: : :

***Response:*** Line 588-598, we followed the reviewer's comments, photosensitized conversion of $NO_2$ was also applied in the model as:

"To investigate the contribution of photosensitized conversion of $NO_2$ (reaction 3b) on the diurnal HONO based on the second hypothesis, the following model calculation (Model 5) was performed:

$$\frac{d[HONO]}{dt} = k_3[OH][NO] + k_{het}[NO_2] + \frac{1}{4}(\gamma_a \frac{S_a}{V} + \gamma_g \frac{S_g}{V})\upsilon_{NO2}J_{NO2}[NO_2] - J_{HONO}[HONO] -$$

$$k_{10}[HONO][OH] - \frac{1}{4}\gamma_{HONO,ground}[HONO]\frac{\upsilon_{HONO,ground}}{H} \qquad \text{(Eq. 13)}$$

Here the $\gamma_a$ and $\gamma_g$ are the light-enhanced $NO_2$ uptake coefficient of $2.5\times10^{-4}$ and $2.0\times10^{-5}$ (Stemmler et al., 2006) on the aerosol surface and ground surface, respectively. And $J_{NO2}$ was multiplied with $\frac{light\ intensity}{400}$ when the light intensity $\geq$ 400 W m$^{-2}$. As shown in Figure 6 (Model 5, cyan line), the photosensitized $NO_2$ on the aerosol and ground surface could not reproduce the HONO morning peak. This favors the third hypothesis that dew evaporation processes release HONO resulting in the sudden morning peak." This may due to a relative low $NO_2$ concentration around 2-5 ppbv.

In addition, the plots of "unknown source" vs a) the product (HNO$_3$/NO$_3^-$ x J$_{(HNO3)}$) and b) (NO$_2$ x J$_{(NO2)}$) show that the latter correlation is better but both of these reactions could not explain a HONO morning peak as below.

[Figure]

[Figure]

**Figure R2**: Plots of "unknown source" vs a) the product ($HNO_3/NO_3^-$ x $J_{(HNO3)}$) and b) ($NO_2$ x $J_{(NO2)}$). Both of this correlation cannot explain the observed HONO morning peak.

580

Line 507: Should be equation 11.

Equations 13, 14, 15: the last term in the equations is again not correct (the deposition velocity should be exchange by the mean molecular velocity of HONO). And again, the concept of the uptake coefficient does not work for ground surfaces and fast uptake (transport

585    limited uptake at gamma > 10ˆ-5 - 10ˆ-4, depending on Ra and Rb).

*Response:* Line 559, "previous equation 10" was corrected to "equation 9".

In the Equation 11, 12, 13, 15 the mean molecular velocity of HONO, updated mixing height H of 70 m from the backward trajectory and recalculated average uptake coefficient of $(1.0\pm0.4)\times10^{-4}$ were applied in the model as presented above. As discussed above #6 (2), the

590    HONO uptake on the ground surface could be limited by the combination of aerodynamic transport and molecular diffusion regarding on the derived HONO uptake coefficient of $1.7\times10^{-5}$ to $2.8\times10^{-4}$ in the present study. Since such limitations are realistic for the atmosphere, the γ-coefficients calculated here could be broad scale applicability used for simulation of HONO production and loss at night when constrained by the observations. In

595    addition, the equation 7 of $\frac{1}{4}\gamma_{HONO,ground} \times [HONO] \times \frac{\upsilon_{HONO,ground}}{H}$ was used to reproduce the nighttime loss of HONO in the Equation 11, 12, 13, 15.

Line 530: The used value for J(HNO3) for the photolysis of surface HNO3/NO3- is too low, here "enhancement factors" (ratio surface photolysis/gas phase photolysis, see e.g. Environ Sci. Technol., 2018, 52, 13738-13746 and references therein) between 7 and 1000 have been proposed in recent studies for this reaction.

*Response:* Line 582-584, we followed the reviewer and "the photolysis frequency $J_{HNO3}$ was derived from the TUV model by multiplying an enhanced factor 30 due to a faster photolysis of particle-phase $HNO_3$ (Romer et al., 2018).", As a result, the photolysis of $HNO_3/NO_3^-$ (Model 4, pink line) still could not reproduce the HONO morning peak shown in Figure 6. However, this could well reproduce the HONO for the time period of 10:30 to 16:30 (UTC).

Line 546-547: Why are there two values (max/min) of k(emission) for each emission peak?

*Response:* To precise the $k_{emission}$ calculation, as shown in Figure 7, max/min of $k_{emission}$ were obtained in two more line parts in the plot of $\frac{HONO_{unknown}}{99.5-RH}$ vs. the internal time.

Lines 556-559: Again I do not understand the units of the integrated emission/deposition. Is that a concentration from a layer of 1 m or of 500 m? The numbers of released HONO molecules would be different by more than two orders of magnitude: : :

*Response:* We are sorry for the confusion. Exactly the quantity of HONO emission from the dew water was calculated by integrating the HONO morning peak in Model 6, and which was compared with the measured total HONO nighttime loss in the same sampling level. They are not exactly the emission/deposition rate, and we did not measure a HONO flux in our study. Finally, we decided to remove this part.

Section 4.3.3: If I understand the concept used (equation 17) correctly (?) than the authors take all dew nitrite and mix that after hypothetical evaporation as HONO homogeneously into a layer of variable height. Then they plot the resulting average (homogeneous) layer concentration against the height (see Figure S9). But in this case the resulting gradient does not reflect a real gradient in the atmosphere. If the concentration is calculated e.g. for a lower layer of 20 m, than all nitrite is already consumed and there is nothing left for any higher layers. Thus, the real concentrations would be much lower! But maybe I did not understand that correctly: : :

*Response:* We agree with the reviewer that previous Figure S9 cannot represent a HONO gradient, hence we removed the previous Figure S9. Then we applied the Equation 16 to calculate the HONO morning mixing ratio by assuming a homogeneous mixing. We corrected our description in Section 4.3.3 as following

"4.3.3 HONO emission from dew water evaporation in the morning

The hypothetical morning HONO mixing ratio (pptv) due to the complete dew water

evaporation could be estimated from the following equation by taking the measured dew nitrite and the mixing layer height […] Hence, the morning HONO mixing ratio could be estimated as: 2264.1±612.3, 1132.1±306.2, 452.8±122.5, 226.4±61.2 and 75.5±20.4 pptv, respectively, for a mixing height of 20, 40, 100, 200 and 600 m using the mean $F_{NO_2^-}$ from May 11th 2019 for the calculation."

Line 577: delete phenol, nitrophenol and HCHO, that is not the topic here.
*Response:* We deleted "phenol, nitrophenol and HCHO".

Lines 578-579: While emission of NH3 during evaporation might be reasonable, reemission of the highly sticky HNO3 is not expected. While HONO evaporates already e.g. at 80 % r.h., where you still find many formal monolayers of adsorbed water, HNO3 will still strongly stick to such humid surfaces: : : In addition, if you have acids (HNO3, H2SO4) and ammonia in the dew water, low volatile ammonium salts will be formed (e.g. ammonium sulphate) which will also not evaporate when the dew water has gone.
*Response:* We agree with the reviewer. Hence, we removed discussion of $NO_3$ in section 4.3.3.

Line 619: should be 90 % and not 58 %.
*Response:* Line 682, we corrected it as 90%.

Lines 623-625: If my interpretation of the gradient data is correct (see above) delete that section.
*Response:* we corrected and give a description in Line 687-689 as follow: "the morning HONO mixing ratio depending on dew water evaporation […], assuming a homogeneous mixing of evaporated HONO"

Lines 631-633: Correct for the numbers, see above.
*Response:* Line 697, we corrected the ground uptake coefficients for HONO and $NO_2$ at night as: $\gamma_{NO2 \to HONO\_g} = 2.4 \times 10^{-7}$ to $3.5 \times 10^{-6}$, $\gamma_{HONO,ground} = 1.7 \times 10^{-5}$ to $2.8 \times 10^{-4}$.

References: Line 677: Brüggemann

Line 682: 107, D22, 8196,: : :

Line 683: Pätz H.-W.

Line 685: 108, D4, 8247,: : :

Line 691: 157-160

Line 692: Rössler

Line 695: Andrés-Hernández

I stopped here, there are numerous errors in the reference list.

*Response:* We corrected the author name and other information as mentioned by the reviewer and we took a carefully check for all the reference.

Table 1: What is the difference between HONO and HNO2 (unify: : :)? In addition, which HONO data is shown here (LOPAP or MARGA)? Specify, should be the LOPAP data, see intercomparison results.
*Response:* We removed HNO$_2$ data in the Table 1 since it is the value from MARGA.

Figure caption 2: "The gaps were: : :" You find at least three gaps in the HONO data: : :
*Response:* We corrected the "The gap was" as "The gaps were" in Figure 2 caption.

Figure 3: The nitrite MARGA data should be used with caution, because of high interferences by different NO2 reactions, see above. In addition, I would not use the artificial HONO data by the MARGA, see intercomparison results.
*Response:* We agree with the reviewer and we did not use the HONO and nitrite data from MARGA, Figure 3 only to show the measurement by MARGA.

Figure 4: add the HONO/NO2 or HONO/NOx ratio.
*Response:* We added HONO/NOx in Figure 4.

**Supplement:**

Table S1: Reaction 2 and 2a are similar only with different complexity. And reaction 2a only works at ppm levels of NO2, otherwise uptake coefficient of N2O4(g) higher than one are necessary: : :
Add a new photosensitized conversion of NO2 on organic substrates. Here I would use a new number (e.g. 3a), since reaction 2a is a disproportionation reaction (red + ox of NO2) while in reaction 2b (dark) and in the new photosensitized reaction (e.g. 3b) NO2 is an oxidant.

Also use a new reaction (4) for 2c, also different mechanism.

You may remove reaction (4), (6) and (7), they are definitely unimportant.

*Response:* We removed the Table S1 and moved all the reactions to the main text. We added the photosensitized reaction as 3b, and renumbered other reactions following the suggestion of reviewer. We also removed the previous "reaction 4, 6 and 7".

Figure S3: delete the 20 in the lower y-axis

695    *Response:* The Figure S3 was improved.

Figure S5: specify the unit of the colour code. Is that the particle surface density Sa?
Specify in the caption.

*Response:* The color code is the particle number density and added in the caption.

Figure S8: Are the two fits (blue lines) used for the data from both days? Why isn't all shown
700    data fitted (than the slopes should be similar)? See also my question above to table 4.

*Response:* Now the previous Figure S8 was moved to the main text as Figure 7. These two
fits show a slower (min $k_{emission}$) and faster (max $k_{emission}$) HONO emission from the dew water
regarding on the different time process of HONO morning peak (Figure 2).

Figure S9: Please integrate HONO over 600 m (unit should be HONO/area). Is that number
705    similar to the dew nitrite surface density used? I expect it is much larger, see above. If yes
remove that figure and the corresponding section.

*Response:* We agree with the reviewer and removed the previous Figure S9. We also
improved our concern to the corresponding section 4.3.3. as shown in above.

710

**References**

Alicke, B., Platt, U., and Stutz, J.: Impact of nitrous acid photolysis on the total hydroxyl radical budget during the Limitation of Oxidant Production/Pianura Padana Produzione di Ozono study in Milan, J. Geophys. Res. Atmos., 107, 8196, doi:10.1029/2000JD000075, 2002.

Alicke, B., Geyer, A., Hofzumahaus, A., Holland, F., Konrad, S., Pätz, H.-W., Schäfer, J., Stutz, J., Volz-Thomas, A., and Platt, U.: OH formation by HONO photolysis during the BERLIOZ experiment, J. Geophys. Res. Atmos., 108, 8247, doi:10.1029/2001JD000579, 2003.

Hirokawa, J., Kato, T., and Mafuné, F.: Uptake of Gas-Phase Nitrous Acid by pH-Controlled Aqueous Solution Studied by a Wetted Wall Flow Tube, The Journal of Physical Chemistry A, 112, 12143-12150, 10.1021/jp8051483, 2008.

Huff, A. K., and Abbatt, J. P. D.: Kinetics and Product Yields in the Heterogeneous Reactions of HOBr with Ice Surfaces Containing NaBr and NaCl, The Journal of Physical Chemistry A, 106, 5279-5287, 10.1021/jp014296m, 2002.

Kleffmann, J., Becker, K. H., and Wiesen, P.: Heterogeneous $NO_2$ conversion processes on acid surfaces: Possible atmospheric implications, Atmos. Environ., 32, 2721-2729, 10.1016/s1352-2310(98)00065-x, 1998.

Kleffmann, J.: Daytime Sources of Nitrous Acid (HONO) in the Atmospheric Boundary Layer, ChemPhysChem, 8, 1137-1144, 10.1002/cphc.200700016, 2007.

Kurtenbach, R., Becker, K. H., Gomes, J. A. G., Kleffmann, J., Lorzer, J. C., Spittler, M., Wiesen, P., Ackermann, R., Geyer, A., and Platt, U.: Investigations of emissions and heterogeneous formation of HONO in a road traffic tunnel, Atmos. Environ., 35, 3385-3394, 10.1016/s1352-2310(01)00138-8, 2001.

Langenberg, S., Carstens, T., Hupperich, D., Schweighoefer, S., and Schurath, U.: Technical note: Determination of binary gas phase diffusion coefficients of unstable and adsorbing atmospheric trace gases at low temperature – Arrested Flow and Twin Tube method, Atmos. Chem. Phys. Discuss., 2019, 1-24, 10.5194/acp-2019-1050, 2019.

Li, G., Lei, W., Zavala, M., Volkamer, R., Dusanter, S., Stevens, P., and Molina, L. T.: Impacts of HONO sources on the photochemistry in Mexico City during the MCMA-2006/MILAGO Campaign, Atmos. Chem. Phys., 10, 6551-6567, 10.5194/acp-10-6551-2010, 2010.

Li, X., Brauers, T., Häseler, R., Bohn, B., Fuchs, H., Hofzumahaus, A., Holland, F., Lou, S., Lu, K. D., Rohrer, F., Hu, M., Zeng, L. M., Zhang, Y. H., Garland, R. M., Su, H., Nowak, A., Wiedensohler, A., Takegawa, N., Shao, M., and Wahner, A.: Exploring the atmospheric chemistry of nitrous acid (HONO) at a rural site in Southern China, Atmos. Chem. Phys., 12, 1497-1513, 10.5194/acp-12-1497-2012, 2012.

Lu, X., Wang, Y., Li, J., Shen, L., and Fung, J. C. H.: Evidence of heterogeneous HONO formation from aerosols and the regional photochemical impact of this HONO source, Environmental Research Letters, 13, 114002, 10.1088/1748-9326/aae492, 2018.

Romer, P. S., Wooldridge, P. J., Crounse, J. D., Kim, M. J., Wennberg, P. O., Dibb, J. E., Scheuer, E., Blake, D. R., Meinardi, S., Brosius, A. L., Thames, A. B., Miller, D. O., Brune, W. H., Hall, S. R., Ryerson, T. B., and Cohen, R. C.: Constraints on Aerosol Nitrate Photolysis as a Potential Source of HONO and NOx, Environmental Science & Technology, 52, 13738-13746, 10.1021/acs.est.8b03861, 2018.

Seinfeld, J. H., and Pandis, S. N.: Atmospheric Chemistry and Physics: From Air Pollution to Climate Change, Wiley, 2006.

755     Stemmler, K., Ammann, M., Donders, C., Kleffmann, J., and George, C.: Photosensitized reduction of nitrogen dioxide on humic acid as a source of nitrous acid, Nature, 440, 195, 10.1038/nature04603, 2006.

Stutz, J., Alicke, B., Ackermann, R., Geyer, A., Wang, S., White, A. B., Williams, E. J., Spicer, C. W., and Fast, J. D.: Relative humidity dependence of HONO chemistry in urban areas, J. Geophys.
760     Res. Atmos., 109, doi:10.1029/2003JD004135, 2004.

Su, H., Cheng, Y. F., Cheng, P., Zhang, Y. H., Dong, S., Zeng, L. M., Wang, X., Slanina, J., Shao, M., and Wiedensohler, A.: Observation of nighttime nitrous acid (HONO) formation at a non-urban site during PRIDE-PRD2004 in China, Atmospheric Environment, 42, 6219-6232, https://doi.org/10.1016/j.atmosenv.2008.04.006, 2008.

765     VandenBoer, T. C., Brown, S. S., Murphy, J. G., Keene, W. C., Young, C. J., Pszenny, A. A. P., Kim, S., Warneke, C., de Gouw, J. A., Maben, J. R., Wagner, N. L., Riedel, T. P., Thornton, J. A., Wolfe, D. E., Dubé, W. P., Öztürk, F., Brock, C. A., Grossberg, N., Lefer, B., Lerner, B., Middlebrook, A. M., and Roberts, J. M.: Understanding the role of the ground surface in HONO vertical structure: High resolution vertical profiles during NACHTT-11, J. Geophys. Res. Atmos., 118, 10,155-110,171,
770     doi:10.1002/jgrd.50721, 2013.

VandenBoer, T. C., Markovic, M. Z., Sanders, J. E., Ren, X., Pusede, S. E., Browne, E. C., Cohen, R. C., Zhang, L., Thomas, J., Brune, W. H., and Murphy, J. G.: Evidence for a nitrous acid (HONO) reservoir at the ground surface in Bakersfield, CA, during CalNex 2010, Journal of Geophysical Research: Atmospheres, 119, 9093-9106, 10.1002/2013jd020971, 2014.

775     Wang, J., Zhang, X., Guo, J., Wang, Z., and Zhang, M.: Observation of nitrous acid (HONO) in Beijing, China: Seasonal variation, nocturnal formation and daytime budget, Sci. Total Environ., 587, 350-359, 10.1016/j.scitotenv.2017.02.159, 2017.

Wong, K. W., Oh, H. J., Lefer, B. L., Rappenglück, B., and Stutz, J.: Vertical profiles of nitrous acid in the nocturnal urban atmosphere of Houston, TX, Atmos. Chem. Phys., 11, 3595-3609,
780     10.5194/acp-11-3595-2011, 2011.

Yu, Y., Galle, B., Panday, A., Hodson, E., Prinn, R., and Wang, S.: Observations of high rates of $NO_2$-HONO conversion in the nocturnal atmospheric boundary layer in Kathmandu, Nepal, Atmos. Chem. Phys., 9, 6401-6415, 10.5194/acp-9-6401-2009, 2009.

Zhang, L., Wang, T., Zhang, Q., Zheng, J., Xu, Z., and Lv, M.: Potential sources of nitrous acid (HONO)
785     and their impacts on ozone: A WRF-Chem study in a polluted subtropical region, J. Geophys. Res. Atmos., 121, 3645-3662, doi:10.1002/2015JD024468, 2016.

---

## Author Comment (AC2) · 7 May 2020

"In her/his major concern 3) intercomparison, the referee #1 would also like to point the authors to another recent paper in which also a MARGA system was intercompared with a LOPAP: Xu et al., AMT.12, 6737-6748, 2019, with very similar results to the other reference mentioned there (strong overestimation of HONO by the MARGA).

**Response:** We thank Dr. Su for his comment and this information. Accordingly, Xu et al., 2019 is now referenced and compared with our present result in Line 237-240 "The result is in excellent agreement with the former intercomparison of both instrument types in the Chinese field campaign (Lu et al., 2010;Xu et al., 2019) where the HONO mixing ratio measured with the wet-denuder-ion-chromatography (WD/IC) instrument was affected by a factor of three on average."

**References**

Lu, K., Zhang, Y., Su, H., Brauers, T., Chou, C. C., Hofzumahaus, A., Liu, S. C., Kita, K., Kondo, Y., Shao, M., Wahner, A., Wang, J., Wang, X., and Zhu, T.: Oxidant (O3 + NO2) production processes and formation regimes in Beijing, Journal of Geophysical Research: Atmospheres, 115, 10.1029/2009jd012714, 2010.

Xu, Z., Liu, Y., Nie, W., Sun, P., Chi, X., and Ding, A.: Evaluating the measurement interference of wet rotating-denuder–ion chromatography in measuring atmospheric HONO in a highly polluted area, Atmos. Meas. Tech., 12, 6737-6748, 10.5194/amt-12-6737-2019, 2019.

---

## Author Comment (AC3) · 7 May 2020

General Comments

An exploration of the role of dew from field measurements has been long overdue. This work builds very well on prior field observations and lab experiments regarding effective HONO sequestration in dew. The Authors present a convincing study of the uptake and release of HONO from their observations and controlled field collections of dew nitrite. The production and loss process for HONO are parameterized to rates for HONO uptake into dew as well as subsequent release, which are then coupled to a box-model, finding that their observed rapid HONO increases on mornings with dew evaporation can be reproduced. The rates for NO2-to-HONO conversion and HONO deposition are compared against prior uptake observations quite well. Overall, the scientific quality of this manuscript makes it an excellent candidate for publication in Atmospheric Chemistry and Physics following major revisions to improve the manuscript clarity and data quality.

Major Revisions

**1.** Typos, phrasing, and writing clarity throughout have major issues in communicating the scientific findings of this work. Several instances where this makes following the discussion nearly impossible are noted in the detailed comments below. In other places, sentences are started with abbreviations or chemical structures. In many instances abbreviations are used first with the full spelling in brackets, when these should be presented the other way around. The entire manuscript should be revisited for clarity of writing by all of the Authors.

In three sections of the results and discussion (Sections 3.3, 4.2.1, and 4.2.3) there is no synthesis of the cases or findings to complete the sections. They have been left incomplete and should be revised.

**Response:** The typos, phrasing, and writing of the entire manuscript have been improved and

carefully checked by all of the authors. We also completed the section of 3.3, 4.2.1 and 4.2.3.

**2.** 2.1 The intercomparison between the MARGA and LOPAP is a very weak component of this manuscript. Detection limits are not given for either technique and cannot be assumed to be the same as from the prior reports cited by the Authors. These need to be determined at each field site from controlled calibrations and careful collection of field blanks. The collection and correction of field blanks from overflowing the MARGA inlet with zero air are not presented. Were they collected and was a correction applied? How were backgrounds in the MARGA determined?

**Response:** During the field campaign in Melpitz, the temperature of the stripping coil of LOPAP was kept constant at 25 °C by a thermostat. Automatic zero air (Air liquid, Alphagaz 2, 99.9999%) measurements were performed for 30 min per 12 h measurements to correct for zero drifts. In addition, three calibrations using $NO_2^-$ standard solution (Heland et al., 2001) were applied in the beginning, middle and end of the campaign, to derive the HONO mixing ratio of our field campaign with a detection limit of LOPAP as 0.6 pptv. This information was clarified in Line 159-168.

The detection limit of the HONO measurements with the MARGA system is 0.02 μg m$^{-3}$ (10 pptv) (Stieger et al., 2018). We performed blank measurements for the MARGA before the intercomparison campaign in 2018. For this, the MARGA was set to the blank measurement mode that has a duration of six hours. Within the first 4 hours, the MARGA air pump was off and the denuder and SJAC liquids were analyzed. The first- and second-hour samples were discarded as they still included residual concentrations. The evaluation of the blank concentrations was performed for the third- and fourth-hour samples. Within these blank samples, 0.00 μg l$^{-1}$ of nitrite were measured both in the gas and particle phase indicating no background nitrite collection. This information was added in Line 142-150.

2.2 Calibration techniques for each instrument are also not presented and the arguments for the measurement bias being high for the MARGA are incorrect. Prior studies with similar wet denuder systems have shown that the NO2-SO2 and NO2-H2O corrections very small (VandenBoer et al., 2014) and cannot possibly explain this discrepancy. Further to this, the same work also demonstrates that in high NH3 atmospheres, similar to those observed in this work, that the denuder pH is sufficient alkalinity and buffer capacity to collect the observed HONO quantitatively as nitrite. This prior work also makes a comparison with a home-made version of the LOPAP with inlets separated vertically by several meters, where intercomparison was only made when both instruments we calibrated with the same sodium nitrite solution. A strong capability to accurately measure HONO by both instruments was demonstrated. Similar attention to measurement quality must be made by the Authors here to improve the quality of their intercomparison.

**Response:** Xu et al. (2019) found significant contributions of $NO_2^-$ due to the oxidation of $SO_2$ with $NO_2$. However, the mentioned study also summarized that the influence of the reaction of $SO_2$ with $NO_2$ on the resulting MARGA HONO concentration is low for a pH below 6. The reason is the competition of the $SO_2$ reactions between $H_2O_2$ and $NO_2$. Latter reaction is more effective for higher pH. Thus, at a denuder pH of 5.7, the influence of the reaction between $SO_2$ and $NO_2$ on the higher MARGA HONO concentrations should be low.

However, Xu et al. (2019) found in their comparison between MARGA and LOPAP also a large scattering between both instruments and identified influences of $SO_2$ and $NH_3$. They suggested that the sampling of the acidifying and alkalic compounds, respectively, are the main reason for the deviations as both compounds influence the pH of the MARGA absorbance solution and, thus, the formation of potential HONO artefacts within the MARGA analysis. Additionally, high $NH_3$ concentrations could favor the sampling of HONO with the MARGA.

The calibration information of LOPAP was added in Line 164-168. The MARGA instrument used in this campaign was also well calibrated (Line 142-150), all of these procedures can ensure us a high quality of HONO measurement.

2.3 The results presented in the intercomparison (Section 3.1) are challenging to follow. The Authors mention 'batch denuder' (Line 215) and 'offline batch denuder' (Line 217) but this is not explained clearly anywhere. What are these batch denuders? How were they prepared and why are they relevant to measurements being compared between the MARGA (an online instrument) and the LOPAP?

**Response:** We excuse for the confusion. We unified "batch denuder" and "offline batch denuder" to "off-line batch denuder". The off-line batch denuder setup is similar to the MARGA WRD. It consists of a rotating annular denuder in an open system without inlet tubes to avoid interactions of sampled air and walls. The liquid is collected after one-hour sampling and is analyzed off-line with ion chromatography (Stieger et al. 2018). This information are provided in Line 243-247.

Both MARGA and off-line batch denuder used an absorption solution with a pH of approximately 5.7. Thus, the found scattering between both instruments presented in Stieger et al. (2018) cannot be explained by different pH. This comparison is only given as additional information to exclude pH artefacts.

2.4 The Authors conclude their intercomparison to say that a long inlet on the MARGA could explain the higher HONO they are measuring. Is this hypothesized to be from NO2 hydrolysis on the inlet? The mechanism of interference on the inlet is assumed. It must be made clearly. If yes, can all MARGA daytime HONO data below 500 pptv (e.g. two to three hours from every minimum in the afternoons when NO2 conversion to HONO on surfaces is minimized)

be used to determine whether a relationship between measured HONO and NO2 due to an inlet effect is likely? It should also be possible to determine whether the magnitude of this effect is really as high as the 58-90% enhancement observed here.

**Response:** We cannot specify the reactions that occur in the MARGA inlet system. However, heterogeneous formation of HONO within the inlet might explain approximately 30 % of the HONO artefact as we mentioned in Line 257-259. And other ca. 58% of the HONO artefacts could be caused by artefacts in the denuder solution as the by heterogeneous reactions of $NO_2$ and $H_2O$ as well as $NO_2$ and $SO_2$ in water described by Spindler et al. (2003) or VOCs by $NO_2$ in Line 252-258.

In addition, to proposed by Stemmler et al. (2006), the photosensitized $NO_2$ could contribute the daytime HONO formation, hence $NO_2$ conversion to HONO on surfaces could not exactly been minimized in the afternoon. Thus, it was hard to derive a relationship between measured HONO and $NO_2$ due to an inlet effect.

2.5 The photos of the inlet configurations (Figure S1a-b) do not make it very easy to understand what the sampling flows, line volumes, and therefore residence times, of the sampled gases were in M1 versus M2. The red text on Figure S1b is not possible to read. Was the sample residence time in M1 much smaller than in M2 due to a change in the inlet flow rate?

**Response:** The quality of Figure S1 was improved and more detailed information (e.g. sampling follow, line volume etc.) was added in the SI.

To improve the understanding of the chosen intercomparison setup, we refer to the Figure S1. In Figure S1b, the MARGA $PM_{10}$ inlet is shown. This inlet was used for both instruments in the first measurement period M1 to identify the inlet artefact. Therefore, the inlet tube was extended with a Y-connector (Figure S1a) within the measurement container. The tube from the inlet to the connector had a length of approximately 2 m.

For the second intercomparison period M2, the LOPAP inlet was set next to the MARGA inlet on the roof of the container. No interactions between LOPAP and MARGA inlet occurred.

A simple calculation of the MARGA inlet residence time was performed. Including a 2 m long inlet tube with a diameter of approximately 1 cm result in a volume of 157 $cm^3$. Using the air flow of the MARGA with 16.7 l min$^{-1}$, the resulting residence time of the air within the tube is approximately 0.57 seconds in the second intercomparison method. The flow rate for the LOPAP was 1.035 l min$^{-1}$. Combined with the flow of the MARGA, the residence time in the first intercomparison period was 0.532 seconds, which is only marginally shorter and a potential influence of stronger interaction between walls and sampled air can be neglected. This information is in SI.

In Figure 1, the Authors present the findings from their intercomparison and the systematic offset between the two techniques (i.e. the uncertainty in the slope seems small) suggests that the poor comparison is due a calibration or blank-correction issue rather than significant sources of interferences. The plot of the intercomparison regressions does not include the 1-sigma error evaluated in either the slopes or the intercepts and should be added. Looking closely at the measurements in Figure 1, there seem to be a number of observations of much higher HONO by the 5 min LOPAP measurements over the MARGA. Previously, (Sörgel et al., 2011) demonstrated that the LOPAP could sample fog droplets to result in such a positive bias, drawing off of prior work by (Bröske et al., 2003; Kleffmann et al., 2006). Were any fog events observed during the field campaign and was particulate nitrite observed by the MARGA and in the LOPAP HONO channel? Presumably, with so much dew, the meteorological conditions were also favourable for fog formation? This could further support the uptake and deposition of HONO into dew at the surface, as the MARGA nitrite measurements would observed this directly.

**Response:** The 1-sigma error evaluated in the slopes and the intercepts for the intercomparison plot was added in Figure 1.

As the reviewer mentioned, a number of observations have shown that 30 seconds averaged HONO data of LOPAP were higher than that of MARGA (Figure 1a) in the morning and middle-noon, which is not represented well when comparing 1 hour averaged HONO data from both instruments (Figure 1b). That indicates that the MARGA was not sensitive enough for the short time of HONO variation. We agree with the reviewer, that the meteorological conditions of dew formation were also favorable for fog formation. But unfortunately, no obvious fog events were observed. However, we cannot exclude that occasional ground fog events in the early morning were present but these ground fog events would probably not have affected the inlet in approximately 4 m above the ground.

However, particulate nitrite ranged from 0.012 to 0.11 $\mu g \ m^{-3}$ was also measured by MARGA and shown in Figure 3. However, it is really important to notice here that we used the particle-phase $NO_2^-$ concentrations with caution because of positively correlated temperature dependencies of the $NO_2^-$ formation by $NO_2$ reactions (Gutzwiller et al., 2002).

**3.** The reactions from Table S1 are referenced regularly throughout the manuscript and should be moved, either as a table or as separated reactions corresponding to their first presentation throughout the introduction.

**Response:** All the reactions referenced have been moved from previous Table S1 to the manuscript corresponding to their first presentation throughout the introduction and text.

**4.** The Authors suggest that NH3 and HNO3 are released from dew similarly in the morning, as their mixing ratios also increase shortly after sunset. However, the observed

180 particulate NH4NO3 also increases, which is likely more consistent with aerosol aloft being mixed down into the nocturnal boundary layer and repartitioning to release NH3 and HNO3. Volatilization of HNO3 from surfaces containing water does not seem plausible given the strong acid nature of this species. If this were the case, deposition of HNO3 would not be represented as a terminal sink in atmospheric models. While NH3 may be released from dew,

185 this would be challenging to discern from the data presented here. The several instances where this justification is made to bolster the release of HONO from dew should be removed from the manuscript and SI, along with the associated figures. The direct dew observations, model rate parameterizations, and subsequent model-measurement comparison are sufficiently strong to make this case.

190 **Response:** All the discussion about the dew water emission of $NH_3$ and $HNO_3$ has been removed from the manuscript, e.g. in sections 4.3.3 and 5.

**Detailed Comments**

Line 37: 'were' should be 'are'. This correction is needed frequently throughout the

195 manuscript and should be made where appropriate throughout.

**Response:** This part has been deleted from the text since we deleted the Table S1. We also checked all the text for this mistake.

Line 45: 'induced' should be 'activated'. The Authors should also cite the work of (Aubin and Abbatt, 2007) on this topic.

200 **Response:** Line 49, the "induced" was corrected to "activated"; and the reference of Aubin and Abbatt, 2007, is cited.

Lines 56-65: This is an extremely long list of reactions that need to be broken down into organized categories. Typically, where lists have entries that contain commas, the list items are separated using ';' so they can be easily distinguished.

205 **Response:** Line 55-65, this part has been reformulated and the reaction list items are separated using ";".

Lines 65-67: The potential role of dew releasing HONO was also discussed in (Lammel and Cape, 1996; Lammel and Perner, 1988; VandenBoer et al., 2014).

**Response:** References of (Lammel and Perner, 1988;Lammel and Cape, 1996;VandenBoer et

210 al., 2014) are cited in Line 75.

Lines 81, 88, and 90: Instrument full names should be given first, followed by their abbreviations.

**Response:** Done by following the comments in Line 89, 96 and 99.

Line 82: 'detect' should be 'detects'

215   **Response:** "detect" was corrected to "detects" in Line 90.

Line 90: should be 'found excellent agreement'
**Response:** "the" was deleted in Line 98.

Lines 94-97: Why does an HNO3 artefact matter? Should this be HONO? Given the subsequent lack of clarity in discussing the intercomparison issues below, additional details
220   about the artefact between the MARGA and batch and coated denuders with shorter inlet lines should be very clearly outlined here.

**Response:** We are sorry for the mistake, the description of $HNO_3$ has been removed here and the description of HONO intercomparison was added "The cited group found a large scattering ($R^2$ =0.41) for the HONO comparison between MARGA and an off-line batch
225   denuder without an inlet system. The probable reason was the off-line analysis of the batch denuder sample as the resulting longer interaction of gas and liquid phase during the transport led to further heterogenous reactions." in Line 104-107. In addition, more detailed information of the batch denuder and possible artefacts have been added in Line 243-252.

Line 117: Sentences should not start with numbers. Consider rephrasing to 'An inlet flow
230   of : : : '
**Response:** Line 128-129, the sentence was corrected to "An inlet flow of 1 $m^3$ $hr^{-1}$ …"

Lines 127-129: Was LiBr used in both the gas and particulate channels? It seems unlikely that this is easily done in the SJAC. Please clarify. Also, the second sentence should read ': : :both collected over the course: : :'
235   **Response:** LiBr was used in both gas and particulate channel added during the sample injection to the IC and was added in Line 138-140; "in" was corrected to "over" in Line 140.

Lines 136-137: Two reactions for the derivatization of nitrite to the azo dye are mentioned here but not presented anywhere. The reaction notation conflicts with the reaction numbers presented in Table S1. Please clarify and add the reactions, if desired.
240   **Response:** We clarified the reactions by presenting other reactions as "reaction 1…" as shown in text, also we referenced the publications (Kleffmann et al., 2006;Heland et al., 2001) for the derivatization reactions R1 and R2 in Line 160-161.

Line 139: 'air zero' should be 'zero air'
**Response:** Line 162, "air zero" was corrected to "zero air".

245   Lines 153-155: The methodology for cleaning the glass plates and collecting samples is presented, but no blank collections are presented where deionised water applied to the glass plates was measured. This should be included in Table 2 to increase the strength of these findings. It would also be valuable, if the Authors have such data, to present the nitrite

recovered from washing these glass plates after nights when no dew formation occurred to compare the deposited quantities. This would strongly support the magnitude of calculated uptake of HONO into the bulk water on these surfaces during dew formation events.

**Response:** The blank values were already subtracted in Table 2 but we added the blank $NO_2^-$ concentration to Table 2 as suggested by the reviewer and more information in lines 333-334. However, we do not have such data about the recovered $NO_2^-$ from washing these glass plates after nights when no dew formation occurred, an issue which should be addressed in the future.

Line 224: Why was PM10 nitrite not measured? If there was fog, or reactive coarse particulate matter as observed by (He et al., 2006; VandenBoer et al., 2015), then you may observe the partitioned HONO directly with the MARGA as in (VandenBoer et al., 2014).

**Response:** The PM10 nitrite was also measured and shown in Figure 3 in right Y axis. However, as mentioned above, we used the particle-phase $NO_2^-$ concentrations with caution because of positively correlated temperature dependencies of the $NO_2^-$ formation by $NO_2$ reactions (Gutzwiller et al., 2002) within the SJAC.

Line 249 onward: The discussion from here on is vague about which measurement is being used for each section. This needs to be clearly denoted. Presumably the LOPAP measurement is being taken as having the best accuracy for measuring HONO, but this needs to be clearly stated.

**Response:** We clarified this in Line 261-262, it now reads: "As a result, we chose the LOPAP-measured HONO in the following sections because of its high precision".

Line 262: 'concentrated' should be 'concentration'
**Response:** Line 306, 'concentrated' was corrected to 'concentration'.

Lines 264-266: This sentence is unclear and hard to follow. Rephrase.
**Response:** Line 297-298, We agree and the sentence was corrected as "The HONO morning peak might possibly be caused by the photolysis of particle-phase $HNO_3/NO_3^-$ (Zhou et al., 2003;Ye et al., 2016;Zhou et al., 2011).".

Line 267: 'during the campaign' is referring to the prior campaigns just cited in the preceding sentence or in this work? Please clarify.
**Response:** Line 309, we clarified it as "during our campaign in Melpitz".

Line 268: 'HONO would be reemitted in the atmosphere', also here 'lead' should be 'led' (there are many instances of this throughout the manuscript that need to be corrected)
**Response:** Line 310-311, "is" was changed to "would be"; "lead" was corrected to "led" and was checked all over the text.

Line 276: 'NO2-to-HONO conversion frequency'

**Response:** Line 319, we corrected this sentence as "… surpassing the HONO formation from the $NO_2$-to-HONO conversion".

Lines 279-284: These are two sets of observations, not cases. Please indicate why these are nicely categorized for further exploration by clarifying their utility. At Line 282 the Authors indicate that HONO increased with wind speed, but usually this corresponds to a decrease in concentration due to dilution. Is this increase related to wind direction? This section then ends without further exploration of the two cases and needs to be completed.

**Response:** Since these two sets of observations have been described in Line 291-292 and also Section 4.2.3, we decided to remove this paragraph. We agree with the reviewer, HONO mixing ratios could have decreased by enhanced dilution through stronger wind. As shown in Figure 2 and 5, this increase did not relate to the wind direction. However, we assume the increasing wind speed enhanced the dew water evaporation and the decrease of RH leading to the morning peak.

Lines 286-289: These two sentences are misleading and contradict an accurate discussion that follows regarding the pH-dependent effective partitioning of HONO between the gas phase and aqueous solution. Acidic water would readily volatilize HONO. Revise this for consistency with the following discussion. In particular, the work of (He et al., 2006) demonstrated that leaf surface washings were alkaline, which drove favorable HONO partitioning to the surface. Were any grass washings collected here to investigate the effective pH of the vegetated surface? What about the glass plates after exposure over nights where no dew formed? Either of these, but ideally both, would make a stronger case for effective dew uptake of HONO.

**Response:** We are sorry for the mistake. The wrong statement "might be slightly acidic due to acid contribution from acidic aerosols (such as $NH_4HSO_4$)" has been removed from the text. In addition, as shown in Table 2, the pH of dew water collected from the glass plate is neutral. However, we did not collect the grass washings and glass plates washing after exposure over nights where no dew formed. These valuable applications will be done in the future.

Line 295: 'concern' should be 'focuses'

**Response:** We agree, in Line 331, the "concern" was corrected to "focuses".

Lines 303-307: This needs to be clarified. I see no temperatures where dew water would be frozen in April. Did this actually occur in May? And why would higher F(NO2-) be observed on frozen dew? As the Authors state, the phase transition would act to inhibit HONO partitioning. This section needs to be revisited and revised for clarity. It is also surprising that this high F(NO2-) is included in the averaging over the other dew F(NO2-) observations. This

would deliberately bias the use of this value later. A better approach would be to statistically exclude the outlier using the Grubb's test or, better yet, include all of the observations in the average since the number of dew samples is very limited.

320

**Response:** Yes, frozen dew occurred also when the measured air temperature (in 5 m above the ground) was above 0 ℃. This was also the case in May 2019. No temperature below 0 ℃ was observed in a height of 5 m, but colder temperature near the surface resulted in freezing dew.

325

We are sorry for the misunderstanding. Exactly the higher $F(NO_2^-)$ was observed on May 11[th] where dew water was not frozen. Frozen dew water was observed for the other days (May 8[th], May 13[th] and May 14[th]) with lower $F(NO_2^-)$. These samples were discarded because of possible interferences in the HONO partitioning. Finally, $F(NO_2^-)$ obtained on May 11[th] was used for the following sections. We revised our description as below: "higher $F_{NO2^-}$ was

330

obtained on May 11[th], where dew water was not frozen. On other days (May 8[th], May 13[th] and May 14[th]) frozen dew water was observed, which likely inhibited HONO to dissolve. Hence, these frozen samples were not considered in this paper. On May 11[th], the final $F_{NO2^-}$ …".

On May 11[th], a third dew water sample was collected from 3:30 to 5:20 (UTC) after collecting the first sample (18:00 – 3:20 UTC) as shown in Table 2. The $NO_2^-$ concentration in the third

335

sample is lower. However, the total value of $F(NO_2^-)$ for this morning would be the sum of the first (8.0 μg m[-2]) and the third sample (1.43 μg m[-2]), which was used in our study for following calculations. We also clarified this in Line 335-336 and Line 342-345 and Table 2.

Line 312: End first sentence after 'evaluated'. Start next sentence with 'Generally'

**Response:** We agree and changed in Line 353.

340

Lines 315-319: If the Authors have no data that meet these criteria, then why bother listing them. Simply state that a direct emission number could not be quantified here and then apply one that has been widely used, such as the study by (Kurtenbach et al., 2001) cited on Line 314. This can then be applied to correct the HONO formation and loss rates at night, when the correction can be accurately used. This cannot be simply ignored.

345

**Response:** We removed the "criteria" from the section 4.1. Since (Kurtenbach et al., 2001) reported the emission factor (HONO/$NO_x$) as 0.3-0.8% in Wuppertal Germany and regarding on the condition (low HONO direct emission) of field campaign, a low emission factor of 0.3% obtained by (Kurtenbach et al., 2001) was used to correct the HONO emission from HONO formation of $NO_2$-to-HONO conversion. As a result, a lower [HONO]/[$NO_2$] ratio was

350

obtained than before. However, this has no change on $k_{het}$ within large uncertainty (Table 3).

Lines 331-334: This is a most unusual exercise. This pathway has been long ignored as it is well known to be negligible.

**Response:** This part about the nighttime OH has been removed.

Line 341: 'cases' should be 'conditions'

**Response:** Line 371, 'cases' was corrected to 'conditions'.

Line 362: High HONO/NO2 values were also reported in (VandenBoer et al., 2013).
**Response:** The reference (VandenBoer et al., 2013) was added in Line 388.

Line 370: This discussion is incomplete. What chemistry is happening here and why?
**Response:** Line 397-399, we conclude this section by "This could be ascribed to the higher S/V surface in the rural site because of the higher leaf area index (LAI, $m^2/m^2$) compared to an urban which might have enhanced heterogeneous $NO_2$-HONO conversion."

Lines 388-392: This needs to be revised. This logic is very hard to follow and is the inverse approach from what is typically presented to make such a comparison between a modeled rate and observations. The uptake coefficient comparison does not seem to follow logically from the prior calculation. The Authors should revisit this. It would be better to show that typical aerosol uptake coefficients used by (Tsai et al., 2018; Wong et al., 2012) fail to produce the observed quantities of HONO and to quantify the fraction that aerosol conversion represents so it is clear that aerosol conversion is trivial.

**Response:** As reported by Li et al. (2012), the uptake coefficient of $NO_2$ to HONO could be calculated if the entire HONO formation were taking place on aerosol surfaces. VandenBoer et al. (2013) also applied the same method to obtain an uptake coefficient of $NO_2$ to HONO. In this work, we corrected our uptake coefficient of $NO_2$ which was wrong because of the bad RH correction. Then this obtained uptake coefficient of $NO_2$ was used in the following model in section 4.3.2. We revised our description in Line 412-417 to make it clearer. The text was changed to "If the entire HONO formation was taking place on the particle surface, the calculated $\gamma_{NO2 \rightarrow HONO\_a}$ varied from $1.5 \times 10^{-6}$ to $1.9 \times 10^{-5}$ with a mean value of $(8.8 \pm 5.0) \times 10^{-6}$, which is lower than the reported values from VandenBoer et al. (2013) as $10^{-4}$ but it is in the good agreement with those observed in studies on relevant surfaces, which ranged between $1 \times 10^{-6}$ to $1 \times 10^{-5}$ (Kleffmann et al., 1998;Kurtenbach et al., 2001).".

Line 414: These references are not for ground proxies, but atmospheric aerosols. Remove and add studies that use real soils and soil proxies, such as (Donaldson et al., 2014; VandenBoer et al., 2015). These are both consistent with the observations made in this work and are more representative.

**Response:** (Donaldson et al., 2014; VandenBoer et al., 2015) was used to replace the earlier references in Line 442.

Lines 430-431: The trends discussed are not plotted on Figure 7a and need to be added. It seems unlikely that these trends are very robust. The kinetics of reaction 2 are not well constrained for increasing surface availability of H2O and this sentence should be rephrased

carefully.

**Response:** The trend has been added in Figure 7a. The statement about the kinetics of reaction 2 has been removed.

Line 453: The Authors should expand on why the difference between their observations and those from Boulder are so dramatic. The observations constraining HONO uptake on the ground surface presented in (VandenBoer et al., 2013) are all for nights where dew did not form, but deposition was observed to increase with RH (see equation 3 in Section 3.1 of that paper). Perhaps it is possible to estimate an effective HONO uptake coefficient from the flow tube work of (He et al., 2006) to compare to?

**Response:** We are sorry for the wrong value of HONO uptake coefficient in the previous version, we recalculated the HONO uptake coefficient by using mean molecular velocity of HONO and mixing layer height H from backward trajectory and then $\gamma_{HONO,ground}$ uptake coefficient was obtained from $1.7 \times 10^{-5}$ to $2.8 \times 10^{-4}$ with average of $(1.0 \pm 0.4) \times 10^{-4}$ (Line 481-482). This value is in the good agreement with the value of VandenBoer et al., 2013.

Line 461: Discussion is incomplete (see Major comment on this). Summarize the importance of your findings.

**Response:** We now used the Resistance Model and the according results are discussed in Line 495-510.

Line 465: 'above: : :' should be 'to reach an average minimum: : :'

**Response:** Line 514, "above" was corrected to "to reach".

Line 466: Why is the daytime maximum presented here? It is distracting from the point of this part of the discussion. Remove.

**Response:** "with the maximum mixing ratio of 1400±100 pptv" was deleted.

Lines 469-471: See Major comment on NH4NO3 thermodynamic partitioning and mixing. Remove this argument.

**Response:** "It should be noted that gaseous $NH_3$, $HNO_3$ and particulate $NH_4^+$, $NO_3^-$ also present the same trend like HONO as shown in Figure 4." was deleted.

Lines 499-501: Fix subscript typo on 'unknown' in the equation. The final sentence is hard to follow and this section does not end very clearly. What is the 'additional source'? P(unknown)? Revise for clarity.

**Response:** Line 548, subscript typo 'unknow' was corrected to be 'unknown'. Additionally, we added our explanation as "This could be well explained by the photochemical processes such as reactions 3b and 6 and would be discussed deeply in the next section." in Line 551-552.

Lines 514-515: The parameterization of boundary layer height used in the box model has not been explained and needs to be added somewhere. Were static or dynamic conditions of boundary layer height used? What measurements were used to set these boundary layer conditions and how are they justified to be suitable for use in this model?

**Response:** The mixing layer height H was calculated from the backward trajectory based on GDAS data as shown in Figure S7 and dynamic conditions of boundary layer height were used in the present study. We mentioned this parameter in Line 568-570 "The mixing layer height $H$ was calculated from the backward trajectory based on GDAS data as shown in Figure S7 and a dynamic conditions of boundary layer height was used."

Line 518: Following '12.5' the authors should add 'that we calculated from our observations'
**Response:** Line 571, 'that we calculated from our observations' was added after $\gamma_{HONO,ground}$ value.

Line 522: 'would be' should be 'was'
**Response:** Line 576, 'would be' was corrected to 'was'

Line 526: 'preceded' should be 'made'
**Response:** Line 579, 'preceded' was corrected to 'made'

Lines 529-531: There is a lot of literature stating that particulate NO3- photolyzes 10-1000 times faster than gaseous HNO3. How do the Authors justify why this was not included in the model?

**Response:** We agree with the reviewer, hence a factor of 30 from the recently publication (Romer et al., 2018) was multiplied to $J_{HNO3}$ due to a faster photolysis of particle-phase $HNO_3$ as mentioned in Line 583-584.

Lines 539-540: Was it dry ground or rain that was observed on the night of 23 April? Please specify.
**Response:** Line 604, we clarified that it was dry ground surface on the night of April 23[th].

Line 548: What is 'this value'? Is it the average? Or is it one of the maximum or minimum?
**Response:** Line 613, "this value" was corrected to "the average value".

Lines 577-579: Remove NH4NO3 discussion.
**Response:** Section 4.3.3, the discussion of $NH_4NO_3$ was removed.

Line 581: 'consistently' should be 'previously'
**Response:** Line 642, 'consistently' was corrected to 'previously'.

Line 583: 'these' should be 'our'
**Response:** Line 644, 'these' was corrected to 'our'.

Line 613: 'are rich of ground surface (forest and grass)' is inaccurate. Consider rephrasing to 'experience frequent dew formation'. The type of surface likely does not matter much when bulk water is available. If this were the case, the glass dew collectors would not have done a representative job of collecting the dew composition.

**Response:** Line 676, 'are rich of ground surface (forest and grass)' was corrected to 'experience frequent dew formation'.

Line 616: Conclusions and Atmospheric Implications: Rewrite fully to reflect manuscript changes made.

**Response:** These parts have been strongly rewritten.

Table 1: Clearly indicate the MARGA measurements.

**Response:** The obtained data of MARGA was noted as [b] in Table 1.

Table 2: The methods are unclear whether the pH of the dew was measured on a subsample of the total volume. Add some details on this to the caption and to the method section. It could be possible that direct measurement of the dew sample could lead to ion contamination from the salt bridge of the pH meter being in contact with the sample. Add blank sample values of collected nitrite here (i.e. deionised water passed over clean glass plates and trough into sampling vessel) for comparison. It would also be useful to see what nitrite deposition occurs to these surfaces at night in the absence of dew, so as to contrast the magnitude of change that the presence of dew makes.

**Response:** "pH was measured by a pH meter (mod. Lab 850, Schott Instruments) on a subsample of the total volume" has been noted as [b] in the Table 2 and method section 2.4.
The blank sample value was also added in Table 2. However, we do not have the data of "nitrite deposition occurs to these surfaces at night in the absence of dew" but we would like to apply it in the future.

Figure 1: The panel order in this figure is unusual. Typically, panels are lettered from a, starting at the top, with the letters located outside of the axes. The top panel intercomparison is not easy to read. The slopes should be positive on the plot, with HONO_LOPAP (pptv) on the bottom axis, not the top.

**Response:** The Figure 1 was improved by following the Reviewer's suggestion.

Figure 3: Change 'HONO_MARGA' to 'HONO'

**Response:** The 'HONO_MARGA' was corrected to 'HONO' in Y axis of Figure 3.

Figure 4: Remove panels e) and f)

**Response:** The panel e and f were described in Line 294-296 to discuss the hypothesis of morning peak could be caused by the photolysis of particle-phase $HNO_3/NO_3^-$. Hence the

490    panel e and f were kept here.

Figure 5: Are these measurements from the LOPAP? How was the error determined? This was not presented in the methods and should be added.

**Response:** We mentioned that it is the HONO data of LOPAP in the caption and the method of error calculation was added in Line 167-168 as "and then the error of HONO mixing ratio
495    was estimated based on the detection limit and a relative error as 10%."

Figure 6: This is not very convincing as there is a lot of overlapping data. Maybe create RH bins for each 20 % increment. Each bin would be centred at the average HONO/NO2 with x and y error bars corresponding to standard deviations of HONO and NO2 of all data collected in that RH bin (e.g. 0-20 %)?

500    **Response:** The previous Figure 6 about the "Correlation between HONO and $NO_2$ at night" was removed since the unconvinced discussion as the HONO to $NO_2$ ratio typically increase during night-time and could be artificial correlation between RH with HONO/$NO_2$.

Figure 7: Lettered labels on the panels are different from others above. Please make these consistent across all figures. It is very hard to take anything away from 7b. Consider moving
505    to the SI. Trend lines need to be added to 7a to address comments above.

**Response:** The lettered labels were made consistent with others Figures. The "previous Figure 7" was moved to SI as Figure S8. And trend line was added in Figure S8a.

Figure 8: I would like to see a third panel here that depicts the OH radical concentration, measured HONO, and HONO from Model 5 that includes upper and lower limits on the
510    model output according to the minimum and maximum k(emission) rates observed. This would provide a better comparison to the range of observations and would likely package the argument of dew partitioning even better.

Again, move the panel letters outside of the axes and keep them consistent with prior figure formatting.

515    **Response:** We agree with the Reviewer that a third panel with measured HONO and HONO from the model output according to the observed minimum and maximum k(emission) rates could provide a better comparison to the range of observations. However, since the Figure 6 (previous Figure 8) is already compact, we created a Figure S11 in the SI, which was explained in Line 619-623 as "In Figure S11, the observed HONO atmospheric mixing ratio
520    and the calculated HONO mixing ratio by model 6 using a minimum dew HONO emission $k_{emission} = 0.006$ pptv %$^{-1}$ s$^{-1}$ and maximum dew HONO emission $k_{emission} = 0.026$ pptv %$^{-1}$ s$^{-1}$, respectively, show that HONO emission from the dew water evaporation… ".

Figure S1: Why are both sets of photos separately labeled? Add schematics to depict tubing diameters, lengths, and flows for each sampling configuration. The schematic needs to be

525      consistent with the requested revisions to the sampling methods section. Fix caption to be accurate.

**Response:** Figure S1 was improved to indicate the sampling flow of LOPAP and MARGA, the length and diameter of the sampling line. The sampling strategy was also described in the SI.

530      Figure S2: What is the trough material made of?

**Response:** The trough is made of polyvinyl chloride.

Figure S8: This should be in the main manuscript. It is a great figure for this paper.

**Response:** The previous Figure S8 was moved to the main manuscript as Figure 7.

Figure S9: Should 'evolution' be 'structure'? Evolution implies time dependence which is not

535      what is described in the text of the manuscript.

**Response:** The previous Figure S9 was removed because of the wrong defined concert.

**References**

[revised manuscript text omitted]

---

## Author Response (AR2)

**Leibniz Institute for Tropospheric Research**

**Prof. Hartmut Herrmann**
Head of TROPOS Atmospheric Chemistry Department
herrmann@tropos.de
phon: +49 341 2717 7024
fax: +49 341 2717 99 7023
Permoserstraße 15
04318 Leipzig
24.07.2020

Leibniz-Institut für Troposphärenforschung  Permoserstraße 15  D-04318 Leipzig

The Editor
Atmospheric Chemistry and Physics

**Submission of revision for the Atmospheric Chemistry and Physics "Major Revision" manuscript 'Role of the dew water on the ground surface in HONO distribution: a case measurement in Melpitz' (MS No.: acp-2019-1088) by by Yangang Ren, Bastian Stieger, Gerald Spindler, Benoit Grosselin, Abdelwahid Mellouki, Thomas Tuch, Alfred Wiedensohler and Hartmut Herrmann**

Dear Editor,

Please find attached here our response to the reviewer comments for the manuscript mentioned above together with its revised versions of the manuscript and supplement. We would like to thank both of the reviewers for all of their valuable and insightful comments to improve the manuscript. We have carefully considered all the reviewer comments and revised the manuscript accordingly. Below, we provide responses to the comments in blue, with changes made in blue in the manuscript.

Sincerely yours,

Prof. Dr. H. Herrmann
Professor of Atmospheric Chemistry
Head of TROPOS Atmospheric Chemistry Department

Leibniz Institute for Tropospheric Research
Phone: +49 341 235-3210
Fax: +49 341 235-2139
info@tropos.de
http://www.tropos.de

Commerzbank Leipzig
Account No: 102 14 50
Sort Code: 860 400 00
IBAN: DE77 8604 0000 0102 1450 00
SWIFT CODE: COBADEFF 860

Mitglied der
Leibniz-Gemeinschaft

**Referee #1**
The authors gratefully thank the reviewer for the comments and suggestions. We have revised our manuscript according to the reviewer's suggestions and comments. **All the changes and responses to the reviewers' comments are listed below point-by-point in blue according to a new line numbering in the revised manuscript. The major changes are highlighted with blue in the revised manuscript.**

Comments to the revised manuscript by Ren et al.

In their revised manuscript Ren et al. considered most concerns of the two reviewers. However, I still have three major concerns (and some minor) which should be considered before publication in ACP.

Major Concerns:

1) Calculated NO2 uptake coefficients for particles surfaces (page 12, section 4.2.2):
In their revised manuscript Ren et al. now use the correct equation for converting the uptake coefficient into a first order rate coefficient (see Eq. 3). But the calculated theoretical uptake coefficients to explain night-time formation of HONO on particles are still unrealistically low (1.5x10^-6 – 1.9 x10-5, see line 414) in contrast to former studies who determined values of 10^-4 and larger and from which HONO formation on particles could be easily excluded (there is no such fast NO2 kinetics known from lab studies…). Reason for the low number of the authors (with which the HONO formation could be explained!? in contrast to the author's conclusion…) is the unrealistically high S/V(a) ratio of 9x10^-3 m^2 m^-3, which they gave only in the response letter (page 9) and which should be also specified in the manuscript.
I found S/V(a) values of 3x10^-4 m^2 m^-3 for heavily urban conditions (Finlayson-Pitts and Pitts textbook) and a value of ca. 200 um^2 m^-3 for Beijing (ACP, 17, 2017, 12327) which can be converted into 2x10^-4 m^2 m^-3. Thus, in Melpitz (low pollution) the aerosol surface density should be 45 times higher (!) than in Beijing (polluted)? The authors should check their data again. If the lower S/V ratio from Beijing was used, the theoretical uptake coefficient would be 45 times higher (and for Melpitz even more…) bringing the theoretical uptake coefficient into a reasonable range (10^-4) similar to former studies and thus confirming the author's argument (no HONO formation on particles…)!

**Response:** The particle surface density $S/V_a$ was calculated using the following equation: $\sum_l^u (\pi D_\rho^2 n)$ by assuming the particles are in spherical shape where l and u are the lower and upper channel boundary, respectively. $D_\rho$ is the particle diameter (channel midpoint) and n is number weighted concentration per channel. However, unfortunately, the particle number n was confused with $dN/dlog D_\rho$. We are sorry for this mistake and thanks to the reviewer for

recognizing it. Accordingly, S/V$_a$ was recalculated as (5.1-9.9)$\times10^{-4}$ m$^2$ m$^{-3}$, was further corrected with a hygroscopic factor $f$(RH)=1+a$\times$(RH/100)$^b$ (empirical factors a and b were set to 2.06 and 3.6, respectively) resulting in (0.6-1.9) $\times10^{-3}$ m$^2$ m$^{-3}$. This value now relates much more reasonable to S/V ratios reported in the literature, e.g. for Beijing as 0.2-3.4$\times10^{-3}$ m$^2$ m$^{-3}$ (Liu et al., 2014, Wang et al., 2016). The value of ca. 2$\times10^{-4}$ m$^2$ m$^{-3}$ reported by Cai et al., (2017) was for observed new particle formation (NPF) events in Beijing and the value ranged from 3.5$\times10^{-4}$ m$^2$ m$^{-3}$ to 1.1$\times10^{-3}$ m$^2$ m$^{-3}$ for non-NPF event.   Finally, the calculated $\gamma_{NO2\rightarrow HONO\_a}$ varied from 2.8$\times10^{-5}$ to 3.8$\times10^{-4}$ with a mean value of (1.7$\pm$1.0) $\times10^{-4}$ in this work. All values have now been updated in the manuscript as:

Line 417-419, "The particle surface density $S_a$ was calculated as (5.1-9.9) $\times10^{-4}$ m$^2$ m$^{-3}$ from the particle size distribution …. The particle surface density $S_a$ was further corrected to be (0.6-1.9) $\times10^{-3}$ m$^2$ m$^{-3}$ …"

Line 431-435, "the calculated $\gamma_{NO2\rightarrow HONO\_a}$ varied from 2.8$\times10^{-5}$ to 3.8$\times10^{-4}$ with a mean value of (1.7$\pm$1.0) $\times10^{-4}$. This theoretical uptake coefficient falls into a reasonable range of 10$^{-6}$ - 10$^{-4}$ similar to former studies (Kleffmann et al., 1998, Kurtenbach et al., 2001, Wong et al., 2011, VandenBoer et al., 2013)."

2) Plot of k(het) against the inverse wind speed:
In response to my concern from the first review the authors have added now a new plot (Fig. S8b), which I recommended to confirm that HONO formation takes place on the ground and not on aerosols. First, now the authors have added this plot/argument in section 4.2.3 (HONO deposition on the ground) where it makes no sense!? Second, in their plot they not used the same data by which the values of k(het) were determined (see Table 3 and Fig S4). For k(het) the authors correctly used only the first initial increase of HONO/NO2, which was not too much affected by HONO deposition (Tab. 3). In contrast for Figure S8b data from all the night is evaluated (18:00-4:00), which is significantly affected by the HONO deposition making k(het) significantly lower than for data in Tab. 3. Thus, the same plot should be repeated for the data from Tab. 3 and then should be plotted against the inverse average wind speed for the same time period (apples and apples…).

**Response:** The first version of our plot of HONO/NO$_2$ against wind speed was presented in the manuscript in the ACPD 'Interactive Discussion stage' (as below Figure R1) and we discussed the impact of wind speed on NO$_2$-to-HONO conversion and HONO deposition. Hence, we put the plot in the section 4.2.3 (HONO deposition on the ground). As the reviewer suggested in the first round of review, we plotted the k$_{(het)}$ against the inverse wind speed (previous Figure S8b, now as Figure S8a) and kept it in the section 4.2.3 to discuss the same concern because it also included one data of observation "HONO peak observed at 0:00-2:00 (UTC) of April 25$^{th}$ (Figure 5) in line 306".

Our previous Figure S8b (now Figure S8a) includes the data of Tab. 3. Six conditions were selected to calculate the $NO_2$-HONO frequency following the criteria of Li et al. 2018, line 389) while four conditions were not considered in Tab. 3 because of their low $R^2$ (<0.4). All these k(het) values were calculated using only the first initial increase of HONO/$NO_2$. The previous Figure S8b (now Figure S8a) also included one data of observation "HONO peak observed at 0:00-2:00 (UTC) of April 25[th] (Figure 5) in line 306". This data shows a high k(het) of 0.06 h[-1] for high wind speed of 4.08 m s[-1]. It is assumed that the evaporation of dew droplets resulted in the temporary HONO peak.

In the revised version, we plot the data from Tab. 3 and also one data point according to the second set of observation (mentioned in section 3.3 and Figure 5) against the inverse average wind speed as shown in Figure S8a and move the plot to section 4.2.2 'Relative importance of particle and ground surface in nocturnal HONO production' as:

Line 466-474, "In addition, the relationship of $NO_2$-HONO conversion frequency ($k_{het}$ presented in Table 3) with the inverse of wind speed is illustrated in Figure S8a. As indicated in Figure S8a, wind speed was predominantly less than 3 m s[-1] during the field campaign period in Melpitz. High conversion frequency of $NO_2$-to-HONO mostly happened when wind speed was less than 1 m s[-1], which confirms that HONO formation mainly takes place on the ground. However, one point (in blue in Figure S8a) showed highest $NO_2$-HONO conversion frequency ($k_{het}$) when wind speed was ca. 4 m s[-1] according to the second set of observation mentioned in section 3.3 and Figure 5. The likely reason for the temporary HONO peak is the dew droplet evaporation after increasing wind speed."

[Figure]

**Figure R1.** Scatter plot of HONO/$NO_2$ against wind speed in the time interval of 18:00-04:00 (UTC).

3) Explanation of the morning HONO peaks by dew water nitrite.

The quantitative calculation of the peak HONO by evaporation of dew water nitrite is not correct and too simple. First, why do the authors not simply take F(NO2-) in Eq. 16 but multiply that by 2 and by the LAI (x6), i.e. why do they used a 12 times higher values than measured? If dew is condensing on the ground the volume of the water is limited by the amount of humid air which gets into contact with the cold surfaces but it is not limited by the

surface area!? Thus, on any higher surface area of plants (compared to the geometric surface of their glass plates…) the total dew volume would be the same! Thus, they should not use the enhancement factor of 12 here (2x6).

**Response:** There are multiple kinds of environmental surfaces as well as smaller surfaces may be more effective in collecting dew (Kotzen 2015). As mentioned by Wentworth et al., (2016), the volume of dew ($V_{dew}$) obtained from the collector is not necessarily representative of $V_{dew}$ that forms naturally on the grassland canopy because of their different cold contacted surface area with humid air. The $V_{dew}$ could also be affected by the collector materials (del Campo et al., 2006, Guan et al., 2014, Kotzen 2015), they noted that the single walled tree shelter was a better condensation collector as it had 40% more available surface area to collect dew due to its corrugations on one side. Kotzen (2015) concluded that the main ways to enhance dew formation and collection is to increase surface area and facilitating maximum radiative cooling to the open sky based on the testing of nearly two hundred materials.

In this work, we collected the dew water using a glass plate and the $NO_2^-$ concentration **per m$^2$ of the sampler surface** ($F_{NO2-}$) was calculated from the following equation:

$$F_{NO2-} = \frac{[NO_2^-] \times V_{dew}}{S \times 1000} \qquad \text{(Eq. 2)}$$

Here, $[NO_2^-]$ is the sample concentration in $\mu g\ L^{-1}$, $V_{dew}$ is the sample volume in ml of the glass sampler, S is the surface area of the glass sampler 1.5 m$^2$ as explained in line 353-358. Then the hypothetical morning HONO mixing ratio (pptv) due to the complete dew water evaporation could be estimated from the following equation by taking the measured dew nitrite and the mixing layer height:

$$[HONO] = \frac{2 \times LAI \times F_{NO2-}}{mixing\ height} \qquad \text{(previous Eq. 16)}$$

In which the complete Eq.16 should be:

$$[HONO] = \frac{2 \times LAI \times S \times F_{NO2-}}{H \times S} = \frac{2 \times LAI \times F_{NO2-}}{H}$$

$F_{NO2}^-$ is the $NO_2^-$ concentration per m$^2$ of the glass sampler surface, S represents the flat ground surface (analog to the surface area of the glass sampler). But the $V_{dew}$ on the glass sampler could be enhanced due to the larger cold surfaces from grass which can get in contact with humid air than the plat glass sampler. This enhancement factor was calculated as $2 \times LAI$ to take the vegetation-covered areas on the ground and the areas on the both sides of the leaves into account. This is why we used a 12 times higher values than measured $F_{NO2}^-$.

Accordingly, we modified Equation 16 to clarify in line 643-651:

$$\text{"}[HONO] = \frac{\alpha \times S_g \times F_{NO2-}}{H \times S_g} = \frac{\alpha \times F_{NO2-}}{H} \qquad \text{(Eq. 17)}$$

$F_{NO2}^-$ is the $NO_2^-$ concentration per m$^2$ of the glass sampler surface, $S_g$ represents the surface area of the flat ground (analog to the surface area of the glass sampler), α is the enhanced factor for $V_{dew}$ (dew water sample volume of the glass sampler in Eq.2) due to the larger cold

surfaces from grass which can get in contact with humid air than the flat glass sampler. α was calculated as $2 \times$ LAI to take the areas on the both sides of the leaves and the vegetation-covered areas on the ground into account. Regarding the grass height during the dew measurements (~30cm) that is approximately the height in April 2018 and May 2019, we used a factor of 6 for LAI.

Second, the dew evaporation took place during early daytime when HONO photolysis is already significant. At 7:00 (HONO peak time) J(HONO) will be approximately half of the noon time value and thus evaporated HONO is significantly lost by photolysis over its evaporation time period (ca. 3 h, see Fig. 6). This problem can be only solved by a simple model including HONO production (by dew nitrite evaporation) and loss by photolysis. Since I do not have the (J(HONO) data I simply used a linear increase of J(HONO) from zero at 4:00 to $5x10^{-4}$ $s^{-1}$ (at 7:00, see Fig. 6) in a model and used the maximum $F(NO_2^-)$ of 8 ug $m^{-2}$ ($1x10^{17}$ HONO $m^{-2}$). For a reasonable mixing height of 100 m this can be converted into a HONO concentration of 42 ppt. But the production term over a time period of 3 hours (see Fig 6, 4:00 – 7:00) is only P(HONO from dew) = 0.004 ppt/s (42ppt/10800s). Now if I use the model including also HONO photolysis (see above) the peak HONO concentration after ca. 2 h (then HONO is decreasing again by increasing photolytic loss…) is only 14 ppt (and not 42…). This value is much lower than the experimental increase of HONO during the morning peak of ca. 450 ppt shown in Figure 6. In addition, the average dew nitrite concentration was not 8 ug/m2 but 3.5 ug/m2 (see table 2). Thus, the nitrite is by far not enough to explain the HONO morning peaks.

Here one argument could be the dew measurements one year after the HONO measurements (see my first review) comparing apples and oranges. But maybe also other sources are still not correctly considered here.

**Response:** Firstly, we need to note that, higher $F_{NO2^-}$ was obtained on May 11[th] where dew water was not frozen as shown in Table 2. On other days (May 8[th], May 13[th] and May 14[th]) frozen dew water was observed, which likely inhibited HONO to dissolve. Hence, these frozen samples were not considered in this paper. On May 11[th], the final $F_{NO2}^-$ could be obtained by averaging $F_{NO2}^-$ of the sum (9.43 μg $m^{-2}$) of the first and third sample with the second sample (6.40 μg $m^{-2}$) on 11[th] May resulting in 7.91±2.14 μg $m^{-2}$. We explained this in line 358-364. Hence, the overall concentration increase from this source would be 453 pptv if dissolved HONO was released into the overlying air column of 100 m mixing height.

Secondly, we agree with the reviewer that our quantitative calculation of the HONO peak by dew water evaporation is simple not taking into account the HONO photolysis. Considering the HONO photolysis using the photolysis rate of HONO from the TUV model, a HONO maximum of 211 pptv would be released by dew evaporation for a mixing height of 100 m at 7:00 UTC after the process started at 4:00 UTC (Figure R2). This would account for ~30% of

the observed HONO morning peak in Figure 6. This low percentage might be a result of the different sampling time of dew measurement compared with HONO measurement. Although the above calculations may be well simplified, the results do suggest that the release of the deposited HONO on wet/moist canopy surfaces may contribute to the morning HONO concentrations right after dew evaporation. Further research is needed to quantify exactly the amount of released HONO on the atmospheric HONO concentrations.

Following this revision, we have modified the description in line 653-666, "Hence, the overall concentration increase from this source would be 2264±612, 1132±306, 453±122, 226±61 and 76±20 pptv, respectively, if deposited HONO released into the overlying air column for a mixing height of 20, 40, 100, 200 and 600 m. Since the released HONO was subjected to photolysis, using a $J_{HONO}$ from TUV model scaled by global radiation (section 2.7), a maximum [HONO] of 1053±45, 527±22, 211±9, 105±4 and 35±1 pptv for the mixing height 20, 40, 100, 200 and 600 m, respectively, would be contributed from the surface nitrite release at 7:00 UTC after the process started from 4:00 UTC. For a reasonable 100 m mixing height, this would account for ~30% of the observed HONO morning peak in Figure 6 and this low percentage might be a result of the different sampling time of dew measurement compared with HONO measurement. Although the above calculations may be well simplified, the results do suggest that the release of the deposited HONO on wet/moist canopy surfaces may contribute to the morning HONO concentrations in the overlying atmosphere right after dew evaporation."

[Figure]

**Figure R2**. Plot of releasing evolution of deposited HONO subjected to it photolysis from 4:00 UTC using the equation: $[HONO]_t=[HONO]_0×exp(-J_{HONO}×t)$. $[HONO]_0$ and $[HONO]_t$ are HONO concentration at time 0 and time t (60 s in this case). The $[HONO]_0$ at 4:00 UTC

was calculated from Eq. 16 by assuming all the deposited HONO was released.

Specific Concerns:

The following concerns are listed in the order how they appear in the manuscript.
Lines 48-49: either use an alphabetic or chronologic order of the references.

**Response:** We use an alphabetic order of the references for all the manuscript.

Sections 2.2 and 2.3
Please unify the units used for both instruments (e.g. detection limit in ug/m3 or ppt) for comparison

**Response:** We changed the detection limit of MARGA to 10 pptv in Line 148.

Section 2.7:
Please specify that the clear sky J-values by the TUV model were scaled by the measured global radiation, see my first review!

**Response:** We specify that the clear sky J-values by the TUV model were scaled by the measured global radiation in Line 246.

Line 234-235: you should not compare high time resolution data with the one hour averaged MARGA data, delete the sentence, that is trivial…

**Response:** We modified the sentence "The hourly HONO mixing ratio obtained from MARGA with the 30 seconds and hourly averaged HONO mixing ratios from LOPAP are shown in Figure 1a and 1b, respectively" in line 252-253.

Line 263: Should be accuracy and not precision. The precision of the data shown in the intercomparison is much higher than the 90 % difference.

**Response:** We changed the "precision" to "accuracy" in line 278.

Line 295: Should be "the early morning variation trend", since the statement is only true for the first three daytime hours and not for whole diurnal data!

**Response:** We changed the "diurnal" to "early morning" in line 309.

Line 309: Where do I find argument (a)? The conclusion from Särgel et al.?

**Response:** We modified the sentence to make it clear, "As reported by Stemmler et al., (2006), the photosensitized $NO_2$ on humic acid could act as a source of HONO during the daytime. It was expected that the photosensitized $NO_2$ on humic acid might lead to a HONO morning peak within hypothesis (b)" in line 320-323.

Line 313-316: The reason for the similar increase of HONO and NO2 is the variation of the vertical mixing increasing the level of all near ground emitted of formed species. Radon would show the same behavior…

**Response:** We changed the sentence to "… This could be explained by the variation of the vertical mixing increasing the level of all near ground emitted of formed species or the heterogeneous conversion of $NO_2$ to HONO during nighttime and will be discussed in Section 4." in line 329-332.

Lines 341-348: I would recommend using the average dew nitrite (3.5+-3 ug/m2), since dew and HONO data are from different campaigns… And later I would also use the average HONO data (see Fig. 4) for comparison and not a single day (see Fig. 6).

**Response:** As we explained in line 358-364, "dew water was frozen until 1 hour after sunrise on May $8^{th}$, $13^{th}$ and $14^{th}$ 2019 but not on May $11^{th}$ 2019", which likely inhibited HONO to dissolve. This frost phenomenon was not found during the HONO measurement campaign in 2018. Hence we used value of $7.91 \pm 2.14$ $\mu g$ $m^{-2}$ on $11^{th}$ May for the following calculation and discussion. Exactly both Figure 4 and Figure 6 are the average HONO data, but the previous Figure 4 was included two sets of observation in Figure 5, which should be excluded from the average calculation. In this version, we correct the calculation and update the Figure 4.

Line 370-371: Have the authors subtracted 0.3 % from the experimental HONO/NO2 data? By the way in Kurtenbach et al. HONO/NOx was determined.

**Response:** As defined in Kurtenbach et al., (2001) and other studies like in Li et al., (2018), the $HONO/NO_X$ ratio was usually chosen to derive the emission factor of HONO in the freshly emitted plumes. Here, we firstly used this emission factor to correct the HONO concentration, then using this corrected HONO concentration to calculate $k_{het}$ from the least linear regression for $[HONO]/[NO_2]$ ratios against time, but not directly subtract 0.3% from the $HONO/NO_2$ ratio.
Line 387-388, we modified the sentence to be "Then in this work, a low emission factor of 0.3% was used to correct the directly HONO emission from vehicles (Kurtenbach et al., 2001)

to result a [HONO]$_{corr}$" to make it clear that "the authors subtracted 0.3 % from directly HONO emission from the NO$_2$-HONO conversion".

Line 397-399: No, as I already mentioned in my first review, the main reason is the different time periods considered (here only initial increase for the first 2 evening hours in contrast to other studies, where often the whole night was considered and for which parallel deposition leads to smaller values of k(het)).

**Response:** Because of the increasing role of HONO sink in the second half of night, as mentioned in the rural site studies of Alicke et al., (2003), Li et al., (2012), Acker and Möller (2007) and urban site study of Alicke et al., (2002), Wang et al., (2017), Acker and Möller (2007), the authors restricted the analysis of k$_{het}$ to the time frame from 18:00 to 24:00 like our work, so the argument of "This could be ascribed to the higher S/V surface in the rural site because of the high leaf area index (LAI, m$^2$/m$^2$) compared to an urban and might have enhanced the heterogeneous NO$_2$-HONO conversion." could be right. However, we changed the sentence to be "The higher value may suggest that a more efficient heterogeneous conversion from NO$_2$ to HONO is present in rural sites than in urban sites" in line 413-414.

Section 4.2.2: Please specify the S/V(a) used

**Response:** We specified the S/V(a) in line 417-421.

Equation 3: Please specify that you considered a 100 % HONO yield here (NO2+Org/soot/etc., see my first review).

**Response:** We already mentioned in line 430-431 "If the entire HONO formation was taking place on the particle surface…", here we add another sentence in line 425-426 "by considering 100% HONO yield on the particle surface (NO$_2$+Org/soot/etc)" to make it clear.

Equation 4: Please specify that you considered a 50 % HONO yield here (2 NO2 + H2O, see my first review).

**Response:** We clarified it in line 443 "by considering a 50% HONO yield from reaction 2a".

Line 443-447: Here you find exactly the argument for my major concern 1)…

**Response:** We thanks the reviewer for the insightful point here, we modified the argument in line 461-465 "It should be noted that the obtained NO$_2$ uptake coefficient on the ground surface is lower than the reactive surface provided by aerosols, but as the S/V ratio of

particles is typically orders of magnitude lower than for ground surfaces, it is suggested that the heterogeneous reactions of $NO_2$ on ground surface may play a dominant role for the nighttime HONO formation."

Line 503: Should be "… speed is smaller than …" In contrast to what is used here also Rb is depending on the wind speed (see e.g. VDI3782) and the lower the WS the larger is the quasi-laminar layer on the surfaces. Thus transport limitation gets important at low WS.

**Response:** We changed the sentence in line 523 as the reviewer suggested. Also we mentioned the transport limitation at low WS in line 520-521 "the HONO uptake will be transport-limited if the real uptake coefficients are $\geq 2.8 \times 10^{-4}$ and wind speed was less than 0.5 m s$^{-1}$".

Line 593: The upper limit is not given in Stemmler et al., 2006 and this study is not considering aerosol but bulk surfaces?

**Response:** We changed the citation and decide to use the same value of $2.0 \times 10^{-5}$ for the aerosol and ground surface as the literature (Zhang et al., 2016). The sentence was modified in line 606-607 "Here the $\gamma_a$ and $\gamma_g$ are the light-enhanced $NO_2$ uptake coefficient of $2.0 \times 10^{-5}$ (Zhang et al., 2016) on both of the aerosol surface and ground surface, respectively." The according Figure 6 was also updated.

Line 611-612: If k(emission) is determined from the experimental data, it is trivial that the model is doing an excellent job… But the values cannot be explained by the dew nitrite, see major concern 3)

**Response:** We thank the reviewer for this comment. As discussed in line 618-626, we defined a temporary HONO emission from dew water $k_{emission}$ related to RH using experimental and model result, which provide us a possibility to model the HONO emission from dew evaporation not only in this work but also for other studies in the future. As we already mentioned in the major concern 3, the HONO emission could not be fully explained by the dew measurement mainly due to the different sampling period for HONO and dew measurement. And a future simultaneously HONO and dew measurement could be planned.

Line 639: From where is that number (1400 ppt). In figure 6 (one day) the delta is ca. 450 ppt for the avarge campaign (Fig. 4a) it is ca. 300 ppt…?

**Response:** We are sorry for the confusion. Because the time of HONO morning peak was different within each day (Figure 2), so we averaged these morning peak value as 1400 pptv.

Both Figure 4 and Figure 6 show time averaged values. We updated them in this version to make it clear. Hence, we modified the sentence in line 660-663 "For a reasonable 100 m mixing height, this would account for ~30% of the observed HONO morning peak in Figure 6. This low percentage might be a result of the different sampling time of dew measurement compared with HONO measurement".

Line 668: It is J(O1D) from the TUV model scaled by the global radiation…

**Response:** We clarified it in line 694.

Line 664-665: No not the diurnal cycle of the mixing ratio, that is different by 90%! You mean the relative shape of the diurnal data…

**Response:** We mean that even the HONO mixing ratio measured by MARGA was higher than LOPAP, we also present the diurnal cycle from MARGA measurement in this work. Here we delete the sentence "However, the diurnal cycles of HONO mixing ratio were captured by both instruments" to make it clear.

Line 687: It is the maximum (…) value; on average it is 3.5ug/m2.

**Response:** We mentioned the "…maximum…" in line 711.

Line 688: Change the numbers (14 ppt in my example…) and it should be e.g. 1230+-160…

**Response:** We changed the numbers in line 712-715 "under consideration of photolytical losses and homogeneous mixing, the maximum contribution to the HONO morning peak from dew water evaporation could be calculated and ranged from $1053 \pm 45$ to $35 \pm 1$ pptv for mixing height of 20 to 600 m, respectively."

Line 695: From where is that number (970 ppt)? (see first version…)

**Response:** The total deposited HONO (in pptv) on the ground surface was assumed same as the total night-time HONO loss of $970 \pm 730$ pptv (6 h), calculated by integrating $L_{HONO}$ from 22:00 to 4:00 (UTC) from the nighttime measurement. We are sorry for the mistake. We should correct it as "$0.16 \pm 0.12$ ppbv h$^{-1}$" in the first version (the same value in the abstract), here we correct it in line 721.

Line 706: Should be minus(!) 0.016…(inverse correlation…)

**Response:** As defined in Eq. 14: $k_{\text{emission}} = \dfrac{d(\frac{HONO_{unknown}}{99.5-RH})}{dt} = \dfrac{\frac{HONO_{unknown}}{99.5-RH}(t_2) - \frac{HONO_{unknown}}{99.5-RH}(t_1)}{(t_2-t_1)}$, here $k_{\text{emission}}$ has a positive relationship with (99.5-RH).

I did not check the references…

**Response:** We checked the references carefully.

Figure 4: Please add a plot of J(HONO) to check the photolytic loss, see major concern.

**Response:** The plot of $J_{\text{HONO}}$ was added in Figure 4a.

Figure 8: In the legend it should be P(HONO->OH) and P(O3->OH), it is not the production of O3 but of OH… In addition how can the production of OH by HONO photolysis be negative? The term by HONO + OH is normally not significant?

**Response:** We changed the $P_{\text{HONO}}$ and $P_{\text{O3}}$ to $P_{\text{HONO->OH}}$ and $P_{\text{O3->OH}}$, respectively, in Figure 8 and also in section 4.3.4.
As we defined in section 4.3.4, the net rate of OH production from HONO photolysis ($P_{\text{HONO->OH}}$) was calculated from Eq. 17:
$P_{\text{HONO->OH}} = J_{\text{HONO}}[HONO] - k_{3a}[NO][OH] - k_9[HONO][OH]$
Where the consumption of OH by its reaction with NO and HONO was also considered. Hence, the net OH production rate in the early morning is negative as shown in Figure 8. And the term of HONO+NO could be important in the early morning but become less after the photolysis of HONO was strong.

"The detection limits and the blanks for the MARGA system were performed before the intercomparison campaign in 2018. The detection limit of HONO was determined as 10 pptv. The blanks were analyzed when the system was set up in the field to consider potential contaminations. For blank measurements, the MARGA blank measurement mode was used that has a duration of six hours. Within the first 4 hours, the MARGA air pump was off and the denuder and SJAC liquids were analyzed. The first- and second-hour samples are discarded as they still include residual concentrations. The evaluation of the blank concentrations was performed for the third- and fourth-hour samples. No discernable peaks above the instrument detection limits were identified in both the gas and particle phase channels.

When the solvent blanks for the MARGA were analyzed, was the system set up in the field location or in the lab? If they were collected in the lab, how can the Authors rule out contamination of the solvent or instrument components as the source of the discrepancy in the subsequent intercomparison? This is not clearly stated and confounds the quality of the

intercomparison work reported.

**Response:** During the blank measurements, the MARGA was located at the Melpitz field site to consider potential contaminations. And we should note that the solvent blanks were analyzed when the system was set up in the field to consider potential contaminations.

Lastly, the Authors report 0.00 ug/L of nitrite in their blanks, which is misleading. The instrument has a detection limit somewhere in the neighborhood of 0.02 ug/m3, which is a non-zero value. The detection limit is derived from the instrument signal to noise, which means nitrite could be present below the detection limits in these blanks. Consider instead a statement that 'no discernable peaks above the instrument detection limits were possible to identify both in the gas and particle channels'

**Response:** The reviewer is right on this point. In Lines 155-156, we rewrote: "Within these blank samples, 0.00 μg l 1 of nitrite both in the gas and particle phase were measured indicating no background nitrite collection." was changed to "no discernable peaks above the instrument detection limits were identified both in the gas and particle channels"

Page 5, Lines 164-168: The LOPAP detection limit is given, but the duration of the measurement it applies to does not. Since the comparison between the MARGA and the LOPAP is only valid at the hourly timesecale, this detection limit is required. For the remainder of the data analysis in the manuscript, which uses the LOPAP observations, the detection limit that applies to the time resolution of the dataset needs to be provided – and depicted on figures where appropriate.

**Response:** We should note that both acidic stripping solution and 0.8 mM n-(1-naphthyl)ethylenediamine-dihydrochloride were not changed during the campaign. As we mentioned in the manuscript, calibrations were conducted on April 17[th], April 20[th], 24[th], 25[th], April 29[th] during the campaign. These calibrations were used to determine the detection limits for different time resolution (30 seconds and 30 minutes) and also to check the stability of LOPAP in the campaign. We added this information now in line 173-175:

"Both the acidic stripping solution and 0.8mM n-(1-naphthyl)ethylenediamine-dihydrochloride solution were kept in the dark and were not changed during the whole campaign period."

And in line 178-181:

"In addition, calibrations using $NO_2^-$ standard solution (Heland et al., 2001) were applied in

the beginning (April 17th), middle (April 20th, 24th, 25th) and end (April 29th) of the campaign to derive the HONO mixing ratio. The detection limit of LOPAP was 0.6 pptv and 0.1 pptv for the time resolution of 30 seconds and 30 minutes, respectively…"

For the 30 seconds LOPAP data in the manuscript, the error of HONO was calculated based on the detection limit (0.6 pptv) and a relative error 10%. The error is used for our investigations in Figure 1a, Figure 2, Figure 5 and Figure 6.

The LOPAP accuracy is stated to be derived from a 'relative standard deviation' of 10 %, but how this was determined is unclear. Please clarify how this was calculated using the principles of analytical instrumentation (i.e. multiple evaluations of the calibration response, evaluation of an injected check standard). Has the propagated error from subtraction of signal in the second channel of the instrument been considered? It seems that this may not be a conservative estimate of the instrument performance and the Authors should be careful not to overstate this.

**Response:** The relative error is calculated by error propagation of all systematic errors, i.e. uncertainties in the gas flow ca. 2%, the liquid flow ca. 2 %, the error in the nitrite concentration during calibration 1 % +errors for the used pipettes/flasks and the 10 % is a conservative upper limit. Because all glass wares were not used exactly at 20 ℃ like recommended by the manufacturer, 2 times of the specified errors for all volumetric glass wares was applied. And finally we used a relative error of 10% in this work. We add this information in line 183-187.

"The relative error is calculated by error propagation of all systematic errors, i.e. uncertainties in the gas flow ca. 2%, the liquid flow ca. 2 %, the error in the nitrite concentration during calibration 1 % and errors for the used pipettes/flasks (two times of the specified errors of all volumetric glass ware since all glass ware was not used exactly at 20 ℃ like recommended by the manufacturer)."

Also the "real" HONO concentration is finally calculated by subtracting the value of channel 2 from channel 1, the propagated error from subtraction was also considered in the "real" HONO concentration.

Figure 1: The intercomparison regression has been performed using an orthogonal regression, yet it is unclear whether the measurement error has been included in this assessment. There are no error bars depicted on the datapoints, which suggests that appropriate regression using error-based weighting has not been applied. The lack of such consideratio could be valid if both instruments are subject to similar accuracy and precision metrics, but this information for the MARGA is not presented (see first comment in this section) makes the validity impossible to determine.

**Response:** We thank the reviewer for the insightful view. We added error bar for both the LOPAP data and MARGA data in Figure 1 and an orthogonal regression using error-based weighting was applied, which result the slopes of 1.57 and 1.66 for period M1 and M2, respectively. We updated this information in Figure 1 and section 3.1.

In panel (a) of the figure, it is clear that there is a major lag issue in the MARGA (i.e. the decay constant in the HONO measurement from a local maximum appears to be constant across many days, following the nocturnal maximum). The same observation is not seen in the LOPAP and also appears to be absent during M1. Clearly there is an inlet effect that is developing over time with HONO partitioning into the Teflon tubing, which would be expected if the inlet is not heated and can retain significant surface water. Further to this, particles will be depositing to the inlet material, as it is not conductive, resulting in chemical reactions between the gas flow and surface-deposited material. Both of these issues should be discussed as confounding factors in the relevant section of the manuscript.

**Response:** The reviewer is right with these points. It cannot be excluded that water or deposited particles have an influence on the gas- and particle phase composition by chemical reactions within the inlet. The inlet tubing of the MARGA outside of the container is surrounded by a shelter. Within the shelter, the inlet tubing is continuously ventilated with ambient air to reduce condensation or evaporation by temperature differences between sampled and air within the shelter that could favor condensation within the inlet tube.
Deviation between denuder measurements and other instruments for HONO measurements were also observed in previous studies. Volten et al., (2012) compared a miniDOAS system with an AMOR instrument for $NH_3$ measurements. The AMOR instrument is based on a continuous-flow wet denuder system similar to the MARGA. This group found a fast response for the miniDOAS measurements on short time scales, while the AMOR measurements showed offsets and a delay because of inlet memory effects by particles or water. Additionally, they suggest that the aqueous sample transport between sampling and analysis devices could explain a further delay. The same was described by Dammers et al., (2017), who compared a MARGA system with the miniDOAS.

These points are now discussed within the manuscript at Line 255-262: "In addition, the comparison between both instruments in Figure 1a shows a delay of the MARGA concentrations after reaching the maximum concentrations in the morning. This pattern was also observed in previous studies of Volten et al., (2012) and Dammers et al., (2017), who compared miniDOAS instruments with wet denuder systems. Compared to fast responses of the miniDOAS, the denuder-based instruments showed offsets and delays because of inlet memory artefacts by particles or water. Both groups also suggested transport effects of the

liquid samples from the sampling to the analysis unit resulting in delays and slow responses."

The Authors concluded that inlet production and denuder artefacts generating HONO from other atmospheric constituents are the source of the systematic bias observed. The argument for inlet HONO production is convincing from the results of M1 and should be possible to correct the MARGA dataset for based on tubing length and the atmospheric sample residence time, including the remaining inlet surface area upstream of the y-split fitting. However, the denuder artefacts argument is not convincing as the correlations between the two measurements are quite strong (perhaps stronger than currently depicted due to the absence of error-weighted regression). If interferences in the denuder were driving variability, they would depend on atmospheric composition that is decoupled from HONO chemistry, resulting in random error instead of systematic error. The second channel of the LOPAP should track interferences that could arise in the MARGA denuder quite well, although the magnitude would differ. The Authors could investigate the relative magnitude of the second channel signal of the LOPAP compared to the primary channel to discern the potential for additional interferences in the MARGA denuder.

**Response:** Kleffmann and Wiesen (2008) mentioned that most of the available intercomparison studies support that interferences are a general problem associated with chemical instruments. And the rates of interfering liquid phase reactions, i.e. PAN hydrolysis, $NO_2+SO_2$, $NO_2+phenols$, $NO_2+aromatic$ amines (line 269-271) tend to increase with increasing pH (Kleffmann and Wiesen 2008). These chemical interferences are expected to be even more severe for instruments that collect air samples under neutral or even basic conditions (see for example, Spindler et al., 2003; Genfa et al., 2003). The MARGA system used a solution at pH=5.7 (line 132) and LOPAP used a solution at pH=0. We agree with the reviewer that the interferences in the denuder would depend on atmospheric composition, but most related to the $NO_2$ as expected. Figure A1 shows the plot of interference in channel 2 vs. the atmospheric $NO_2$ mixing ratio and which indicate a good relationship ($R^2$=0.778) between $NO_2$ and interference. In addition, since $NO_2$ is also the mainly precursor of atmospheric HONO (nighttime $NO_2$ heterogeneous conversion R2, R2a, R2b and photo enhanced $NO_2$ conversion R3b), which may resulting in a likely systematic error rather than random error.

In addition, we also plot the Channel 2 vs the Channel 1 of LOPAP in Figure A2 as suggested by the reviewer. A ratio of Channel 2 to the Channel 1 as 0.0078 was obtained from the linear fitting in the Figure A2. This low interference value of LOPAP compared with interference value of MARGA in denuder (58%) could due to a different wet surface in channel 2 and denuder.

[Figure]

**Figure A1.** Plot of interference in Channel 2 of LOPAP vs the NO$_2$ mixing ratio in the atmosphere during the campaign period.

[Figure]

**Figure A2.** Plot of Channel 2 vs Channel 1 of LOPAP during the campaign period.

Finally, the ratio of MARGA to LOPAP observations from M1 are extremely close to the ratio of molecular weights between NaNO2 and the nitrite ion. The primary standard used for calibration of each instrument should be re-evaluated to ensure that these measurements are sound. There is a break in the data between M1 and M2, presumably due to instrument maintenance, as stated in the manuscript. Were both instruments recalibrated during this time? Were the same standard solutions used to calibrate both instruments? In the Reviewer's prior experience, these two considerations have been major sources of error in intercomparing instruments that rely on their calibration from aqueous nitrite standards, resulting in exactly this type of systematic error.

**Response:** A Titrisol (Merck) standard solution of 1000 mg of $NO_2^-$ was used to prepare the nitrite standard solution in a 1 l volumetric flask and used to calibrate LOPAP. The same standard solution has also been used to calibrate the MARGA. The break in the data between M1 and M2 might be due to the rearrangement of LOPAP sampling unit. Additionally, the LOPAP was recalibrated after we changed the position of the LOPAP sampling unit. The MARGA setup was not changed. We should mention that the LOPAP and MARGA have been used in the groups of ICARE and TROPOS, respectively, for more than ten years and several papers have been published (Bernard et al., 2016, Laufs et al., 2017, Stieger et al., 2018).

Technical Revisions:

Page 2, Line 40: Consider denoting reactions as '(R1)' with increasing numbers for subsequent reactions. This will simplify the notation used throughout the manuscript and create greater clarity.

**Response:** We agree with the Reviewer, and changed e.g. "reaction 1 …" to "R1 …".

Page 2, Line 62: Should 'homogeneous nucleation' be 'homogenous reaction'?

**Response:** "homogeneous nucleation" was changed to "homogenous reaction" in line 62.

Page 2, Line 64: 'However' is not necessary here and is used incorrectly in many places throughout the manuscript. Its use should denote a contrasting result or statement to the preceding sentence. Similar adverbs are used where they are not required throughout the manuscript, sometimes resulting in unclear meaning of the scientific results.
Also in this sentence, the Authors should be careful to state that multiple mechanisms may contribute a significant proportion of produced HONO. The way this is currently written suggests that only one will dominate a given dataset, yet the balance of the literature is clearly demonstrating to us that there are many mechanisms at work that can vary in their importance depending on the time of day. For example, dew!

**Response:** We thank the Reviewer for the grammar correction and we checked all the "however" using through the manuscript. Some other adverbs like "then, therefore, when, where, since … " were also corrected. We agree with the reviewer about the statement here and hence decide to remove the sentence of "However, the dominant HONO formation mechanism is still under discussion."

Pages 2-3, Lines 66-73: The reactions presented should increase by one and be in order. The

reasoning behind denoting reactions '2a' and '2b' or '3a' and '3b' are not given and are, frankly, very distracting to keep track of given the exploration of these reactions throughout the manuscript. Further to this, there is no 'reaction 8' presented. Is there one that is used in the model? If not, please revise the numbering of the reactions and consider using the shorthand notation of 'R1, R2, R3…' throughout the manuscript.

**Response:** We revised the numbering of the reactions "R1, R2, R3…" throughout the manuscript as suggested by the reviewer.

Also, HA, A, and X are not defined in reaction 3b and need to be stated somewhere in the preceding manuscript.

**Response:** We defined "HA, A, and X" as "humic acid, activation of reductive centers and oxidants, respectively" in line 53-54.

Page 3, Line 82: This is an example where more organized reaction notation will improve clarity. 'according to reaction 1' can be replaced with (R1).

**Response:** We changed the 'according to reaction 1' to "(R1)" in line 82.

Pages 3-4, Lines 104-109: This addition to the manuscript is difficult to follow. The direction and magnitude of the discrepancy in the intercomparison are more important than the correlation coefficient. Which method was biased high? How can that be rationalized by considering the methodology used and the chemistry it (or the sample matrix) can promote? For example, if the denuders were always lower, there are many instances in the literature that show nitrite is oxidized to nitrate in the condensed phase in the presence of ozone. If the denuder results are systematically lower than those from the MARGA, then a chemical loss can be hypothesized. Overall, this addition to the paper needs to draw of a key result to motivate the intercomparison in this work. As it currently stands, the reasoning is not possible to follow without reading the Stieger paper in detail.

**Response:** We rephrased this addition: "Within the cited study HONO concentrations measured by a MARGA system and an off-line batch denuder without an inlet system were compared. Although the slope between both instruments was 1.10 with slightly higher MARGA concentrations in average, both instruments biased equally in the measured concentrations resulting in a high scattering with a coefficient of determination of $R^2 = 0.41$. The probable reason was the off-line analysis of the batch denuder sample as the resulting longer interaction of gas and liquid phase during the transport led to further heterogenous reactions. As both instruments are based on the same sampling technique, the present study

could be a good starting point for an inter-comparison between MARGA and LOPAP for HONO measurements to find possible reasons in the denuder deviations."

Page 4, Line 117: The final point of this work would be better described as 'the relative importance of dew as a sink and source of HONO'.

**Response:** We modified the sentence as "the relative importance of dew as a sink and source of HONO" in line 121-122.

Page 4, Line 136: Injection of 25 mL of sample onto an IC is not possible. Clarify the volume of the collected hourly samples for the MARGA and clearly describe the volume used to quantify the atmospheric analytes (e.g. 25 uL injection loop or 10 mL treated with a preconcentration column).

**Response:** We rephrased: "Then, the aqueous samples of the WRD (gas phase) and the SJAC (particle phase) were successively injected into two ion-chromatographs (IC) with conductivity detectors (Metrohm, Switzerland) by two syringe pumps for analyzing the anions and cations. The volume of the injection loops for the anions and cations were 250 μl and 500 µl, respectively."

Page 5, Line 170: This is confusing. Should 'settled' be another word here, such as 'selected'? This is an example of unclear writing that is present throughout the manuscript.

**Response:** "settled" was changed to "selected" in line 189.

Page 6, Lines 178 and 193-195: There is a concerning contradiction here on how the dew samples were analyzed. First the Authors state that they were stored for a year, which would be highly problematic as nitrite is known to be unstable in aqueous solution. Later, the Authors state that samples were analyzed within 6 hours on the MARGA. Which is it? Clarify.

**Response:** We are sorry for this confusion. We changed the sentence in line 196-198 to: "To evaluate the HONO emission from the dew water in the morning, the dew water was collected one year later after the HONO comparison campaign and was analyzed on May 8[th], 11[th], 13[th] and 14[th] 2019."

Page 6, Line 188: How do the volumes of the blanks that were collected compare to the samples? Or was the F(NO2-) from the blanks used to correct the dew water samples? Please clarify by specifically stating how these were used towards the work-up of the dew water

dataset.

**Response:** The volume of the blanks was approximately 50 mL since we used 2 L ultrapure water to clean the plate and gutter. We added "(~ 50 mL)" in the line 207 to mention the blank volume.

We added the $NO_2^-$ concentration of blank in Table 2 in the first review, and note that " Final $NO_2^-$ = Raw $NO_2^-$ - Blank $NO_2^-$ ".

Pages 7-8, Section 3.1: This entire section needs to be revisited in light of addressing the major revision above. The Authors use misleading statements in the current revisions, such as at Line 237 where they state their results are 'in excellent agreement' with prior findings that the MARGA instrument compares terribly with another technique. This gives the wrong impression. Consider using 'this result is consistent with' instead.

**Response:** We revised the section 3.1 following the Major revision. "in excellent agreement" was changed to "this result is consistent with" in line 265.

The added section on offline batch denuders here is very confusing. The relevance to the intercomparison, and evidence to back up the length of the discourse, are vague. Why is the offline batch denuder something to consider when the second channel of the LOPAP should be a direct measure of potential interfering species from the atmospheric matrix? Why does the pH matter if the Authors conclude that it was not an issue in this work? How does this relate to the systematic bias between the MARGA and LOPAP HONO measurements?

**Response:** We are sorry for the confusing and we decided to remove this part of discussion as it was mentioned already in the introduction.

Page 8, Lines 279-280: The daytime sample population of HONO mixing ratios is not normally distributed, the Authors should consider reporting the median values here instead of the means.

**Response:** The mean values were changed to "median values of 370±300 pptv and 280±210 pptv" in line 294.

Page 9, Line 311: 'led' should be 'lead'

**Response:** "led" was changed to "lead" in line 326.

Page 10, Line 340: Why are units in the denominator of this equation?

**Response:** we modified the Eq. 2 as $F_{NO2^-} = \frac{[NO_2^-] \times V_{dew}}{S \times 1000}$. S is the surface area of the glass sampler as 1.5 m$^2$.

Page 11, Line 381: Should be 'least squares regression'

**Response:** "least linear regression" was changed to "least squares regression" in line 396.

Page 11, Lines 384-385: Give the ratio or the percentage only. No need for both.

**Response:** The percentage was deleted.

Page 12, Lines 397-399: The added statement is too speculative on a single specific mechanism. Consider stepping back from a single mechanism to explain the differences and instead emphasize the increasing need for more diverse environmental observations to understand HONO production chemistry in the nocturnal atmosphere.

**Response:** The statement of "This could be ascribed to the higher S/V surface in the rural site because of the high leaf area index (LAI, m$^2$/m$^2$) compared to an urban and might have enhanced the heterogeneous NO$_2$-HONO conversion." was changed to "The higher value may suggest that a more efficient heterogeneous conversion from NO$_2$ to HONO is present in rural sites than in urban sites." In line 413-414.

Page 12, Line 405: This equation should be inserted as all the other equations in the manuscript have been. All equation numbers through the manuscript will need to be revised following this.

**Response**: "$f$(RH)=1+a$\times$(RH/100)$^b$" was named as Eq. 3 in line 422.

Page 12, Lines 412-421: This section is a contradiction of itself between the first sentence and the last. First the Authors state that all the HONO could be produced on the aerosol, using reactive uptake coefficients consistent with the literature from lab studies, but inconsistent with observations of the full nocturnal boundary layer profiles of HONO and its precursors. The Authors then state that the correlation between [HONO]/[NO2] versus Sa is sufficient evidence that this mechanism is not important, yet the correlation is moderate in magnitude. This is a weak argument that needs to be reworked. The Authors are encouraged to visit the recent work from the group of Jochen Stutz to better place their findings and reasoning regarding aerosol-mediated NO2-to-HONO conversion into the context of our current

understanding.

**Response:** In this section, we firstly assume all the HONO could be formed on the aerosol, then an uptake coefficient of $NO_2$ on the aerosol surface $\gamma_{NO2 \rightarrow HONO\_a}$ as $2.8 \times 10^{-5}$ - $3.8 \times 10^{-4}$ was obtained in this work. We corrected this value from previous version since we made a mistake for the Sa ($dN/dlogD_p$ was wrongly used from size distribution, here we correctly use a real particle number N). And this theoretical uptake coefficient falls into a reasonable range ($10^{-6}$ - $10^{-4}$) similar to former studies which normally is regarded as unimportant for the HONO formation through heterogeneous $NO_2$ conversion on particle surfaces. This conclusion was also confirmed from a weak correlation between HONOcorr/$NO_2$ vs Sa. We modified this part as "Assuming the entire HONO formation was taking place on the particle surface, the calculated $\gamma_{NO2 \rightarrow HONO\_a}$ from the Eq. 4 varied from $2.8 \times 10^{-5}$ to $3.8 \times 10^{-4}$ with a mean value of $(1.7 \pm 1.0) \times 10^{-4}$. This theoretical uptake coefficient falls into a reasonable range of $10^{-6}$ - $10^{-4}$ similar to former studies (Kleffmann et al., 1998, Kurtenbach et al., 2001, Wong et al., 2011, VandenBoer et al., 2013). However, considering the weak correlation between $HONO_{corr}$ ($R^2=0.566$), $HONO_{corr}$/ $NO_2$ ($R^2=0.208$) and $S_a$ (Figure S6), the relative amount of HONO formed on particle surfaces might be small as previously reported (Kalberer et al., 1999, Sörgel et al., 2011, Wong et al., 2011)." in line 429-437.

Page 14, Lines 484-493: The Authors need to be careful here with their reasoning. Bulk water pH collected from a glass surface into a bottle does not directly translate into bulk water pH for dew found on soil or vegetated surfaces. The chemical nature of the material, with which the water is in contact, can influence the effective pH. A cautionary statement should be made here.

**Response:** We add a cautionary statement as "However, it should be noted that the measured pH of collected dew from the glass plate might differ compared to the pH of dew found on soil or vegetated surfaces. The chemical nature of the material, with which the water is in contact, can influence the effective pH." In line 511-514.

At Line 490 there are too many significant digits. Correct this.

**Response:** We changed the number as "42 and 165 mg L$^{-1}$" in line 507.

Page 16, Line 531: Clarity can be improved. Consider 'used are' instead of 'are referred to'

**Response:** "are referred to" was changed to "used are" in line 547.

Page 16, Line 542: Clarity can be improved. Consider 'expect that the reaction between NO

and OH' instead of 'indicate that the reaction 3a'.

**Response:** We changed "expect that the reaction between NO and OH" instead of "indicate that the reaction R7". R7 is a new number of previous reaction 3a.

Page 16, Line 544-546: This sentence is unclear. Rephrase.

**Response:** We rephrased the sentence as "However, regarding on the large uncertainty of [OH] (a factor of 2), the "unknown HONO sources" exist but could be not crucial" in line 560-561.

Page 19, Line 630: Why is mixing height spelled out here instead of using the variable 'H' in this equation?

**Response:** We corrected our Eq. 17 as:

$$[HONO] = \frac{\alpha \times S_g \times F_{NO2-}}{H \times S_g} = \frac{\alpha \times F_{NO2-}}{H} \tag{Eq. 17}$$

$F_{NO2}^-$ is the $NO_2^-$ concentration per $m^2$ of the glass sampler surface. The mean $F_{NO2-}$ from May 11th 2019 was used for the calculation. $S_g$ represents the surface area of the flat ground (analog to the surface area of the glass sampler), $\alpha$ is the enhanced factor for $V_{dew}$ (dew water sample volume of the glass sampler in Eq.2) due to the larger cold surfaces from grass which can get in contact with humid air than the flat glass sampler. $\alpha$ was calculated as $2 \times LAI$ to take the areas on both sides of the leaves and the vegetation-covered areas on the ground into account. Regarding the grass height during the dew measurements (~30cm) that is approximately the height in April 2018 and May 2019, we used a factor of 6 for LAI.

Page 20, Lines 664-665: Reference formatting is incorrect. Ensure this is to journal guidelines throughout the manuscript prior to resubmission.

**Response:** We corrected and checked all the reference format of manuscript.

Page 30, Table 2: The nitrite detection limits for the MARGA in terms of concentration need to be considered. Some of the entries here are very small and may be below the instrument detection limits. In such cases, appropriate notation should be given for this and the detection limit given in the footnotes.

**Response:** We added the notation "[a] note that the blank $NO_2^-$ concentration is lower than the detection limit of MARGA 0.0.02 μg $L^{-1}$." for Table 2. The detection limit is 0.0.02 μg $L^{-1}$. Please add as footnote!

Page 35, Figure 3: The scales for the different axes overlap on the left and right sides of the figure and space between them should be added. The particulate nitrite measurements look like they are often below the detection limit and either the LOD needs to be depicted on that figure or those points should have different symbols/be excluded. Why is the presence of PM10 nitrite not discussed in the manuscript, particularly for the intercomparison? Could deposition of this PM10 nitrite in the inlet lines or denuder contribute to the systematic difference between the MARGA and LOPAP observations?

**Response:** The Figure 3 was improved as suggested by the reviewer.

A Steam-Jet Aerosol Collector (SJAC) is used in the MARGA system, which uses 100 ℃ hot water steam forming droplets. On the droplets, different reactions of $NO_2$ can form nitrite as a function of temperature (e.g. $NO_2$ + organics, Gutzwiller et al.). Thus, we used our aerosol nitrite MARGA data with caution and present them in Figure 3. In addition, Figure 3 shows that particulate $NO_2^-$ is mainly observed during high HONO concentrations indicating a HONO breakthrough towards the SJAC. Assuming that measured $NO_2^-$ is measured HONO within the SJAC, the ratio between measured $NO_2^-$ and HONO concentrations was 1.8 % in the present study. This is low compared with systematic difference between the MARGA and LOPAP observations. A sum of WRD and SJAC $NO_2^-$ concentrations would further increase the MARGA HONO concentration leading stronger deviations. Because of likely interferences, we decided to remove the $NO_2^-$ values from Figure 3.

Page 38, Figure 6: The note on the reactions should be removed from the caption after re-assigning appropriate reaction numbers throughout the manuscript. Then R1, R3, or R9 in the figure legend are easily cross-referenced and several lines of caption are no longer necessary.

**Response:** The notes on the reactions have been removed from the caption and we also improved the Figure legend.

The figure and associated legend are too complicated and can be easily simplified The Authors state in the discussion that their base case is Model 3, and therefore, should not depict Models 1 or 2 on this figure. The composition of 'Model 3' should be given in the caption and then only the additional terms would need to be stated for the subsequent model runs. The runs, as they currently are described, are confusing.

**Response:** In the discussion part, we mention that Model 1 and Model 2 are used to discuss the HONO contribution from the gas-phase reaction of NO with OH radical and HONO deposition on the ground surface independent on RH, respectively. The base case for Model 4, 5 and 6 is Model 3. Hence, we kept all the Models (1, 2, 3, 4, 5 and 6) in the Figure and

simplified the legend by note them in the caption as "Pss presents model results by assuming an instantaneous photo-equilibrium between the gas-phase formation (R7) and gas-phase loss processes (R1 and R11) of HONO; Model 1 includes R1+R7+R11. Model 2 includes R1+R2+R7+R11+surface deposition (00:00-00:00), whereas Model 3 describes R1+R2+R7+R11+surface deposition (17:00-8:00). And Model 3 is used to be the base to investigate the effect of R9 (Model 4), R5 (Model 5) and the combination of R5+R9+Dew HONO emission (4:30-7:00) (Model 6).".

Panel a from this figure is not necessary. It can either be removed entirely or relocated to the SI.

**Response:** We removed the Panel a.

Page 39, Figure 7: The terms on the axis labels for this figure are confusing and should be revised. It is not clear why 'internal time' is necessary to retain, when UTC can be used to better effect.

**Response:** As we can see in Figure 7 and also Figure 2, the time of HONO morning peak was little different with each day. Hence we defined the relationship between temporary HONO emission from dew water and decreasing RH with internal time not UTC in Eq. 15. Here we improved the Figure 7 by moving UTC to the bottom.

Supporting Information, Page 2, Figure S1: The three photos should be denoted as panels a, b, and c. The inlet photo should be panel a, followed by the M1 setup in panel b, and then the M2 setup in panel c. Delete the 'Figure S1a' and 'Figure S1b' from below the photos.

**Response:** We improved the Figure S1 following the reviewer's suggestion.

Supporting Information, Page 5, Figure S4: Fix axis overlap to improve clarity.

**Response:** We improved the Figure S4.

Supporting Information, Page 9, Figure S8: What range of dates does the depicted data originate from?

**Response:** We mentioned the dates "during the campaign period (April 19[th] to 29[th], 2018)" in the caption.

Supporting Information, Page 12, Figure S10: This figure is very confusing to follow, mostly

due to the presentation of panel b. Why is it noted before panel a? Why is RH on the horizontal axis? Fix this. Place the panel b to the right panel a, and depict HONO_unknown increasing as a function of RH if anything. Since this plot includes all the different days of the observations, it is very hard to see the utility of it. Presumably, this is to show that rapid humidity changes result in the release of HONO. Either a case study showing this from a single day should be given, or a more robust analysis performed to demonstrate this phenomenon.

**Response:** We improved the Figure S10 as suggested by the reviewer.

Supporting Information, Page 13, Figure S11: Delete the 'Note' since the reaction notation throughout the manuscript should be fixed anyways, making this no longer necessary.

**Response:** We deleted the note and also modified the legend.

[revised manuscript text omitted]

---

## Author Response (AR3)

Leibniz Institute for
Tropospheric Research

**Prof. Hartmut Herrmann**
Head of TROPOS Atmospheric
Chemistry Department
herrmann@tropos.de
phon: +49 341 2717 7024
fax: +49 341 2717 99 7023
Permoserstraße 15
04318 Leipzig
27.08.2020

Leibniz-Institut für Troposphärenforschung  Permoserstraße 15  D-04318 Leipzig

The Editor
Atmospheric Chemistry and Physics

**Submission of revision for the Atmospheric Chemistry and Physics "Minor Revision" manuscript 'Role of the dew water on the ground surface in HONO distribution: a case measurement in Melpitz' (MS No.: acp-2019-1088) by by Yangang Ren, Bastian Stieger, Gerald Spindler, Benoit Grosselin, Abdelwahid Mellouki, Thomas Tuch, Alfred Wiedensohler and Hartmut Herrmann**

Dear Editor,

Please find attached here our response to the reviewer comments for the manuscript mentioned above together with its revised versions of the manuscript and supplement. We would like to thank both the editor and reviewer for all of their valuable and insightful comments to improve the manuscript. We have carefully considered all the reviewer comments and revised the manuscript accordingly. Below, we provide responses to the comments in blue, with changes made in green in the manuscript.

Sincerely yours,

Prof. Dr. H. Herrmann
Professor of Atmospheric Chemistry
Head of TROPOS Atmospheric Chemistry Department

Leibniz Institute for Tropospheric Research
Phone: +49 341 235-3210
Fax: +49 341 235-2139
info@tropos.de
http://www.tropos.de

Commerzbank Leipzig
Account No: 102 14 50
Sort Code: 860 400 00
IBAN: DE77 8604 0000 0102 1450 00
SWIFT CODE: COBADEFF 860

Mitglied der
Leibniz-Gemeinschaft

[Figure]

The authors gratefully thank the editor for the comments and suggestions. We have revised our manuscript according to the reviewer's suggestions and comments. **All the changes and responses to the reviewers' comments are listed below point-by-point in blue according to a new line numbering in the revised manuscript. The major changes are highlighted with green in the revised manuscript.**

**Editor Decision: Publish subject to minor revisions (review by editor)** (17 Aug 2020) by Hang Su

Comments to the Author:

Dear authors,

Please find enclosed some additional comments from the reviewer. There are still some technical issues to be answered/clarified. Please revise the manuscript accordingly.

For the reviewer's concern about S/V calculations, I'd suggest you to include complementary information such as PM2.5 concentration and aerosol number concentration at both your sites and the Beijing site that you were referring to. A comparison of these numbers will help to justify your calculations.

**Response:** We thank the editor for the suggestion. We should mention that the reported $S_a$ values of $(5.1\text{-}9.9) \times 10^{-4}$ $m^2$ $m^{-3}$ and RH corrected value of $(0.6\text{-}1.9) \times 10^{-3}$ $m^2$ $m^{-3}$ correspond to the night-time period 18:00-22:00 (UTC) since only nighttime data was used to calculate the HONO formation through heterogeneous conversion of $NO_2$. As shown in Figure R1 (we also add it in SI as Figure S5b), the $S_a$ values ranged from $4.2 \times 10^{-5}$ to $9.9 \times 10^{-4}$ $m^2$ $m^{-3}$ for whole day period during our field measurement period of April $19^{th}$ -$29^{th}$ 2018, this value is one magnitude lower than $S_a$ value of $0.2\text{-}3.4 \times 10^{-3}$ $m^2$ $m^{-3}$ in Beijing (Liu et al., 2014;Wang et al., 2016). However, due to the high RH (RH ~100% during nighttime in Figure R1), the $S_a$ values would be strongly enhanced by the RH correction (Figure R1) to be $(0.5\text{-}1.9) \times 10^{-3}$ $m^2$ $m^{-3}$.

In addition, we also plot 30 mins averaged $PM_{2.5}$ concentration in Figure R2, which ranged from 2 μg $m^{-3}$ to 36 μg $m^{-3}$ but could increase to ~257 μg $m^{-3}$ for some nighttime period (e.g. 21:00-22:00 UTC April $19^{th}$ 2018), according to our calculated $S_a$ values of $(0.4\text{-}9.9) \times 10^{-4}$ $m^2$ $m^{-3}$. Exactly, Wang et al. (2016) reported $S_a$ value of $1.3 \times 10^{-3}$ $m^2$ $m^{-3}$ and $3.4 \times 10^{-3}$ $m^2$ $m^{-3}$, respectively, for a clean period ($PM_{2.5}$ 0.2-107 μg $m^{-3}$) and a polluted period ($PM_{2.5}$ 74-192 μg $m^{-3}$) in the campaign Beijing 2015. Cai et al. (2017) reported a $S_a$ value of $2 \times 10^{-4}$ $m^2$ $m^{-3}$ according to $PM_{2.5}$ mass concentrations of 30 μg $m^{-3}$ during Beijing 2016. Hence, our calculated $S_a$ value of $(0.4\text{-}9.9) \times 10^{-4}$ $m^2$ $m^{-3}$ is reasonable.

[Figure]

**Figure R1.** The particle surface density $S_a$ and RH corrected $S_a$ value calculated from the particle size distribution for our field measurement period of April 19th-29th 2018.

[Figure]

**Figure R2.** Time series of $PM_{2.5}$ for our field measurement period of April 19th-29th 2018.

Another question is about the calculation of dew volume. You seem to consider the dew formation as a kinetic-limited process and thus assume a dew volume proportional to the condensable surface area. Is this well-established/accepted? Otherwise, if dew formation is controlled by the thermodynamic equilibrium, it will not depend on the surface areas, but rather the difference between atmospheric temperature and the dew point temperature.

**Response:** As generally well known, dew occurs when the surface temperature is lower than or equal to the dew-point temperature, and water vapor from the air in contact with the cold surface condenses to form dew (Vuollekoski et al., 2015;Agam and Berliner, 2006). Exactly, the actual amount of dew in a specific place is strongly dependent on temperature and wetting properties of the surface. These two parameters control the nucleation rate and the latter has in addition major consequences on the form and growth of the droplet pattern. (Nilsson et al., 1994;Beysens, 1995). Beysens (1995) also mention that the wetting properties of a surface

can be easily modified by surface treatments. Recently, the study of Kotzen (2015) clearly indicated that main ways to maximize dew formation and collection is to increase the surface area of the collector and facilitating maximum radiative cooling to the open sky on the surface. Concerning on this work, we used one 1.5 m$^2$ glass sampler. Firstly, this artificial material could be less effective in collecting dew than grasses leaves since nature also tells us that multiple surfaces as well as smaller surfaces may be more effective in collecting dew. (Kotzen, 2015). Secondly this flat 1.5 m$^2$ glass sampler could not represent the realistic surface area of grasses leaves on the ground, where the surface area of the grasses leaves really control the dew volume around Melpitz station. Wentworth et al. (2016) and Groh et al. (2018) indicated that the dew yield of artificial devices used to collect dew would differ from that of dynamic and heterogeneous natural land surface coverages. Hence, in this work, an enhanced factor was calculated as 2×LAI to take the areas on both sides of the leaves and the vegetation-covered areas on the ground into account. In Wohlfahrt et al. (2001), the LAI for meadows with different grass heights are given. Regarding on the possibly different grass height during the HONO field measurement and dew measurements in April 2018 and May 2019, respectively, we would use a range of 1-6 for LAI in this version.

**References**

Agam, N., and Berliner, P. R.: Dew formation and water vapor adsorption in semi-arid environments—A review, Journal of Arid Environments, 65, 572-590, https://doi.org/10.1016/j.jaridenv.2005.09.004, 2006.

Beysens, D.: The formation of dew, Atmospheric Research, 39, 215-237, https://doi.org/10.1016/0169-8095(95)00015-J, 1995.

Cai, R., Yang, D., Fu, Y., Wang, X., Li, X., Ma, Y., Hao, J., Zheng, J., and Jiang, J.: Aerosol surface area concentration: a governing factor in new particle formation in Beijing, Atmos. Chem. Phys., 17, 12327-12340, 10.5194/acp-17-12327-2017, 2017.

Groh, J., Slawitsch, V., Herndl, M., Graf, A., Vereecken, H., and Putz, T.: Determining dew and hoar frost formation for a low mountain range and alpine grassland site by weighable lysimeter, Journal of Hydrology, 563, 372-381, 10.1016/j.jhydrol.2018.06.009, 2018.

Kotzen, B.: Innovation and evolution of forms and materials for maximising dew collection, Ecocycles, 1, 39-50, 10.19040/ecocycles.v1i1.33, 2015.

Liu, Z., Wang, Y., Costabile, F., Amoroso, A., Zhao, C., Huey, L. G., Stickel, R., Liao, J., and Zhu, T.: Evidence of Aerosols as a Media for Rapid Daytime HONO Production over China, Environmental Science & Technology, 48, 14386-14391, 10.1021/es504163z, 2014.

Nilsson, T. M. J., Vargas, W. E., Niklasson, G. A., and Granqvist, C. G.: Condensation of water by radiative cooling, Renewable Energy, 5, 310-317, https://doi.org/10.1016/0960-1481(94)90388-3, 1994.

Vuollekoski, H., Vogt, M., Sinclair, V. A., Duplissy, J., Järvinen, H., Kyrö, E. M., Makkonen, R., Petäjä, T., Prisle, N. L., Räisänen, P., Sipilä, M., Ylhäisi, J., and Kulmala, M.: Estimates of global dew collection potential on artificial surfaces, Hydrol. Earth Syst. Sci., 19, 601-613, 10.5194/hess-19-601-2015, 2015.

Wang, G., Zhang, R., Gomez, M. E., Yang, L., Levy Zamora, M., Hu, M., Lin, Y., Peng, J., Guo, S., Meng, J., Li, J., Cheng, C., Hu, T., Ren, Y., Wang, Y., Gao, J., Cao, J., An, Z., Zhou, W., Li, G., Wang, J., Tian, P., Marrero-Ortiz, W., Secrest, J., Du, Z., Zheng, J., Shang, D., Zeng, L., Shao, M., Wang, W., Huang, Y., Wang, Y., Zhu, Y., Li, Y., Hu, J., Pan, B., Cai, L., Cheng, Y., Ji, Y., Zhang, F., Rosenfeld, D., Liss, P. S., Duce, R. A., Kolb, C. E., and Molina, M. J.: Persistent sulfate formation from London Fog to Chinese haze, Proceedings of the National Academy of Sciences, 113, 13630-13635, 10.1073/pnas.1616540113, 2016.

Wentworth, G. R., Murphy, J. G., Benedict, K. B., Bangs, E. J., and Collett Jr., J. L.: The role of dew as a night-time reservoir and morning source for atmospheric ammonia, Atmos. Chem. Phys., 16, 7435-7449, 10.5194/acp-16-7435-2016, 2016.

Wohlfahrt, G., Sapinsky, S., Tappeiner, U., and Cernusca, A.: Estimation of plant area index of grasslands from measurements of canopy radiation profiles, Agric. For. Meteorol., 109, 1-12, https://doi.org/10.1016/S0168-1923(01)00259-3, 2001.

The authors gratefully thank the reviewer for the comments and suggestions. We have revised our manuscript according to the reviewer's suggestions and comments. **All the changes and responses to the reviewers' comments are listed below point-by-point in blue according to a new line numbering in the revised manuscript. The major changes are highlighted with green in the revised manuscript.**

Comments to the revised manuscript by Ren et al.

In their second revised manuscript Ren et al. considered most concerns of the two reviewers. Find below further comments, first to their answers to the reviewer's comments and then to the revised manuscript and the SI.

Some answers were not convincing for me (see below) – here the editor should find a final decision. In contrast several minor points could be easily corrected in a final version. However, caused by the significant amount of text (answers, manuscript, SI) and this third correction, I do not want to see a further version again.

Major Concerns (see answers to the reviewer's comments):

Reviewer #1:

1) Calculated NO2 uptake coefficients for particles surfaces:

In the revised version the authors corrected the particle surface area leading to more reasonable theoretical uptake coefficient if all HONO formation would take place on the particles. However first, the range of S/V in the very clean environment of Melpitz (see lines 286-292 main text…) is still in the same range than values published for one of the most polluted environments of the word, i.e. Beijing. This is highly unreasonable. Please check again and at least, give any explanation for these high values (sand storms?). Lower S/V would further increase the theoretical $NO_2$ uptake coefficient.

**Response:** As the reviewer suggested, we checked the measured particle size distribution and recalculated the particle surface density $S_a$ again, no error was found for the $S_a$ calculation. We should mention that the reported $S_a$ values of $(5.1-9.9) \times 10^{-4}$ $m^2$ $m^{-3}$ and RH corrected value of $(0.6-1.9) \times 10^{-3}$ $m^2$ $m^{-3}$ in lines 417 and 420 (previous version) correspond only the night-time period 18:00-22:00 (UTC) since only nighttime data was used to calculate the HONO formation through heterogeneous conversion of $NO_2$. As shown in Figure R1 (we also add it in SI as Figure S5b), the $S_a$ values ranged from $4.2 \times 10^{-5}$ to $9.9 \times 10^{-4}$ $m^2$ $m^{-3}$ for whole day period during our field measurement period of April $19^{th}$ -$29^{th}$ 2018. This value is one order magnitude lower than $S_a$ value of $0.2-3.4 \times 10^{-3}$ $m^2$ $m^{-3}$ in Beijing (Liu et al., 2014;Wang et al., 2016). However, due to the high RH (RH ~100% during nighttime in Figure R1), the $S_a$ values would be strongly enhanced by the RH correction (Figure R1) to be $(0.5-1.9) \times 10^{-3}$ $m^2$ $m^{-3}$. For the clarity, we improve the sentence as "The particle surface

density $S_a$ was calculated as (0.4-9.9) $\times 10^{-4}$ m$^2$ m$^{-3}$ from the particle size distribution (Figure S5a) ranged from 5 nm to 10 μm of APSS and D-MPSS data by assuming the particle are in spherical shape for the whole day period of April 19$^{th}$-29$^{th}$ 2018. Due to the high RH (RH ~100% during nighttime in Figure S5b), the particle surface density $S_a$ would be strongly enhanced (by one order magnitude) by the RH correction to be (0.5-1.9) $\times 10^{-3}$ m$^2$ m$^{-3}$ with a hygroscopic factor $f$(RH) following the method of Li et al. (2012) and Liu et al. (2008)" in line 416-421."

[Figure]

Figure R1. The particle surface density $S_a$ and RH corrected $S_a$ value calculated from the particle size distribution for our field measurement period of April 19$^{th}$-29$^{th}$ 2018.

Second, still the authors do not well describe (or understand?) the main conclusion from this section (see their answer. "This theoretical uptake coefficient falls into a reasonable range of…"), which becomes even more obvious considering the answer to the same point for reviewer 2 ("… similar to former studies, which normally is regarded as unimportant for the HONO formation…"). If the uptake were as high (>10^-4), than HONO would be formed on the particles and this mechanism would be certainly important!

The correct interpretation is however:
The average uptake of NO2 in the dark which the authors now calculate is 1.7x10^-4. This number is 2-3 orders of magnitude higher than typical uptake coefficients determined in the lab for the uptake of NO2 in the dark (!) on different substrates (e.g. R2/Teflon/glass/NaCl/TiO2/etc.: some 10^-7 to 10^-8, R4/soot/SS: 10^-6 to <10^-8, R4/phenols: ca. 10^-6, …). There is no study were a dark (!) uptake of 10^-4 was determined, except for fresh soot for the first seconds of reaction (later passivation, s. above). Thus, the calculations given by the authors (and the theoretical uptake would be even higher for realistic S/V, see above) clearly shows that formation on particles is not important, in agreement with their own conclusions in the manuscript. This should be made clear now.

**Response:** We thank the reviewer for the correction. We have replaced the sentence:

"the calculated $\gamma_{NO2 \rightarrow HONO\_a}$ from the Eq. 4 varied from $2.8 \times 10^{-5}$ to $3.8 \times 10^{-4}$ with a mean value of $(1.7 \pm 1.0) \times 10^{-4}$. This theoretical uptake coefficient falls into a reasonable range of $10^{-6}$ - $10^{-4}$ similar to former studies (Kleffmann et al., 1998;Kurtenbach et al., 2001;Wong et al., 2011;VandenBoer et al., 2013). However, considering the weak correlations between $HONO_{corr}$ ($R^2$=0.566), $HONO_{corr}$/ $NO_2$ ($R^2$=0.208) and $S_a$ (Figure S6), the relative amount of HONO formed on particle surfaces might be small as previously reported (Wong et al., 2011;Sörgel et al., 2011;Kalberer et al., 1999)." by:

"the calculated $\gamma_{NO2 \rightarrow HONO\_a}$ from the Eq. 4 varied from $2.8 \times 10^{-5}$ to $3.8 \times 10^{-4}$ with a mean value of $(1.7 \pm 1.0) \times 10^{-4}$. This number is 2-3 orders of magnitude higher than typical uptake coefficients determined in the lab for the uptake of $NO_2$ in the dark on different substrates, e.g. Teflon/glass/NaCl/$TiO_2$/soot/Phenol etc: $10^{-6}$ to <$10^{-8}$ (Kleffmann et al., 1998;Kurtenbach et al., 2001;Ammann et al., 1998;Gutzwiller et al., 2002). Thus, this theoretical uptake coefficient clearly shows that formation on particles is not important. In addition, the weak correlations between $HONO_{corr}$ ($R^2$=0.566), $HONO_{corr}$/ $NO_2$ ($R^2$=0.208) and $S_a$ (Figure S6) confirm that the HONO formed on particle surfaces could be unimportant as previously reported (Wong et al., 2011;Sörgel et al., 2011;Kalberer et al., 1999)." in line 431-440.

2) Explanation of the morning HONO peaks by dew water nitrite.

I still cannot follow the use of the enhancement factor for the dew nitrite. First, I do not think that the dew volume follows perfect linearly the available surface area on the ground and I still think the volume of humid air getting into contact will also control the dew formation. This is an important point which should be verified by other studies since a correction of the amount of nitrite (=> HONO morning peak) is increased by a factor of 12!

**Response:** Firstly we agree with the reviewer that the volume of humid air getting into contact will also control the dew formation. But more dew could be formed if there is more cold surface area in contact with certain amount of humid air, this has been well described by Kotzen (2015). In other words, we could collect more dew (larger volume of dew ($V_{dew}$)) if we used for example a dew sampler with corrugations on one side than a flat dew sampler. This is also the case of the nature, larger area of leaves of grasses than, for example, 1 $m^2$ of ground surface could strongly enhance the dew formation. Hence, the $V_{dew}$ obtained from the collector is not necessarily representative of $V_{dew}$ that forms naturally on the grassland canopy because of their different cold contacted surface area with humid air (Wentworth et al., 2016;Groh et al., 2018).

Secondly, we also agree with the reviewer that dew volume cannot follow perfect linearly the available surface area on the ground. Hence in this version, we assumed an enhanced factor of

2×LAI as 1-6 to take the areas on both sides of the leaves and the vegetation-covered areas on the ground into account. We strongly point out here, that this enhanced factor is an assumption. The collection of dew is increasingly focused in several studies within the last years (Jacobs et al., 1999;Takenaka et al., 2003;del Campo et al., 2006;Ye et al., 2007;Guan et al., 2014;Kotzen, 2015;Vuollekoski et al., 2015;Groh et al., 2018) as dew is expected to form a natural reservoir of reactive tropospheric species at night. An exact evaluation or comparison of the volume between an artificial dew collector and the "real" dew on grasses was not performed yet. This is an important question that has to be answered in future studies.

The discussion in Section 4.3.3 was changed to "…α was calculated as 2×LAI to take the areas on both sides of the leaves and the vegetation-covered areas on the ground into account. And a factor of 6 for LAI was assumed and used in section 4.2.2. However, regarding the possibly different grass height during the HONO field measurement and dew measurements in April 2018 and May 2019, respectively, we would use a range of 1-6 for LAI in this section. During the HONO peak at 6 or 7 UTC, the mixing height ranged between 175 m and 600 m, while the mixing height ranged from 20 m to 200 m at 0:00 – 5:00 UTC. Hence, the overall concentration increase from this source would be 377-2264, 189-1132, 76-122, 38-226 and 13-76 pptv, if all of the deposited HONO is released into the overlying air column for a mixing height of 20, 40, 100, 200 and 600 m, respectively. Since the released HONO was subjected to photolysis, using a $J_{HONO}$ from TUV model scaled by global radiation (section 2.7), a maximum [HONO] of 176-1053, 88-527, 35-211, 18-105 and 6-35 pptv for the mixing height 20, 40, 100, 200 and 600 m, respectively, would be contributed from the surface nitrite release at 7:00 UTC after the process started from 4:00 UTC. For a reasonable 100 m mixing height, this would account for 5-30% of the observed HONO morning peak in Figure 6. This low percentage might be a result of the different sampling time of dew measurement compared with HONO measurement and further studies are required for the exact quantification. …"

Second, even if the authors are right with the increasing dew volume on grass (x12) than for increasing volume of the dew water the resulting liquid concentration of nitrite would be 12 times lower than on the glass substrates used, since the amount of HONO in the mixing layer will control the soluble nitrite (see the assumptions by the authors using the H in their equations…). Thus, the amount of dew nitrite is for sure strongly overestimated.

**Response:** Since the surface area of our glass sampler is only 1.5 m², which is much smaller than the grass area around the Melpitz station and cannot affect the bulk volume of dew water on the grass around the Melpitz station. Hence, our measured liquid concentration of nitrite on the collector can represent the nitrite concentration in dew water on the grass. But $V_{dew}$ obtained from the collector is not necessarily representative of $V_{dew}$ that forms naturally on

the grassland canopy because of their different cold contacted surface area with humid air (Wentworth et al., 2016). Hence, in this version, an enhanced factor of $2\times$LAI with LAI values in the range of 1-6 would help to represent the $V_{dew}$ on the grassland canopy from the measure $V_{dew}$ on the collector. However, according to the observed grass height during the campaign, a LAI value of 6 would be preferred regarding the publication of Wohlfahrt et al. (2001).

Now even with this 12 times too high nitrite, considering HONO photolysis (this is done very well now!) and using a reasonable mixing height of H=100 m in the morning (two hours after sunrise!) only 30 % of the HONO morning peaks observed can be explained and this would be <3 % for a reasonable amount nitrite in dew!

So although I can follow qualitatively the author's explanation of the experimental data (peak HONO when rh decrease…) the quantitative explanation is not valid. And the main point is most probably that the authors compare data of two different years (apples and oranges…).

**Response:** We agree with the reviewer that the main point for this low value (5-30% in this version) is most probably that we compared data of two different years, and we already mention it in line 661-663 "This low percentage might be a result of the different sampling time of dew measurement compared with HONO measurement." We should although mention that this is a first assumption with qualitative results and further studies are required for the exact quantification.

Reviewer #2:

1) Details to the MARGA:

In their answer to the accuracy of the MARGA the authors specify:
"For the accuracy of the ion chromatography system, liquid NO2- standards were twice injected to the MARGA with concentrations of 70, 120 and 150 µg L-1. The resulting slope of 1.13 (R2 = 0.99) indicates slightly lower measured NO2- concentrations …"
I do not understand that section? Was the raw data of the IC of the two injections plotted against each other (70/70, 120/120, 150/150) and the slope was 1.13? In this case the precision (!) is only 13 %!? Or was nitrite injected into the rotated denuder under zero air (complete system check…) and compared with the direct calibration of the IC only (I expect the latter, but the first is mentioned in the text…).

**Response:** We are sorry for this confusion. We tested the ion chromatography by injecting standard solutions with defined $NO_2^-$ concentrations of 70, 120 and 150 µg L. The correlation

between both the predefined concentrations and the measured concentrations by the MARGA IC resulted in a slope of 1.13. The MARGA is internally calibrated by an LiBr solution. Hence, no calibration solutions were needed for the MARGA measurements. We will improve the description in the main manuscript as follows:

"To test the robustness of the ion chromatography within the MARGA, standard solutions with defined $NO_2^-$ concentrations of 70, 120 and 150 μg $L^{-1}$ were injected in the IC system. The correlation between both the predefined concentrations within the standard solutions and the measured concentrations by the MARGA IC resulted in a slope of 1.13. This value…" in line 157-161.

Than for the blank tests the authors mention that the first two hours of the samples were not considered because of "still … residual concentrations". Here the authors should give a true response time of the instrument which is than clearly longer than the one hours sample time. This would also help to understand some differences in the timing of the signals of the two instruments mentioned by reviewer#2 below. If the true response time is e.g. two hours, than use the same averaging for the LOPAP when comparing both instruments (this may further decrease the noise of the intercomparsion data…).

**Response:** The response time of both instruments is fast. The MARGA output gives high concentrations within the same hour. However, it is expected that the residual concentrations of the previous hour, which might be in maximum 10 %, could be added to the real atmospheric concentrations of the current hour. This would lead to less sharp concentrations courses.

We tested the correlation between MARGA and LOPAP for both measurement periods and shifted the hourly concentrations of the MARGA one and two hours back and forward regarding the LOPAP concentrations. The high correlation ($R^2$) described in Figure 1 decreased strongly to values below 0.5 when the MARGA concentrations were shifted one hour and decreased below 0.3 for a shift of two hours.

Hence, the correlation described in Figure 1 and in the main manuscript gives the best results.

2) Details to the LOPAP:

Typically the LOPAP instrument has a time response of 5 min. In this case the authors cannot specify a DL for a time resolution of 30 s (one data point collected?)!
In addition, I cannot understand why the authors obtained a detection limit of only 0.1 ppt using the zero noise of 30 min data (2 sigma or 3 sigma?). The lowest DL of the instrument which I saw was 0.2 ppt (see line 92) by the group who developed the instrument and using extreme experimental conditions (very long optical path length, high gas flow etc., see

Kleffmann and Wiesen, 2008). The authors should explain how they obtained this outstanding DL (or correct it to the typical 1-2 ppt, I am sure this is again an error…).

**Response:** We thank the reviewer for the correction. The detection limit was correct in line 181-182 "The detection limit of LOPAP was approximately 1-2 pptv with a response time of 5 min."

3) New weighted regression of the intercomparison data:

While the slope of the intercomparison for M1 did not significantly change, the slope for M2 is now only 1.66 und not 1.9 from the former versions, which is significant!? Is that only caused by a weighted/non-weighted regression? Both data sets correlate very well with no large outliers. Thus even if outliers have larger relative errors than other points (and for me the relative errors (%) look the same for all data, in this case a weighted is similar to a non-weighted orthogonal regression…) than the slope should not change too much (see M1). Please check the data again.

**Response:**  As the reviewer suggested, we carefully checked the data. We again performed the calculation of the unweighted (values in the first manuscript version) and the weighted (values in the revised manuscript version) Deming regression. In case of the unweighted version, we again received for M1 a slope of 1.58±0.08 and an intercept of 29±33 pptv. For period M2, the slope and intercept were 1.90±0.07 and 120±16 pptv, respectively. For the weighted case, the slopes increased to 1.71±0.08 and 2.17±0.09 for M1 and M2, respectively. The intercept was -15±21 and 79±10 pptv for M1 and M2, respectively. The correlation in both periods were significant (p < 0.001). We are sorry for the wrong values in the last version, and the right values were added in line 263-265 "The comparisons of the MARGA and LOPAP HONO measurements for period M1 and period M2 in Figure 1c result in slopes of 1.71 and 2.17 using error weighted Deming regression, respectively." and 272-273 "… which could account for ca. 71% (M1, where both LOPAP and MARGA used the common MARGA inlet) of these ca. 117% of overestimated HONO measurement from MARGA. Additional artefacts such as heterogeneous formation of HONO due to the long MARGA inlet system should be responsible for another ca. 46% (the difference between slopes M2 and M1). …"

Concerns to the revised manuscript:

The following concerns are listed in the order how they appear in the manuscript.

Lines 25-27: The authors should correct this number to 30 % (or <3%, see major concern), see their calculations to the dew nitrite…

**Response:** We should mention that the "percentage of 90-100%" in line 26 was obtained from the chemical model. As the reviewer mentioned above, the number of 5-30% is most probably due to the fact that we compared data of two different years, further field measurement will be planned.

Line 75-76: Just a comment (no correction), yes I agree, but in this study there is also no simultaneous data…

**Response:** The sentence of "Few of them have simultaneously quantified both dew and atmospheric composition." was removed.

Line 158-162: see above

**Response:** We changed the text according to our answer in the major comments.

Line 181: see above

**Response:** This part was corrected to "The detection limit of LOPAP was approximately 1-2 pptv with a response time of 5 min." in line 181-182.

Lines 254-255: No, if you consider the 1 h data, also the peak values of the LOPAP are lower. And it is trivial that the high resolution data peaks of the LOPAP are higher! In Fig 1a) apples (30 s data LOPAP) and oranges (1h data Marga) are compared!

**Response:** The Figure 1a used to show the original measured data (not averaged) for both instruments: LOPAP and MARGA, and the high time resolution data of LOPAP could present us more clearly time evolution of HONO in the early morning rather than MARGA. We corrected the sentence of "It indicates that the MARGA values were higher than the values of LOPAP but not during the peak events" to "It indicates that the MARGA values were higher than the values of LOPAP".

Line 255-261: check for the lower time response of the Marga and correct text accordingly (when averaging the LOPAP data for two hours, most probably the differences disappear…).

**Response:** An averaging of the LOPAP and MARGA values for two hours resulted in low deviation in the regression lines.

M1: y=(1.59±0.09)x+(27±34) ← unweighted        ($R^2 = 0.945$)

M1: y=(1.69±0.08)x+(-7±21) ← weighted

M2: y=(1.99±0.10)x+(103±20) ← unweighted      ($R^2 = 0.818$)

M2: y=(2.18±0.08)x+(74±21) ← weighted

Hence, the response time for both instruments is reasonable. We corrected the values in lines 263-265 "The comparisons of the MARGA and LOPAP HONO measurements for period M1 and period M2 in Figure 1c result in slopes of 1.71 and 2.17 using error weighted Deming regression, respectively."

Line 264: see above

**Response:** We changed the values according to our answer in the major comment.

Line 275: this was more than 30 % in the former versions. Conclusions would change: if the data is correct (s. above?) than formation in the inlet is not too significant and in between the combined errors…

**Response:** The corrected value of 46% was added in lines 273-275 "Additional artefacts such as heterogeneous formation of HONO due to the long MARGA inlet system should be responsible for another ca. 46% (the difference between slopes M2 and M1)."

Line 290 and 294: Give either the average or the median for both…

**Response:** The HONO average value of 162±96 pptv and 254±114 pptv during daytime and nighttime, respectively, were provided in line 294.

Lines 310-327: The different cases a-c are not very clear here. I would first sum up the three cases in short sentences (see lines 570-573) and then explain in more detail…

**Response:** This section was changed to "Such daytime pattern was also found in Spain, for a site surround by forests and sandy soils (Sörgel et al., 2011). Sörgel et al. (2011) explained this by local emissions, which are trapped in the stable boundary layer before its breakup of the inversion in the morning based on a similar diurnal cycle for NO and $NO_2$, which is different with this work. In this work, the $NO_2$ mixing ratio decreased from the midnight until noon and NO peaked at 5:00 (UTC) then kept low concentration (<1 ppbv) for 18 hours of one day. However, three hypotheses could be expected to explain this HONO morning peak: hypothesis (a) of HONO morning peak might possibly be caused by the photolysis of particle-phase $HNO_3/NO_3^-$ (Zhou et al., 2003;Ye et al., 2016;Zhou et al., 2011), since as shown in Figure 4a, 4e and 4f, the early morning variation trend of HONO during daytime

was similar to the one of $NH_3$ in the gas phase as well as $NO_3^-$ and $NH_4^+$ in $PM_{10}$. Hypothesis (b), as reported by Stemmler et al. (2006), the photosensitized $NO_2$ on humic acid could act as a source of HONO during the daytime. For hypothesis (c), this morning peak of HONO has been reported for Melpitz (April 4[th]-14[th], 2008) by Acker et al. (2004), who expected that the storage of HONO on wet surfaces can be a source for observed daytime HONO. Exactly, it was observed that dew was formed overnight during our campaign in Melpitz. Gaseous HONO could be deposited in these droplets. Due to evaporation after sunrise, HONO would be reemitted in the atmosphere and lead to a HONO morning peak. These hypotheses will be further discussed in Section 4.".

Line 331: …species or by the hetero…

**Response:** "by' was added in line 330.

Line 420: see above

**Response:** This part in lines 416-421 was changed to "The particle surface density $S_a$ was calculated as (0.4-9.9) $\times 10^{-4}$ m$^2$ m$^{-3}$ from the particle size distribution (Figure S5a) ranged from 5 nm to 10 μm of APSS and D-MPSS data by assuming the particle are in spherical shape for the whole day period of April 19[th]-29[th] 2018. Due to the high RH (RH ~100% during nighttime in Figure S5b), the particle surface density $S_a$ would be strongly enhanced (one magnitude) by the RH correction to be (0.5-1.9) $\times 10^{-3}$ m$^2$ m$^{-3}$ with a hygroscopic factor $f$(RH) …" according to the major concern.

Lines 433-438: see above.

**Response:** This part in lines 433-440 was changed to "This number is 2-3 orders of magnitude higher than typical uptake coefficients determined in the lab for the uptake of $NO_2$ in the dark on different substrates, e.g. Teflon/glass/NaCl/TiO$_2$/soot/Phenol etc: $10^{-6}$ to $<10^{-8}$ (Kleffmann et al., 1998;Kurtenbach et al., 2001;Ammann et al., 1998;Gutzwiller et al., 2002). Thus, this theoretical uptake coefficient clearly shows that formation on particles is not important. In addition, the weak correlations between HONO$_{corr}$ ($R^2$=0.566), HONO$_{corr}$/ $NO_2$ ($R^2$=0.208) and $S_a$ (Figure S6) confirm that the HONO formed on particle surfaces could be unimportant as previously reported (Wong et al., 2011;Sörgel et al., 2011;Kalberer et al., 1999)." according to the major concern.

Lines 461-462: The sentence makes no sense? "…uptake coefficient …is lower than the reactive surface…" (apples and oranges…).

**Response:** The sentence of "It should be noted here that the obtained $NO_2$ uptake coefficient on the ground surface is lower than the reactive surface provided by aerosols." has been removed.

Line 481: Please check for that average number of 0.16 ppb/h. If I take the average data shown in Fig. 4 a (decreasing HONO from 20:00-4:00), I get an HONO uptake of only 0.033 ppb/h, nearly four times lower?

**Response:** The average number of 0.16 ppb/h was well confirmed. However, we should correct previous Eq.7 of $L_{HONO=}\frac{d[HONO]}{dt}$ - $k_{het} \times [NO_2]$ to "$L_{HONO=}\frac{d[HONO]}{dt}$ + $k_{het} \times [NO_2]$" for the wrong typing.

We calculated the HONO deposition rate ($L_{HONO}$) for each day of our field measurement as below in the Table R1:

Table R1, Calculated HONO deposition rate ($L_{HONO}$) during nighttime for our field measurement period of April $19^{th}$-$29^{th}$ 2018.

| Date | UTC | $L_{HONO}$ (ppbv $h^{-1}$) |
|---|---|---|
| 19/04/2018 | 19:51-4:00 | 0.45 |
| 20/04/2018 | 19:31-4:00 | 0.21 |
| 21/04/2018 | 20:31-4:00 | 0.09 |
| 22/04/2018 | 21:21-4:00 | 0.14 |
| 23/04/2018 | 18:51-4:00 | 0.04 |
| 24/04/2018 | 17:51-4:00 | 0.14 |
| 25/04/2018 | 21:21-4:00 | 0.09 |
| 26/04/2018 | 18:31-4:00 | 0.07 |
| 27/04/2018 | 22:41-4:00 | 0.11 |
| 28/04/2018 | 19:50-4:00 | 0.28 |
| Average | | 0.16±0.12 |

Eq. 8: delete the "ground" for the velocity of HONO (this is a gas kinetics number…)

**Response:** The "ground" was deleted for Eq. 8, 12, 13, 14, 15, 16 and also line 494.

Line 506: should by microgram per Liter and not mg/L!

**Response:** "mg $L^{-1}$" was corrected to "µg $L^{-1}$" in line 507.

Lines 515-530: please check again for HONO, see below SI

**Response:** We carefully checked the gas phase diffusion coefficient for HONO and explained below in SI.

Line 521-523: I do not understand the second part of the sentence, since Rb is no function of the WS, see Fig. S9.

**Response:** We are sorry for the uncorrected description, "especially when the winds speed is smaller than ~1 m s$^{-1}$." was removed.

Line 536-537: No, R2 and R7 are not very important. The dark conversion of NO2 is 1-2 orders of magnitude less efficient than the daytime conversion of NO2 (see e.g. Kleffmann et al., 2005) and typically NO+OH makes only 10 % of the daytime HONO (see same study, where OH and J(HONO) were measured...). See also own conclusion in line 566…

**Response:** We thank the reviewer for the correction, and we changed the sentence to "the R2 and R7 are not expected to be responsible for this HONO morning peak, but could contribute to the daytime HONO for the period of 10:30-16:30 (UTC)." ln line 535-536.

Line 549 and others: Typically capital letters are used for PSS (or use pss but not Pss…).

**Response:** "Pss" was corrected to "PSS" in line 548 and Figure 6.

Line 557: better refer to "model 1" (see legend of Figure 6) since the colours are not easily visible (red/orange/pink…)

**Response:** We added "Model 1" in line 556 to clarify it.

Line 625: "… respectively (see Table 4) with an average of…" You do not find the average in table 4…

**Response:** We added the average value of 0.016±0.014 pptv %$^{-1}$ s$^{-1}$ in Table 4 as below:

**Table 4.** Summary of the temporary HONO emission rate from dew water, k$_{emission}$ from April 19$^{th}$ to 29$^{th}$, 2018.

| Period | $k_{emission}$ (pptv %$^{-1}$ s$^{-1}$) | |
| --- | --- | --- |
| | Min | Max |
| 21/4/2018 | 0.0054 | 0.0357 |
| 22/4/2018 | 0.0048 | 0.0314 |
| 24/4/2018 | 0.0057 | 0.0192 |

| | | |
|---|---|---|
| 26/4/2018 | 0.0067 | 0.0302 |
| 27/4/2018 | 0.0048 | 0.0215 |
| 28/4/2018 | 0.0079 | 0.017 |
| mean | 0.006±0.001 | 0.026±0.008 |
| Total average | 0.016±0.014 | |

Lines 632-635 and Figure S11: Please check the calculations for the higher k(emission) of 0.026. The peak should be higher than the experimental peak and not lower (in Fig. S11), since an excellent agreement is obtained using the average k(emission) of 0.016 (see Figure 6)!?

**Response:** We thank the reviewer for the correction, we corrected the Figure S11 as below.

[Figure]

**Figure S11.** Observed average HONO atmospheric concentration (black line, ±1σ in shaded area) and the calculated HONO concentration in model 6 using a min dew HONO emission $k_{emission}$ = 0.006 pptv %$^{-1}$ s$^{-1}$ (blue line) and max dew HONO emission $k_{emission}$ = 0.026 pptv %$^{-1}$ s$^{-1}$ (green line), respectively.

Section 4.3.3: see above, correct the 30% to <3 % and change all the discussion...

**Response:** The discussion was changed to "…α was calculated as $2 \times$LAI to take the areas on both sides of the leaves and the vegetation-covered areas on the ground into account. And a factor of 6 for LAI was assumed and used in section 4.2.2. However, regarding on the possibly different grass height during the HONO field measurement and dew measurements in April 2018and May 2019, respectively, we would use a range of 1-6 for LAI in this section. During the HONO peak at 6 or 7 UTC, the mixing height ranged between 175 m and 600 m, while the value ranged from 20 m to 200 m at 0:00 – 5:00 UTC. Hence, the overall concentration increase from this source would be 377-2264, 189-1132, 76-122, 38-226 and 13-76 pptv, if all of the deposited HONO is released into the overlying air column for a mixing height of 20, 40, 100, 200 and 600 m, respectively. Since the released HONO was subjected to photolysis, using a $J_{HONO}$ from TUV model scaled by global radiation (section 2.7), a maximum [HONO] of 176-1053, 88-527, 35-211, 18-105 and 6-35 pptv for the mixing height 20, 40, 100, 200 and 600 m, respectively, would be contributed from the surface nitrite release at 7:00 UTC after the process started from 4:00 UTC. For a reasonable 100 m mixing height, this would account for 5-30% of the observed HONO morning peak in Figure 6. This low percentage might be a result of the different sampling time of dew measurement compared with HONO measurement and further studies are required for the exact quantification. …" in section 4.3.3.

Line 722: "…direct estimation …" see many assumptions…

**Response:** "… direct evaluation" was changed to "direct estimation".

Table caption 3: "…during early nighttime."

**Response:** "… during nighttime" was corrected to "… during early nighttime".

Fig. 1: Check for the correct slope and the averaging time of the LOPAP (= time response MARGA).

**Response:** See above.

Figure caption 6: PSS

**Response:** "Pss" was corrected to "PSS" in the caption of Figure 6.

Figure 7: Specify which data is shown for the linear regressions (22th April for the low slope and 24th April for the high slope?)

**Response:** The caption of Figure 7 was corrected to "Example of $\frac{HONO_{unknown}}{99.5-RH}$ as a function of time (zero point from time 4:30, UTC) to estimate the temporary HONO emission rate from dew water ($k_{emission}$). Blue line is the linear least-square analysis of $\frac{HONO_{unknown}}{99.5-RH}$ vs. internal time to obtain the minimum (e.g. 22[th] April for the low slope) and maximum (e.g. 24[th] April for the high slope) of $k_{emission}$, respectively.".

SI Resistance model/Figure S9:

The authors should check the diffusion coefficient of HONO (the one for NO2 is correct, but I do not have data for HONO). Here the slightly heavier molecule HONO has a much high diffusion coefficient than NO2, which is not plausible (Is that data for low pressure?). Even the diffusion coefficient of NO (30 g/mol) is much lower than the value given for HONO. This will change all discussion on the transport limited uptake of HONO (the blue line will be ca. near the yellow line), which will get important at much more conditions (for gammas >10^-6...), also making the assumption of using the whole mixing layer for the HONO uptake more unreasonable.

Sorry I did not realize this point during the last review (massive work...).

**Response:** Hirokawa et al. (2008) investigated the uptake kinetics of gas phase HONO by a pH-controlled aqueous solution by using a wetted wall flow tube. They found that the uptake rate of the gaseous HONO depends on the pH of the solution. For the uptake by neutral (pH=7.1) and alkaline (pH=11.1) solutions, the gas phase concentration was observed to decay exponentially, suggesting that the uptake was fully limited by the gas phase diffusion. On the other hand, the uptake by the acidic solution was found to be determined by both the gas phase diffusion and the liquid phase processes such as physical absorption and reversible acid dissociation reaction. Finally, Hirokawa et al. (2008) determined the gas phase diffusion coefficient for HONO in the carrier gas mixtures ($Dg$) as $5.7\pm0.5\times10^{-5}$ m$^2$ s$^{-1}$ 294 K and 760 Torr, however, they assumed the aqueous phase diffusion coefficient of HONO, $D_{aq}$, to be equal to $D_{aq}$ for NO$_2$ at 298 K. In our previous version, we used the value of $7.2 \times10^{-5}$ at 660 torr as the study of VandenBoer et al. (2013). Now we recalculate the molecular diffusion resistance $R_b$ using the gas phase diffusion coefficient of $5.7\pm0.5\times10^{-5}$ m$^2$ s$^{-1}$ as shown in new Figure S9. As shown in new Figure S9, the current calculation will not change the discussion on the transport limited uptake of HONO.

In addition, as we already mentioned in the section of "Investigating resistance limitations in transport of HONO and NO$_2$ to the ground surface during the Melpitz measurement" of SI, the surface layer height was set to 15 m as example for nighttime values in Melpitz not the whole mixing layer.

[Figure]

**Figure S9.** Estimated contributions of resistance parameters to the observable ground surface processes for the HONO and $NO_2$ uptake values derived from Melpitz station. A series of grey shaded regions define the borders of the reactive uptake resistance ($R_c$), the $R_c$ values calculated from upper and lower limit uptake values of HONO and $NO_2$ in this work are shown in green and pink column, respectively. The aerodynamic transport resistance ($R_a$, red line) and diffusion resistance ($R_b$, blue line for HONO and yellow for $NO_2$) are shown in the Figure."

[revised manuscript text omitted]
_7[\text{OH}][\text{NO}] + k_{het}[\text{NO}_2] - J_{HONO}[\text{HONO}] - k_{11}[\text{HONO}][\text{OH}] \qquad \text{(Eq. 10)}$$

555 $k_{het}$ derived from this work is 0.027 h$^{-1}$, [NO] and [NO$_2$] are averaged concentrations from field measurement. The results are shown in Figure 6 (orange line, Model 1). It is reasonable to indicate that the reaction of R7 only contribute 30-55% to the HONO increase in the early morning (4:30-7:30 UTC). R7 can continually contribute 50% of the measured HONO from 10:30 to 16:30 (UTC). However, regarding on the large uncertainty of [OH] (a

560 factor of 2), the "unknown HONO sources" exist but could be not crucial. Basically, the additional HONO contribution rate could be estimated from the following equation:

$$\text{P}_{unknown} = \frac{d[HONO]}{dt} + J_{HONO}\,[\text{HONO}] + k_{11}[\text{OH}][\text{HONO}] - k_7[\text{OH}][\text{NO}] \qquad \text{(Eq. 11)}$$

An additional source of 91 ±41 pptv h$^{-1}$ was derived beside OH reaction with NO according to a HONO mixing ratio 98 ±15 pptv for the time period of 10:30 to 16:30 (UTC). This could be
565 well explained by the photochemical processes such as R5 and R9 and would be discussed deeply in the next section.

**4.3.2 Evidence for nighttime deposited HONO as a morning source**

As observed in our field measurement and shown in Figure 2, the HONO concentrations always presented a strong increase from 4:00 – 7:00 (UTC), which induces three hypotheses
570 as also mentioned in section 3.3: (a) photolysis of gas-phase and particulate nitrate, (b) photosensitized conversion of NO$_2$, (c) dew on ground surfaces served as HONO sink during the night and become a morning source by releasing the trapped nitrite back into ambient air. To identify this HONO source, the chemical box model as expressed in Eq. 12 was extended with additional processes. Heterogeneous reaction of NO$_2$ on the wet surface (R2) and HONO
575 deposition on the ground surface were firstly used to quantify the contributions of the well-known HONO production and loss processes. In addition, the HONO deposition on the ground surface independent on RH (24 hours, named Model 2) and with RH dependence (nighttime 17:00-8:00 UTC, named Model 3) are also discussed.

$$\frac{d[HONO]}{dt} = k_7[\text{OH}][\text{NO}] + k_{het}[\text{NO}_2] - J_{HONO}[\text{HONO}] - k_{11}[\text{HONO}][\text{OH}] -$$

[revised manuscript text omitted]